# Training-free Linear Image Inverses via Flows

**Ashwini Pokle** *
**Carnegie Mellon University**
`apokle@andrew.cmu.edu`

**Matthew J. Muckley**
**FAIR at Meta**
`mmuckley@meta.com`

**Ricky T. Q. Chen**
**FAIR at Meta**
`rtqichen@meta.com`

**Brian Karrer**
**FAIR at Meta**
`briankarrer@meta.com`

**Reviewed on OpenReview:** `https://openreview.net/forum?id=PLIt3a4yTm`

## Abstract

Solving inverse problems without any training involves using a pretrained generative model and making appropriate modifications to the generation process to avoid fine-tuning of the generative model. While recent methods have explored the use of diffusion models, they still require the manual tuning of many hyperparameters for different inverse problems. In this work, we propose a training-free method for solving linear inverse problems in non-blind setting by using pretrained flow models, leveraging the simplicity and efficiency of Flow Matching models, using theoretically-justified weighting schemes, and thereby significantly reducing the amount of manual tuning. In particular, we draw inspiration from two main sources: adopting prior gradient correction methods to the flow regime, and a solver scheme based on conditional Optimal Transport paths. As pretrained diffusion models are widely accessible, we also show how to practically adapt diffusion models for our method. Empirically, our approach requires no problem-specific tuning across an extensive suite of noisy linear inverse problems on high-dimensional datasets, ImageNet-64/128 and AFHQ-256, and we observe that our flow-based method for solving inverse problems improves upon closely-related diffusion-based methods in most settings.

## 1 Introduction

Solving an inverse problem involves recovering a clean signal from noisy measurements generated by a known degradation model. Many interesting image processing tasks can be cast as an inverse problem. Some instances of these problems are super-resolution, inpainting, deblurring, colorization, denoising etc. Diffusion models or score-based generative models (Sohl-Dickstein et al., 2015; Ho et al., 2020; Song & Ermon, 2019; Song et al., 2021d) have emerged as a leading family of generative models for solving inverse problems for images (Saharia et al., 2022b;a; Wang et al., 2022; Chung et al., 2022a; Song et al., 2022; Mardani et al., 2023). However, sampling with diffusion models is known to be slow, and the quality of generated images is affected by the curvature of SDE/ODE solution trajectories (Karras et al., 2022). While Karras et al. (2022) observed ODE sampling for image generation could produce better results, sampling via SDE is still common for solving inverse problems, whereas ODE sampling has been rarely considered, perhaps due to the use of diffusion probability paths.

---

*Work done during internship with FAIR at Meta

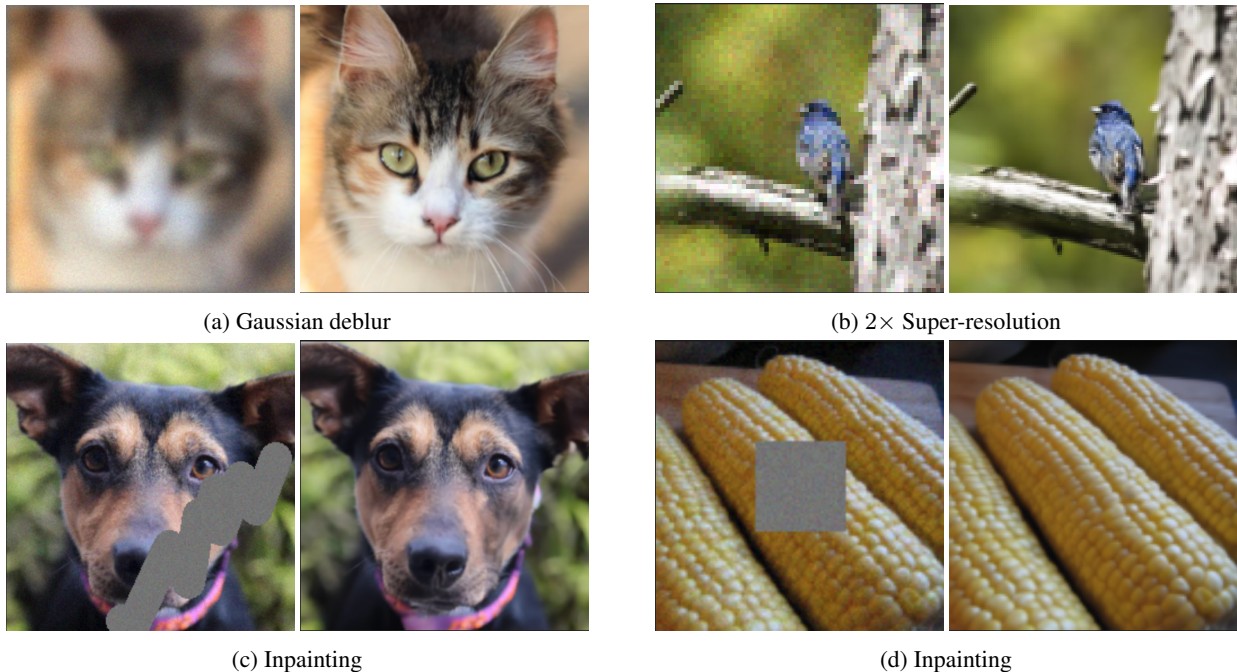

(a) Gaussian deblur

(b) 2× Super-resolution

(c) Inpainting

(d) Inpainting

Figure 1: Corrected images by solving linear inverse problems with flow models. For each pair of images, we show the noisy measurement (left) and the reconstruction (right).

Continuous Normalizing Flow (CNF) (Chen et al., 2018b) trained with Flow Matching (Lipman et al., 2022) has been recently proposed as a powerful alternative to diffusion models. CNF (hereafter denoted flow model) has the ability to model arbitrary probability paths, and includes diffusion probability paths as a special case. Of particular interest to us are Gaussian probability paths that correspond to optimal transport (OT) displacement (McCann, 1997). Recent works (Lipman et al., 2022; Albergo & Vanden-Eijnden, 2022; Liu et al., 2022; Shaul et al., 2023) have shown that these conditional OT probability paths are straighter than diffusion paths, which results in faster training and sampling with these models. Due to these properties, conditional OT flow models are an appealing alternative to diffusion models for solving inverse problems.

In this work, we introduce a training-free method to utilize pretrained unconditional flow models for solving linear inverse problems. Our approach adds a gradient-based adaptation term to the unconditional vector field that takes into account knowledge from the degradation model and converts it to a conditional vector field. Specifically, we introduce an algorithm that extends ΠGDM (Song et al., 2022) gradient adaptation to ODE sampling with an affine Gaussian probability path. Given the wide availability of pretrained diffusion models trained with diffusion probability paths, we also present a way to convert these models to other affine Gaussian probability paths. Empirically, we observe images restored via a conditional OT path exhibit perceptual quality better than that achieved by the model's original diffusion path, as well as recently proposed diffusion approaches such as ΠGDM (Song et al., 2022) and RED-Diff (Mardani et al., 2023), particularly in noisy settings. To summarize, our key contributions are:

- We present a training-free approach to solve linear inverse problems in non-blind setting that can be applied to any continuous-time diffusion or flow model under affine Gaussian probability paths that extends ΠGDM gradient adaptation to this more generic setting.

- We explain the subtleties in converting models between different affine Gaussian probability paths. Specifically, we enable the use of pre-trained continuous-time diffusion models with conditional OT probability paths by an adjusted initialization procedure.

- We demonstrate that images restored via our ODE algorithm using conditional OT probability paths have perceptual quality that is largely on par with, or better than that achieved by diffusion probability paths, and other recent methods like ΠGDM and RED-Diff, without the need for problem-specific hyperparameter tuning.

## 2 Preliminaries

We introduce relevant background knowledge and notation from conditional diffusion and flow modeling, as well as training-free inference with diffusion models.

**Notation.** Both diffusion and flow models consider two distinct processes indexed by time between $[0, 1]$ that convert data to noise and noise to data. Here, we follow the temporal notation used in prior work (Lipman et al., 2022) where the distribution at $t = 1$ is the data distribution and $t = 0$ is a standard Gaussian distribution. Note that this is opposite of the prevalent notation used in diffusion model literature (Song & Ermon, 2019; Ho et al., 2020; Song et al., 2021a;d). We let $\boldsymbol{x}_t$ denote a real-valued vector at time $t$, without regard to which process (*i.e.,* diffusion or flow) it was drawn from. The probability density for the data to noise process is denoted $q$ and the parameterized probability density for the noise to data process is denoted $p_\theta$. Expectations with respect to $q$ are denoted via $\mathbb{E}_q$, where the relevant random variables are noisy $\boldsymbol{x}_t$, clean data $\boldsymbol{x}_1$, and conditioning $\boldsymbol{y}$ with density $q(\boldsymbol{x}_t, \boldsymbol{x}_1, \boldsymbol{y}) = q(\boldsymbol{x}_t | \boldsymbol{x}_1, \boldsymbol{y}) q(\boldsymbol{x}_1, \boldsymbol{y})$. Here $q(\boldsymbol{x}_1, \boldsymbol{y})$ is unknown and $q(\boldsymbol{x}_t | \boldsymbol{x}_1, \boldsymbol{y})$ will be a modeling choice. Conditional diffusion or flow models aim to produce samples $\boldsymbol{x}_1 \sim q(\boldsymbol{x}_1 | \boldsymbol{y})$. We generally keep function arguments of $t$ implicit (i.e. $f(\boldsymbol{x}_t, t)$ is informally written as $f(\boldsymbol{x}_t)$.)

**Conditional diffusion models.** Suppose we have samples $\boldsymbol{x}_1$ (e.g. an image) and conditioning $\boldsymbol{y}$ (e.g. a distorted image) drawn from a data distribution $q(\boldsymbol{x}_1, \boldsymbol{y})$. Conditional diffusion models use latent variables $\boldsymbol{x}_{0:1} = \{\boldsymbol{x}_t | t \in [0, 1)\}$ to model the joint distribution $p_\theta(\boldsymbol{x}_{0:1}, \boldsymbol{x}_1 | \boldsymbol{y})$ for the noise to data process $p_\theta$.[1] The data to noise process $q$ approximates the posterior $q(\boldsymbol{x}_{0:1} | \boldsymbol{x}_1, \boldsymbol{y})$ and is defined as a Markov chain[2] that adds Gaussian noise to data, which in continuous time satisfies a stochastic differential equation (SDE)(Song & Ermon, 2019; Ho et al., 2020; Song et al., 2021d). The parameters of $p_\theta$ are learned via minimizing a regression loss derived from the variational bound on negative log-likelihood with respect to $\widehat{\boldsymbol{x}_1}$:

$$L_{\text{diffusion}}(\widehat{\boldsymbol{x}_1}) = \int_0^1 w(t) \mathbb{E}_{\boldsymbol{x}_t \sim q(\boldsymbol{x}_t | \boldsymbol{x}_1, \boldsymbol{y}), \boldsymbol{x}_1, \boldsymbol{y} \sim q(\boldsymbol{x}_1, \boldsymbol{y})} \left[ \|\widehat{\boldsymbol{x}_1}(\boldsymbol{x}_t, \boldsymbol{y}) - \boldsymbol{x}_1\|^2 \right] dt, \tag{1}$$

where $w(t)$ are positive weights (Kingma et al., 2021; Song et al., 2021b; Kingma & Gao, 2023), and $\widehat{\boldsymbol{x}_1}$ is a deterministic parametrized denoiser. The optimal solution for $\widehat{\boldsymbol{x}_1}$ with the squared $L_2$ error in Eq. (1) is $\mathbb{E}_{\boldsymbol{x}_1 \sim q(\boldsymbol{x}_1 | \boldsymbol{x}_t, \boldsymbol{y})}[\boldsymbol{x}_1 | \boldsymbol{x}_t, \boldsymbol{y}]$. For brevity as well as ease of readability, henceforth we will typically write the expectations with respect to $q$ such as the one that appears in Eq. (1) in short as $\mathbb{E}_q$. Many equivalent parameterizations exist for the loss in Eq. (1) and have known conversions to denoising. Sampling using $p_\theta$ proceeds by starting from $\boldsymbol{x}_0 \sim p_\theta(\boldsymbol{x}_0 | \boldsymbol{y})$ and integrating the SDE using $\widehat{\boldsymbol{x}_1}$ to $t = 1$. If $p_\theta(\boldsymbol{x}_0 | \boldsymbol{y}) = q(\boldsymbol{x}_0 | \boldsymbol{y})$, the SDE is integrated exactly, and $\widehat{\boldsymbol{x}_1}(\boldsymbol{x}_t, \boldsymbol{y}) = \mathbb{E}_q[\boldsymbol{x}_1 | \boldsymbol{x}_t, \boldsymbol{y}]$, the resulting $\boldsymbol{x}_1 \sim q(\boldsymbol{x}_1 | \boldsymbol{y})$ as desired.

**Conditional flow models** Alternatively, continuous normalizing flow models (Chen et al., 2018b) define the data generation process through an ODE. This leads to simpler formulations and does not introduce extra noise during intermediate steps of sample generation. Recently, simulation-free training algorithms have been designed specifically for such models (Lipman et al., 2022; Liu et al., 2022; Albergo & Vanden-Eijnden, 2022), an example being the Conditional Flow Matching loss (Lipman et al., 2022),

$$L_{\text{cfm}}(\widehat{\boldsymbol{v}}) = \int_0^1 \mathbb{E}_{\boldsymbol{x}_t \sim q(\boldsymbol{x}_t | \boldsymbol{x}_1, \boldsymbol{y}), \boldsymbol{x}_1, \boldsymbol{y} \sim q(\boldsymbol{x}_1, \boldsymbol{y})} \left[ \|\widehat{\boldsymbol{v}}(\boldsymbol{x}_t, \boldsymbol{y}) - \boldsymbol{v}(\boldsymbol{x}_t, \boldsymbol{y}, \boldsymbol{x}_1)\|^2 \right] dt, \tag{2}$$

where $\widehat{\boldsymbol{v}}(\boldsymbol{x}_t, \boldsymbol{y})$ denotes a parameterized vector field defining the ODE

$$\frac{d\boldsymbol{x}_t}{dt} = \widehat{\boldsymbol{v}}(\boldsymbol{x}_t, \boldsymbol{y}), \tag{3}$$

and data-conditional vector field $\boldsymbol{v}(\boldsymbol{x}_t, \boldsymbol{y}, \boldsymbol{x}_1)$ is determined by modeling choice $q(\boldsymbol{x}_t | \boldsymbol{x}_1, \boldsymbol{y})$. If trained perfectly, the marginal distributions of $\boldsymbol{x}_t$ from ODE integration, denoted $p_\theta(\boldsymbol{x}_t | \boldsymbol{y})$, will match the marginal distributions of $q(\boldsymbol{x}_t | \boldsymbol{y})$. Hence sampling from $q(\boldsymbol{x}_1 | \boldsymbol{y})$ as desired can be achieved by sampling initial value $\boldsymbol{x}_{t'} \sim q(\boldsymbol{x}_{t'} | \boldsymbol{y})$ and integrating the ODE from $t'$ to 1. Typically, one samples from $t' = 0$ since $q(\boldsymbol{x}_0 | \boldsymbol{y})$ is a tractable distribution.

---

[1]In discrete time, we can write this joint distribution as $p_\theta(\boldsymbol{x}_{0:1} | \boldsymbol{y}) = p_\theta(\boldsymbol{x}_0 | \boldsymbol{y}) \prod_t p_\theta(\boldsymbol{x}_{t+\Delta} | \boldsymbol{x}_t, \boldsymbol{y})$ where $\Delta$ denotes an appropriate step size of time discretization in $[0, 1]$.

[2]In discrete time, the forward process is $q(\boldsymbol{x}_{0:1} | \boldsymbol{x}_1, \boldsymbol{y}) = \prod_t q(x_{t-\Delta} | \boldsymbol{x}_t, \boldsymbol{y})$

**Gaussian probability paths.** The time-dependent densities $q(\boldsymbol{x}_t|\boldsymbol{x}_1, \boldsymbol{y})$ are referred to as conditional probability paths. We focus on the class of affine Gaussian probability paths of the form

$$q(\boldsymbol{x}_t|\boldsymbol{x}_1, \boldsymbol{y}) = q(\boldsymbol{x}_t|\boldsymbol{x}_1) = \mathcal{N}(\alpha_t \boldsymbol{x}_1, \sigma_t^2 \boldsymbol{I}), \tag{4}$$

where non-negative $\alpha_t$ and $\sigma_t$ are monotonically increasing and decreasing respectively. This class includes the probability paths for conditional diffusion as well as the conditional Optimal Transport (OT) path (Lipman et al., 2022), where $\alpha_t = t$ and $\sigma_t = 1 - t$. The conditional OT path used by flow models has been demonstrated to have good empirical properties, including faster inference and better sampling in practice, and has theoretical support in high-dimensions (Shaul et al., 2023). As emphasized in Lin et al. (2023), a desirable property for probability paths, obeyed by conditional OT but not commonly used diffusion paths, is to ensure $q(\boldsymbol{x}_0|\boldsymbol{y})$ is known (i.e. $\mathcal{N}(0, \boldsymbol{I})$), as otherwise one cannot exactly sample $\boldsymbol{x}_0$ which can add substantial error. When using these affine Gaussian probability paths with a conditional flow model, one sets (Lipman et al., 2022)

$$\boldsymbol{v}(\boldsymbol{x}_t, \boldsymbol{y}, \boldsymbol{x}_1) = \frac{d\alpha_t}{dt} \boldsymbol{x}_1 + \frac{d\sigma_t}{dt} \left( \frac{\boldsymbol{x}_t - \alpha_t \boldsymbol{x}_1}{\sigma_t} \right). \tag{5}$$

**Converting between flow and diffusion models.** In our framing for affine Gaussian probability paths, a model is identified as flow or diffusion by whether an ODE or SDE is used for sampling respectively. For this class of paths though, we can convert directly between flow and diffusion models. To see this, note that the optimal $\boldsymbol{v}(\boldsymbol{x}_t, \boldsymbol{y})$ for the Conditional Flow Matching loss in Eq. (2) is $\mathbb{E}_q[\boldsymbol{v}(\boldsymbol{x}_t, \boldsymbol{y}, \boldsymbol{x}_1)|\boldsymbol{x}_t, \boldsymbol{y}]$, which for affine Gaussian probability paths using Eq. 5 is

$$\boldsymbol{v}(\boldsymbol{x}_t, \boldsymbol{y}) = \frac{d\alpha_t}{dt} \mathbb{E}_q[\boldsymbol{x}_1|\boldsymbol{x}_t, \boldsymbol{y}] + \frac{d\sigma_t}{dt} \left( \frac{\boldsymbol{x}_t - \alpha_t \mathbb{E}_q[\boldsymbol{x}_1|\boldsymbol{x}_t, \boldsymbol{y}]}{\sigma_t} \right). \tag{6}$$

This equivalence has been noted by Karras et al. (2022), leveraging a more complex conversion from SDE to probability flow ODE from Song et al. (2021d). Rearranging Eq. (6), a diffusion model's denoiser $\widehat{\boldsymbol{x}_1}(\boldsymbol{x}_t, \boldsymbol{y})$ trained using affine Gaussian probability path $q$ can be interchanged with a flow model's $\widehat{\boldsymbol{v}}(\boldsymbol{x}_t, \boldsymbol{y})$ with the same path via

$$\widehat{\boldsymbol{v}} = \left( \alpha_t \frac{d\ln(\alpha_t/\sigma_t)}{dt} \right) \widehat{\boldsymbol{x}_1} + \frac{d\ln\sigma_t}{dt} \boldsymbol{x}_t. \tag{7}$$

**Training-free conditional inference using unconditional diffusion.** Given pretrained *unconditional* diffusion models that are trained to approximate $\mathbb{E}_q[\boldsymbol{x}_1|\boldsymbol{x}_t]$, training-free approaches for conditional inference aim to approximate $\mathbb{E}_q[\boldsymbol{x}_1|\boldsymbol{x}_t, \boldsymbol{y}]$ without any fine-tuning of the unconditional model. Under affine Gaussian probability paths, the two terms are related by Tweedie's identity (Robbins, 1992) which expresses $\mathbb{E}_q[\boldsymbol{x}_1|\boldsymbol{x}_t, \boldsymbol{y}] = (\boldsymbol{x}_t + \sigma_t^2 \nabla_{\boldsymbol{x}_t} \ln q(\boldsymbol{x}_t|\boldsymbol{y}))/\alpha_t$. Applying this identity (twice for both $\mathbb{E}_q[\boldsymbol{x}_1|\boldsymbol{x}_t, \boldsymbol{y}]$ and $\mathbb{E}_q[\boldsymbol{x}_1|\boldsymbol{x}_t]$) and simplifying gives

$$\mathbb{E}_q[\boldsymbol{x}_1|\boldsymbol{x}_t, \boldsymbol{y}] = \mathbb{E}_q[\boldsymbol{x}_1|\boldsymbol{x}_t] + \frac{\sigma_t^2}{\alpha_t} \nabla_{\boldsymbol{x}_t} \ln q(\boldsymbol{y}|\boldsymbol{x}_t). \tag{8}$$

Following Eq. 8, past approaches (*e.g.,* Chung et al. (2022a); Song et al. (2022)) have used the pretrained model for the first term and approximated the second intractable term to produce an approximate $\widehat{\boldsymbol{x}_1}(\boldsymbol{x}_t, \boldsymbol{y})$. Diffusion posterior sampling (DPS) (Chung et al., 2022a) proposed to approximate $q(\boldsymbol{y}|\boldsymbol{x}_t)$ via $q(\boldsymbol{y}|\boldsymbol{x}_1 = \widehat{\boldsymbol{x}_1}(\boldsymbol{x}_t))$. Later, Pseudo-inverse Guided Diffusion Models (ΠGDM) (Song et al., 2022) improved upon DPS for linear noisy observations where $\boldsymbol{y} = \boldsymbol{A}\boldsymbol{x} + \sigma_y\epsilon$, where $\boldsymbol{A}$ is some measurement matrix and $\epsilon \sim \mathcal{N}(0, \boldsymbol{I})$, by approximating $q(\boldsymbol{y}|\boldsymbol{x}_t)$ as $\mathcal{N}(\boldsymbol{A}\widehat{\boldsymbol{x}_1}(\boldsymbol{x}_t), \sigma_{\boldsymbol{y}}^2 \boldsymbol{I} + r_t^2 \boldsymbol{A}\boldsymbol{A}^T)$, derived via first approximating $q(\boldsymbol{x}_1|\boldsymbol{x}_t)$ as $\mathcal{N}(\widehat{\boldsymbol{x}_1}(\boldsymbol{x}_t), r_t^2 \boldsymbol{I})$, where $r_t$ is an appropriate time-dependent standard deviation. ΠGDM also suggested adaptive weighting, replacing $\sigma_t^2/\alpha_t$ with another function of time to account for the approximation. While these past approaches have used Eq. (8) for diffusion probability paths, this equation is valid for any affine Gaussian probability path.

## 3 Solving Linear Inverse Problems without Conditional Training via Flows

In the standard setup of a linear inverse problem, we observe measurements $\boldsymbol{y} \in \mathbb{R}^n$ such that

$$\boldsymbol{y} = \boldsymbol{A}\boldsymbol{x}_1 + \epsilon \tag{9}$$

where $\boldsymbol{x}_1 \in \mathbb{R}^m$ is drawn from an unknown data distribution $q(\boldsymbol{x}_1)$, $\boldsymbol{A} \in \mathbb{R}^{n \times m}$ is a known measurement matrix, and $\epsilon \sim \mathcal{N}(0, \sigma_y^2 \boldsymbol{I})$ is unknown *i.i.d.* Gaussian noise with known standard deviation $\sigma_y$. Given a pretrained flow model with $\widehat{\boldsymbol{v}}(\boldsymbol{x}_t)$ that can sample from $q(\boldsymbol{x}_1)$, and measurements $\boldsymbol{y}$, our goal is to produce clean samples from the posterior $q(\boldsymbol{x}_1|\boldsymbol{y}) \propto q(\boldsymbol{y}|\boldsymbol{x}_1)q(\boldsymbol{x}_1)$ without training a problem-specific conditional flow model defined by $\widehat{\boldsymbol{v}}(\boldsymbol{x}_t, \boldsymbol{y})$. In this section, we motivate and propose our approach to solving this problem using flows.

## 3.1 Adapting the vector field of unconditional flow models for conditional sampling

To solve linear inverse problems without any training via flow models, we derive an expression similar to Eq. 8 that relates conditional vector fields under Gaussian probability paths to unconditional vector fields.

Let $q$ be a Gaussian probability path described by Eq. 4. Assume we observe $\boldsymbol{y} \sim q(\boldsymbol{y}|\boldsymbol{x}_1)$ for arbitrary $q(\boldsymbol{y}|\boldsymbol{x}_1)$ and $\boldsymbol{v}(\boldsymbol{x}_t)$ is a vector field enabling sampling $\boldsymbol{x}_t \sim q(\boldsymbol{x}_t)$. Note that Eq. (6) without $\boldsymbol{y}$ also holds for optimal unconditional $\boldsymbol{v}(\boldsymbol{x}_t)$. Inserting Eq. (8) into Eq. (6) and taking the difference $(\boldsymbol{v}(\boldsymbol{x}_t, \boldsymbol{y}) - \boldsymbol{v}(\boldsymbol{x}_t))$ yields

$$\boldsymbol{v}(\boldsymbol{x}_t, \boldsymbol{y}) = \boldsymbol{v}(\boldsymbol{x}_t) + \sigma_t^2 \frac{d\ln(\alpha_t/\sigma_t)}{dt} \nabla_{\boldsymbol{x}_t} \ln q(\boldsymbol{y}|\boldsymbol{x}_t). \tag{10}$$

We use Eq. (10) in our training-free algorithm for solving linear inverse using flows by incorporating ΠGDM's adaptation. In particular, given $\widehat{\boldsymbol{v}}(\boldsymbol{x}_t)$ (or $\widehat{\boldsymbol{x}_1}(\boldsymbol{x}_t)$), our approximation will be

$$\widehat{\boldsymbol{v}}(\boldsymbol{x}_t, \boldsymbol{y}) = \widehat{\boldsymbol{v}}(\boldsymbol{x}_t) + \sigma_t^2 \frac{d\ln(\alpha_t/\sigma_t)}{dt} \gamma_t \nabla_{\boldsymbol{x}_t} \ln q^{app}(\boldsymbol{y}|\boldsymbol{x}_t), \tag{11}$$

where $q^{app}(\boldsymbol{y}|\boldsymbol{x}_t)$ denotes an approximation for $q(\boldsymbol{y}|\boldsymbol{x}_t)$ and $\gamma_t$ denotes time-dependent weights. We refer to $\gamma_t = 1$ as unadaptive and other choices as adaptive weights. In general, we view adaptive weights $\gamma_t \neq 1$ as an adjustment for error in $q^{app}(\boldsymbol{y}|\boldsymbol{x}_t)$.

**Approximating $q(\boldsymbol{y}|\boldsymbol{x}_t)$.** The update for adapting the unconditional vector field in Eq. (10) requires $q(\boldsymbol{y}|\boldsymbol{x}_t)$ which is intractable to compute as it involves marginalization over $\boldsymbol{x}_1$

$$q(\boldsymbol{y}|\boldsymbol{x}_t) = \int_{\boldsymbol{x}_1} q(\boldsymbol{y}|\boldsymbol{x}_1)q(\boldsymbol{x}_1|\boldsymbol{x}_t)d\boldsymbol{x}_1. \tag{12}$$

In this equation, the first term $q(\boldsymbol{y}|\boldsymbol{x}_1)$ is tractable as it is equal to $\mathcal{N}(\boldsymbol{A}\boldsymbol{x}_1, \sigma_y^2\boldsymbol{I})$. However, it is computationally expensive to estimate the second term $q(\boldsymbol{x}_1|\boldsymbol{x}_t)$ with a flow model or a diffusion model. We therefore use an approximation for $q(\boldsymbol{y}|\boldsymbol{x}_t)$ and refer to this approximation as $q^{app}(\boldsymbol{y}|\boldsymbol{x}_t)$. Following ΠGDM, we set

$$q(\boldsymbol{x}_1|\boldsymbol{x}_t) \approx \mathcal{N}(\widehat{\boldsymbol{x}_1}(\boldsymbol{x}_t), r_t^2 I). \tag{13}$$

where $r_t$ is an appropriately chosen time dependent standard deviation. We can now compute $q^{app}(\boldsymbol{y}|\boldsymbol{x}_t)$ in closed form as $q^{app}(\boldsymbol{y}|\boldsymbol{x}_t) \approx \mathcal{N}(\boldsymbol{A}\widehat{\boldsymbol{x}_1}(\boldsymbol{x}_t), \sigma_y^2 I + r_t^2 \boldsymbol{A}\boldsymbol{A}^\top)$ which gives the following approximation for $\nabla_{\boldsymbol{x}_t} \ln q^{app}(\boldsymbol{y}|\boldsymbol{x}_t)$:

$$\nabla_{\boldsymbol{x}_t} \ln q^{app}(\boldsymbol{y}|\boldsymbol{x}_t) \approx (\boldsymbol{y} - \boldsymbol{A}\widehat{\boldsymbol{x}_1})^\top (r_t^2 \boldsymbol{A}\boldsymbol{A}^\top + \sigma_y^2 \boldsymbol{I})^{-1} \boldsymbol{A} \frac{\partial \widehat{\boldsymbol{x}_1}}{\partial \boldsymbol{x}_t}. \tag{14}$$

Note that this is a vector-Jacobian product and can computed efficiently with packages for automatic differentiation. With this we generalize ΠGDM to any Gaussian probability path described by Eq. 4 by using an alternate $r_t^2$. We choose $r_t^2$ by following ΠGDM's derivation which assumes that $q(\boldsymbol{x}_1)$ is $\mathcal{N}(0, \boldsymbol{I})$ to derive $r_t^2$. We have $q(\boldsymbol{x}_t|\boldsymbol{x}_1) = \mathcal{N}(\alpha_t \boldsymbol{x}_1, \sigma_t^2 I)$. Thus by Bayes' rule, we can write the posterior as

$$q(\boldsymbol{x}_1|\boldsymbol{x}_t) \propto q(\boldsymbol{x}_1)q(\boldsymbol{x}_t|\boldsymbol{x}_1) = \mathcal{N}\left(\frac{\alpha_t \boldsymbol{x}_t}{\alpha_t^2 + \sigma_t^2}, \frac{\sigma_t^2}{\alpha_t^2 + \sigma_t^2}\boldsymbol{I}\right). \tag{15}$$

With this, we approximate $r_t^2$ as

$$r_t^2 = \frac{\sigma_t^2}{\sigma_t^2 + \alpha_t^2}. \tag{16}$$

When $\alpha_t = 1$, we recover ΠGDM's $r_t^2$ as expected under their Variance-Exploding path specification.

## 3.2 Converting between affine Gaussian probability paths

To complete our derivation, we demonstrate how one can train with path $q'$ and perform sampling with alternative path $q$. This conversion is crucial to enable sampling with any probability path, including particularly the conditional OT probability path, without training given an existing pre-trained model. Such conversions have been noted previously by Karras et al. (2022) using an SDE perspective. Our derivation exposes important subtleties when converting between affine Gaussian probability paths.

Consider two affine Gaussian probability paths, with joint densities $q$ and $q'$, defined by Eq. 4 with $\alpha_t$, $\sigma_t$ and $\alpha'_t$, $\sigma'_t$ respectively. Define $t'(t)$ as the unique solution to $\alpha_t/\sigma_t = \alpha'_{t'}/\sigma'_{t'}$ when it exists for given $t$. The solution for $t'(t)$ is unique due to the monotonicity requirements of both $\alpha$ and $\sigma$. By definition, joint densities $q$ and $q'$ share the same distribution over data $\boldsymbol{x}_1$, $q(\boldsymbol{x}_1|\boldsymbol{y}) = q'(\boldsymbol{x}_1|\boldsymbol{y})$. Then for affine Gaussian probability paths, $q(\boldsymbol{X}_t = \boldsymbol{x}_t|\boldsymbol{x}_1, \boldsymbol{y}) = q'(\boldsymbol{X}'_{t'(t)} = \alpha'_{t'(t)}\boldsymbol{x}_t/\alpha_t|\boldsymbol{x}_1, \boldsymbol{y})$ when $t'(t)$ exists. Since the joint densities are identical, the conditional distributions over $\boldsymbol{x}_1$ used by the optimal denoiser and vector fields are also identical at these values.

So $\widehat{\boldsymbol{x}_1}$ trained under $q'$ can be used for sampling under $q$ via evaluating at $\widehat{\boldsymbol{x}_1}(\alpha'_{t'(t)}\boldsymbol{x}_t/\alpha_t, t'(t), \boldsymbol{y})$ (with explicit time for clarity) whenever $t'(t)$ exists, with identical argument changes for vector fields. In particular, if sampling uses the conditional OT probability path, we have

$$t'(t) = \text{SNR}_{q'}^{-1}(\text{SNR}_q(t)) = \text{SNR}_{q'}^{-1}\left(\frac{t}{1-t}\right). \tag{17}$$

where signal-to-noise ratio $\text{SNR}(t) = \alpha_t/\sigma_t$. The main avenue for non-existence for $t'(t)$ is if the model under $q'$ is trained using a minimum SNR above zero, which induces a minimum $t$ for which $t'(t)$ exists. When a minimum $t$ exists, we can only perform sampling with $q$ starting from $\boldsymbol{x}_t \sim q(\boldsymbol{x}_t|\boldsymbol{y})$. Approximating this sample is entirely analogous to approximating $\boldsymbol{x}_0 \sim q'(\boldsymbol{x}_0|\boldsymbol{y})$. This error already exists for $q'$ because $q'(\boldsymbol{x}_0|\boldsymbol{y})$ is not $\mathcal{N}(0, \boldsymbol{I})$ unless $q'$ is trained to zero SNR. An initialization problem cannot be avoided if $q'$ has limited SNR range by switching paths to $q$. This problem is relevant when converting pre-trained diffusion models as typical diffusion paths have a nonzero minimum SNR.

## 3.3 Our algorithm for solving linear inverse problems

**Starting flow sampling at time $t > 0$.** Initializing conditional diffusion model sampling at $t > 0$ has been proposed by Chung et al. (2022c). For flows, we similarly want $\boldsymbol{x}_t \sim q(\boldsymbol{x}_t|\boldsymbol{y})$ at initialization time $t$. In our experiments, we examine (approximately) initializing at different times $t > 0$ using

$$\boldsymbol{x}_t = \alpha_t \boldsymbol{y} + \sigma_t \epsilon \tag{18}$$

for $\epsilon \sim \mathcal{N}(0, \boldsymbol{I})$ when $\boldsymbol{y}$ is the correct shape. For super-resolution, we use nearest-neighbor interpolation on $\boldsymbol{y}$ instead. We also consider using $\boldsymbol{A}^\dagger y$ as an ablation in the Appendix B (where $\boldsymbol{A}^\dagger$ is the pseudo-inverse of $\boldsymbol{A}$ (Song et al., 2022)). We may be forced to use this initialization for flow sampling due to converting a diffusion model not trained to zero SNR. However as shown in (Chung et al., 2022c) for diffusion, this initialization can improve results more generally. Conceptually, if the resulting $\boldsymbol{x}_t$ is closer to $\boldsymbol{x}_t \sim q(\boldsymbol{x}_t|\boldsymbol{y})$ than achieved via starting from an earlier time $t'$ and integrating, then this initialization can result in less overall error.

**Algorithm summary.** Putting this altogether, our proposed approach using flow sampling and conditional OT probability paths is succinctly summarized in Algorithm 1, derived via inserting $\alpha_t = t$ and $\sigma_t = 1 - t$. This algorithm uses the unconditional vector field adaptation proposed in Eq. (11) and uses this vector field adaptation $\hat{\boldsymbol{v}}_{\text{adapted}}$ to integrate the ODE from some initial time $t_0$ to 1 to get the final corrected image $\boldsymbol{x}_1$. We can integrate the ODE by using any standard numerical methods such as Euler method, Runge-Kutta method etc. As an example, an intermediate update of Euler method from time $t$ to $t + \Delta t$ during ODE integration is given by $\boldsymbol{x}_{t+\Delta t} = \boldsymbol{x}_t + \hat{\boldsymbol{v}}_{\text{adapted}}\Delta t$. Unlike $\Pi$GDM, we propose unadaptive weights $\gamma_t = 1$. By default, we set initialization time $t_0 = 0.2$. The algorithm therefore has no additional hyperparameters to tune over traditional diffusion or flow sampling. In Appendix A, we detail our algorithm for other Gaussian probability paths, and the equivalent formulation when a pretrained vector field is available instead.

---

**Algorithm 1** Solving linear inverse problems via flows using conditional OT probability path

---

**Require:** Pretrained denoiser $\widehat{\boldsymbol{x}_1}(\boldsymbol{x}_t)$ converted to conditional OT probability path, noisy measurement $\boldsymbol{y}$, measurement matrix $\boldsymbol{A}$, initial time $t_0$, and std $\sigma_y$

1: Initialize $\boldsymbol{x}_{t_0} = t_0\mathbf{y} + (1-t_0)\epsilon$, where $\epsilon \sim \mathcal{N}(0, \boldsymbol{I})$               ▷ Initialize $\boldsymbol{x}_t$, Eq. (18)

2: $\boldsymbol{x}_t = \boldsymbol{x}_{t_0}$

3: **for** each time step $t$ of ODE integration **do**               ▷ Integrate ODE from $t = t_0$ to 1.

4:     $r_t^2 = \frac{(1-t)^2}{t^2+(1-t)^2}$            ▷ Value of $r_t^2$ from Eq. (16)

5:     $\widehat{\boldsymbol{v}} = \frac{\widehat{\boldsymbol{x}_1}(\boldsymbol{x}_t)-\boldsymbol{x}_t}{1-t}$          ▷ Convert $\widehat{\boldsymbol{x}_1}$ to vector field, Eq. (7)

6:     $\boldsymbol{g} = (\boldsymbol{y} - \boldsymbol{A}\widehat{\boldsymbol{x}_1})^\top (r_t^2 \boldsymbol{A}\boldsymbol{A}^\top + \sigma_y^2 \boldsymbol{I})^{-1}\boldsymbol{A}\frac{\partial \widehat{\boldsymbol{x}_1}}{\partial \boldsymbol{x}_t}$     ▷ $\nabla_{\boldsymbol{x}_t} \ln q^{app}(y|\boldsymbol{x}_t)$

7:     $\widehat{\boldsymbol{v}}_{\text{adapted}} = \widehat{\boldsymbol{v}} + \frac{1-t}{t}\boldsymbol{g}$        ▷ Adapt the unconditional vector field $\widehat{\boldsymbol{v}}$, Eq. (11)

8:     $\boldsymbol{x}_{t+\Delta t} = \text{ODESolverStep}(\boldsymbol{x}_t, \widehat{\boldsymbol{v}}_{\text{adapted}})$     ▷ One step of ODE solver to update $\boldsymbol{x}_t$

9: **end for**

10: **return** $\boldsymbol{x}_1$           ▷ This is the solution of ODE integration.

---

## 4 Experiments

**Datasets.** We verify the effectiveness of our proposed approach on three datasets: face-blurred ImageNet $64 \times 64$ and $128 \times 128$ (Deng et al., 2009; Russakovsky et al., 2015; Yang et al., 2022), and AnimalFacesHQ (AFHQ) $256 \times 256$ (Choi et al., 2020). We report our results on 10K randomly sampled images from validation split of ImageNet, and 1500 images from test split of AFHQ.

**Tasks.** We report results on the following linear inverse problems: inpainting (center-crop), Gaussian deblurring, super-resolution, and denoising. The exact details of the measurement operators are: 1) For inpainting, we use centered mask of size $20 \times 20$ for ImageNet-64, $40 \times 40$ for ImageNet-128, and $80 \times 80$ for AFHQ. In addition, for images of size $256 \times 256$, we also use free-form masks simulating brush strokes similar to the ones used in Saharia et al. (2022a); Song et al. (2022). 2) For super-resolution, we apply bicubic interpolation to downsample images by $4\times$ for datasets that have images with resolution $256 \times 256$ and downsample images by $2\times$ otherwise. 3) For Gaussian deblurring, we apply Gaussian blur kernel of size $61 \times 61$ with standard deviation of 1 for ImageNet-64 and ImageNet-128, and $61 \times 61$ with standard deviation of 3 for AFHQ. 4) For denoising, we add *i.i.d.* Gaussian noise with $\sigma_y = 0.05$ to the images. For tasks besides denoising, we consider *i.i.d.* Gaussian noise with $\sigma_y = 0$ and $0.05$ to the images. Images $\boldsymbol{x}_1$ are normalized to range $[-1, 1]$.

**Implementation details.** We trained our own continuous-time conditional VP-SDE model, and conditional Optimal Transport (conditional OT) flow model from scratch on the above datasets following the hyperparameters and training procedure outlined in Song et al. (2021d) and Lipman et al. (2022). These models are conditioned on class labels, not noisy images. All derivations hold with class label $c$ since $q(\boldsymbol{y}|c, \boldsymbol{x}_1) = q(\boldsymbol{y}|\boldsymbol{x}_1)$ (i.e. the noisy image is independent of class label given the image). We use the open-source implementation of the Euler method provided in torchdiffeq library (Chen, 2018) to solve the ODE in our experiments. Our choice of Euler is intentionally simple, as we focus on flow sampling with the conditional OT path, and not on the choice of ODE solver.

**Metrics.** We follow prior works (Chung et al., 2022a; Kawar et al., 2022) and report Fréchet Inception Distance (FID) (Heusel et al., 2017), Learned Perceptual Image Patch Similarity (LPIPS) (Zhang et al., 2018), peak signal-to-noise ratio (PSNR), and structural similarity index (SSIM). We use open-source implementations of these metrics in the TorchMetrics library (Detlefsen et al., 2022).

**Methods and baselines.** We use our two pretrained model checkpoints— a conditional OT flow model and continuous VP-SDE diffusion model, and perform flow sampling with both conditional OT and Variance-Preserving (VP) paths, labeling our methods as OT-ODE and VP-ODE respectively. Because qualitative results are identical and quantitative results similar, we only include the VP-SDE diffusion model in the main text, and include the conditional OT flow model in Appendix C. We compare our OT-ODE and VP-ODE methods against ΠGDM (Song et al., 2022) and RED-Diff (Mardani et al., 2023) as relevant baselines. We selected these baselines because they achieve state-of-

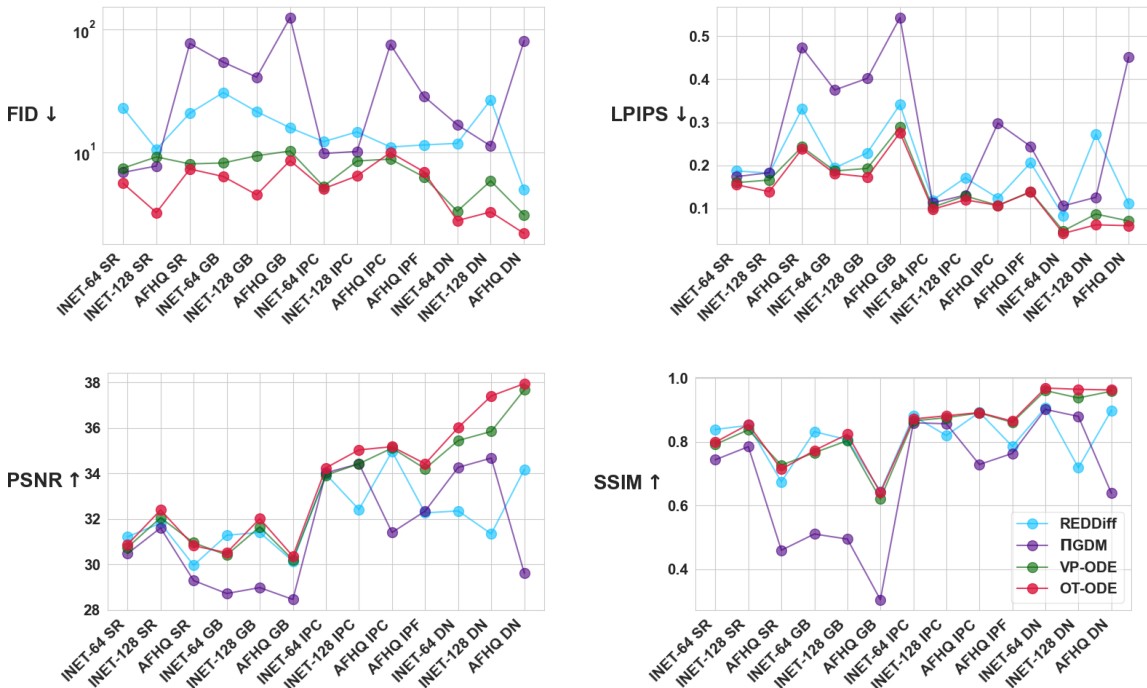

Figure 2: Quantitative evaluation of pretrained VP-SDE model for solving linear inverse problems on super-resolution (SR), gaussian deblurring (GB), inpainting with centered (IPC) and freeform mask (IPF), and denoising (DN) with $\sigma_y = 0.05$. We present results on face-blurred ImageNet-64 (INET-64), face-blurred ImageNet-128 (INET-128), and AFHQ.

the-art performance in solving linear inverse problems using diffusion models. The code for both baseline methods is available on github, and we make minimal changes while reimplementing these methods in our codebase. A fair comparison between methods requires considering the number of function evaluations (NFEs) used during sampling. We utilize at most 100 NFEs for our OT-ODE and VP-ODE sampling (see Appendix C), and utilize 100 for ΠGDM as recommended in Song et al. (2022). We allow RED-Diff 1000 NFEs since it does not require gradients of $\widehat{x_1}$. For OT-ODE following Algorithm 1, we use $\gamma_t = 1$ and initial $t = 0.2$ for all datasets and tasks. For VP-ODE following Algorithm 2 in the Appendix, we use $\gamma_t = \sqrt{\frac{\alpha_t}{\alpha_t^2 + \sigma_t^2}}$ and initial $t = 0.4$ for all datasets and tasks. Ablations of these mildly tuned hyperparameters are shown in Appendix B. We extensively tuned hyperparameters for RED-Diff and ΠGDM as described in Appendix E, including different hyperparameters per dataset and task.

## 4.1 Experimental Results

We report quantitative results for the VP-SDE model, across all datasets and linear measurements, in Figure 2 for $\sigma_y = 0.05$, and in Figure 9 within Appendix C for $\sigma_y = 0$. Additionally, we report results for the conditional OT flow model in Figure 11 and Figure 10 for $\sigma_y = 0.05$ and $\sigma_y = 0$, respectively, in Appendix C. Exact numerical values for all the metrics across all datasets and tasks can also be found in Appendix C. Qualitative images have been selected for demonstration purposes.

**Gaussian deblurring.** We report qualitative noisy results for the VP-SDE model in Figure 3 and for the conditional OT flow (cond-OT) model in Figure 12. We observe that OT-ODE and VP-ODE outperforms ΠGDM and RED-Diff, both qualitatively and quantitatively, across all datasets for $\sigma_y = 0.05$. As shown in these figures, ΠGDM tends to sharpen the images, which sometimes results in unnatural textures in the images. Further, we also observe some unnatural textures and background noise with RED-Diff for $\sigma_y = 0.05$. For $\sigma_y = 0$, OT-ODE has better FID and LPIPS, but ΠGDM shows improved PSNR and SSIM. Figure 21 and Figure 18 show qualitative examples for $\sigma_y = 0$.

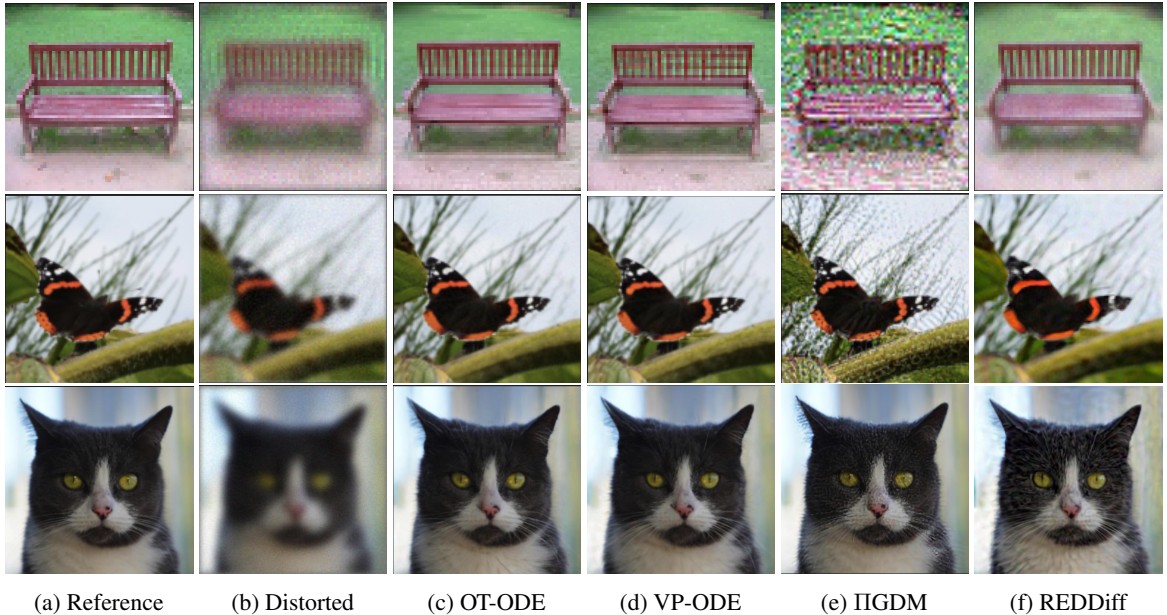

| (a) Reference | (b) Distorted | (c) OT-ODE | (d) VP-ODE | (e) ΠGDM | (f) REDDiff |

Figure 3: Results for Gaussian deblurring with VP-SDE model and $\sigma_y = 0.05$ for (**first row**) ImageNet-64, (**second row**) ImageNet-128, and (**third row**) AFHQ.

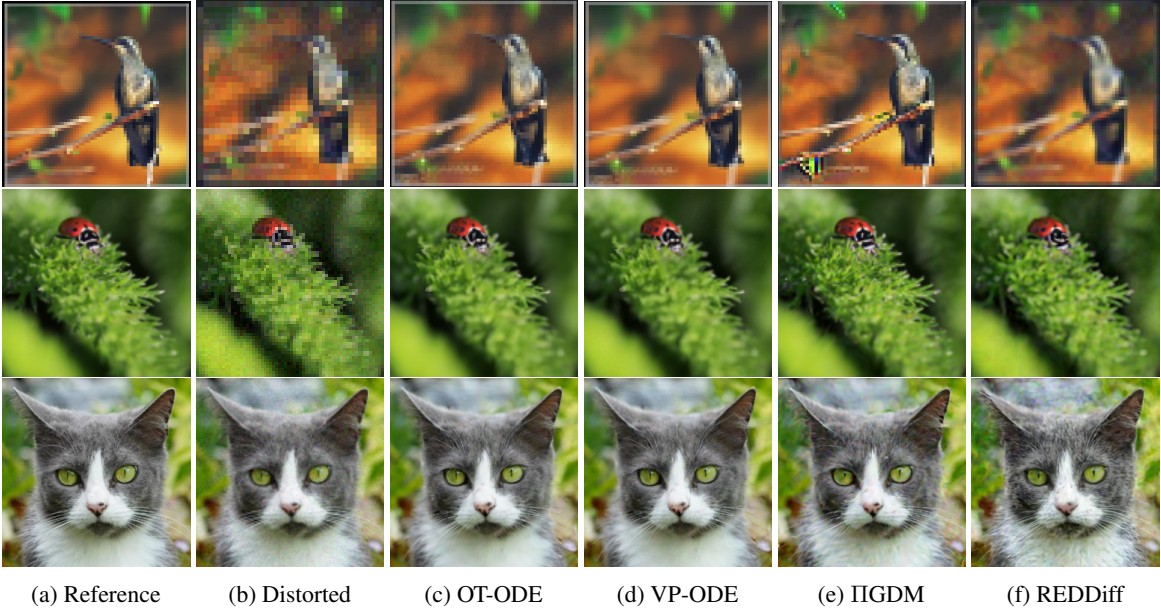

| (a) Reference | (b) Distorted | (c) OT-ODE | (d) VP-ODE | (e) ΠGDM | (f) REDDiff |

Figure 4: Results for super-resolution with VP-SDE model and $\sigma_y = 0.05$ for (**first row**) ImageNet-64 2×, (**second row**) ImageNet-128 2×, and (**third row**) AFHQ 4×.

**Super-resolution.** We report qualitative noisy results for the VP-SDE model in Figure 4 and for the cond-OT model in Figure 13. OT-ODE consistently achieves better FID, LPIPS and PSNR metrics compared to other methods for $\sigma_y = 0.05$ (See Figure 2 and 11). Similar to Gaussian deblurring, ΠGDM tends to produce sharper edges. This is certainly desirable to achieve good super-resolution, but sometimes this results in unnatural textures in the images (See Figure 4). RED-Diff for $\sigma_y = 0.05$ gives slightly blurry images. In our experiments, we observe RED-Diff is sensitive to the values of $\sigma_y$, and we get good quality results for smaller values of $\sigma_y$, but the performance deteriorates with increase in value of $\sigma_y$. For $\sigma_y = 0$, as shown in Figure 19 and Figure 22, all the methods achieve comparable performance and the method declared best varies per metric and dataset.

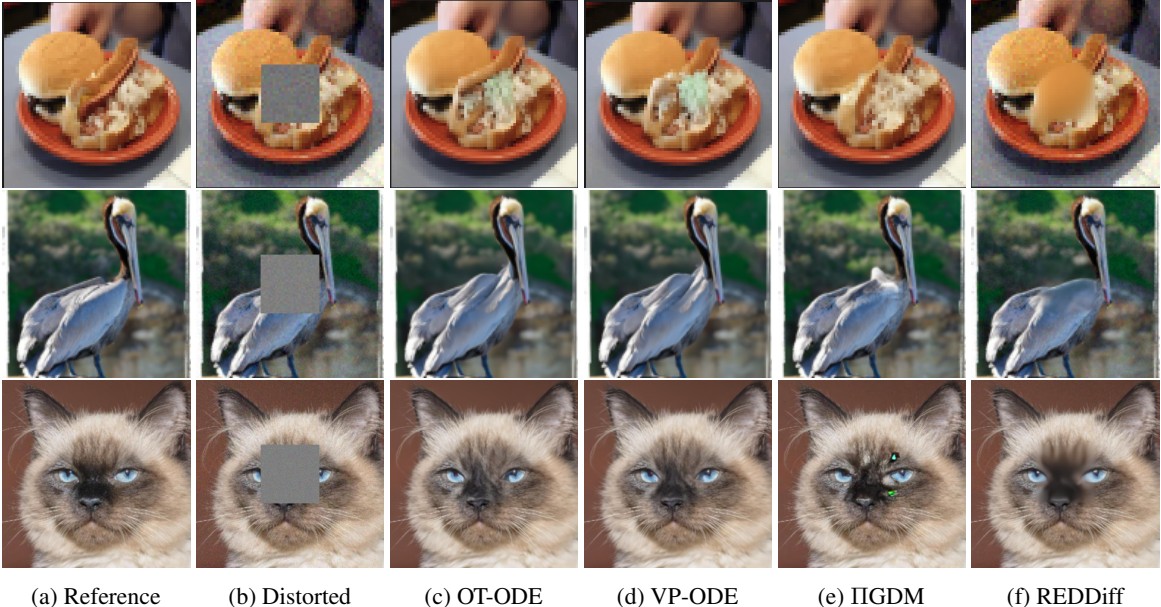

(a) Reference  (b) Distorted  (c) OT-ODE  (d) VP-ODE  (e) ΠGDM  (f) REDDiff

Figure 5: Results for inpainting (centered mask) with VP-SDE model and $\sigma_y = 0.05$ for (**first row**) ImageNet-64, (**second row**) ImageNet-128, and (**third row**) AFHQ.

**Inpainting.** For centered mask inpainting, OT-ODE outperforms ΠGDM and RED-Diff in terms of LPIPS, PSNR and SSIM across all datasets at $\sigma_y = 0.05$. Regarding FID, OT-ODE performs comparably to or better than VP-ODE (See Figure 2 and 11). Similar observations hold true for inpainting with freeform mask on AFHQ. We present qualitative noisy results for the VP-SDE model in Figure 5 and the cond-OT model in Figure 14. As evident in these images, OT-ODE can result in more semantically meaningful inpainting (for instance, the shape of bird's neck, and shape of hot-dog bread in Figure 5). In contrast, the inpainted regions generated by RED-Diff tend to be blurry and less semantically meaningful. However, we note that OT-ODE (and VP-ODE) inpainting occasionally produces artifacts in the inpainted region as the resolution of image increases. We show examples of such negative inpainting results in Appendix C.2. Empirically, we observe that performance of RED-Diff and ΠGDM improves as $\sigma_y$ decreases. For $\sigma_y = 0$, RED-Diff achieves higher PSNR and SSIM, but performs worse than OT-ODE in terms of FID and LPIPS (Refer to Figure 9). OT-ODE's tendency to produce inpainting artifacts for higher resolution images remains for $\sigma_y = 0$, and can occur for the same images as $\sigma_y = 0.05$. These artifacts can significantly degrade the pixel-based metrics PSNR and SSIM more than the perceptual metrics such as FID and LPIPS. We further note that noiseless inpainting for OT-ODE can be improved by incorporating null-space decomposition (Wang et al., 2022). We describe this adjustment in Appendix D.

## 5 Related Work

The challenge of solving noisy linear inverse problems without any training has been tackled in many ways, often with other solution concepts than posterior sampling (Elad et al., 2023). Utilizing a diffusion model has a host of recent research that we build upon. Our state-of-the-art baselines ΠGDM (Song et al., 2022) and RED-Diff (Mardani et al., 2023) correspond to lines of research in gradient-based adaptations and variational inference.

Earlier gradient-based adaptations that approximate $\nabla_{\boldsymbol{x}_t} \ln q(\boldsymbol{y}|\boldsymbol{x}_t)$ in various ways include Diffusion Posterior Sampling (DPS) (Chung et al., 2022a), Manifold Constrained Gradient (Chung et al., 2022b), and an annealed approximation (Jalal et al., 2021). ΠGDM out-performs earlier methods combining adaptive weights and Gaussian posterior approximation with discrete-time denoising diffusion implicit model (DDIM) sampling (Song et al., 2021a). Here we adapt ΠGDM to all Gaussian probability paths and to flow sampling. Our results show adaptive weights are unnecessary for strongly performing conditional OT flow sampling. Denoising Diffusion Null Models (DDNM) (Wang et al., 2022) proposed an alternative approximation of $\mathbb{E}_q[\boldsymbol{x}_1|\boldsymbol{x}_t, \boldsymbol{y}]$ using a null-space decomposition specific to linear inverse problems, which has been explored in combination with our method in Appendix D.

RED-Diff (Mardani et al., 2023) approximates intractable $q(\boldsymbol{x}_1|\boldsymbol{y})$ directly using variational inference, solving for parameters via optimization. RED-Diff was reported to have mode-seeking behavior confirmed by our results where RED-Diff performed better for noiseless inference. Another earlier variational inference method is Denoising Diffusion Restoration Models (DDRM) (Kawar et al., 2022). DDRM showed SVD can be memory-efficient for image applications, and we adapt their SVD implementations for super-resolution and blur. DDRM incorporates noiseless method ILVR (Choi et al., 2021), and leverages a measurement-dependent forward process (i.e. $q(\boldsymbol{x}_t|\boldsymbol{x}_1, \boldsymbol{y}) \neq q(\boldsymbol{x}_t|\boldsymbol{x}_1)$) like earlier SNIPS (Kawar et al., 2021). SNIPS collapses in special cases to variants proposed in Song & Ermon (2019); Song et al. (2021d); Kadkhodaie & Simoncelli (2020) for linear inverse problems. Additional related work can be found in Appendix G.

## 6 Discussion, Limitations, and Future Work

We have presented a training-free approach to solve linear inverse problems using flows that can leverage either pretrained diffusion or flow models. The algorithm is simple, stable, and does not require any problem-specific hyperparameter tuning when used with conditional OT probability paths. Our method combines past ideas from diffusion including ΠGDM and early starting with the conditional OT probability path from flows, and our results demonstrate that this combination can solve inverse problems for both noisy and noiseless cases across a variety of datasets. Our algorithm using the conditional OT path (OT-ODE) produced results superior to the VP path (VP-ODE) and also to ΠGDM and REDDiff for noisy inverse problems. For the noiseless case, the perceptual quality from OT-ODE is on par with ΠGDM for super-resolution and gaussian deblurring, but lagging for inpainting due to image artifacts.

Another important limitation, shared with most of the past related research, is a restriction to linear observations with scalar variance. Our method can extend to arbitrary covariance, but non-linear observations are more complex. Non-linear observations occur with image inverse tasks when utilizing latent, not pixel-space, diffusion or flow models. Applying our approach to such measurements requires devising an alternative $q^{app}(\boldsymbol{y}|\boldsymbol{x}_t)$. Another shared limitation is that we consider the non-blind setting with known $\boldsymbol{A}$ and $\sigma_y$.

Future research could tackle these limitations. For non-linear observations in latent space, we could perhaps build upon Rout et al. (2023) that uses a latent diffusion model for linear inverses. For the blind setting, we might start from blind extensions to DPS and DDRM (Chung et al., 2023a; Murata et al., 2023). As demonstrated here, we may be able to adapt and possibly improve these approaches via conversion to flow sampling using conditional OT paths.

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

# A    Our method for any Gaussian probability path

Algorithm 1 in the main text is specific to conditional OT probability paths. Here we provide Algorithm 2 for any Gaussian probability path specified by Eq. 4. Algorithm 1 and Algorithm 2 are written assuming a denoiser $\widehat{x_1}(x_t)$ is provided from a pretrained diffusion model. For completeness, we also include equivalent Algorithm 3 that assumes $\widehat{v}(x_t)$ is provided from a pretrained flow model. In all cases, the vector field or denoiser is evaluated only once per iteration.

Our VP-ODE sampling results correspond to $\alpha_t$ and $\sigma_t$ given from the Variance-Preserving path, which can be found in (Lipman et al., 2022).

---

**Algorithm 2** A training-free approach to solve inverse problems via flows with a pretrained denoiser

---

**Require:** Pretrained denoiser $\widehat{x_1}(x_t)$ converted to Gaussian probability path with $\alpha_t$ and $\sigma_t$, noisy measurement $y$,
measurement matrix $A$, initial time $t_0$, adaptive weights $\gamma_t$, and std $\sigma_y$

1: Initialize $x_{t_0} = \alpha_{t_0} y + \sigma_{t_0} \epsilon$, where $\epsilon \sim \mathcal{N}(0, I)$        ▷ Initialize $x_t$, Eq. (18)

2: $x_t = x_{t_0}$

3: **for** each time step $t$ of ODE integration **do**       ▷ Integrate ODE from $t = t_0$ to 1.

4:     $r_t^2 = \frac{\sigma_t^2}{\sigma_t^2 + \alpha_t^2}$       ▷ Value of $r_t^2$ from Eq. (16)

5:     $\widehat{v} = \left( \alpha_t \frac{d \ln(\alpha_t/\sigma_t)}{dt} \right) \widehat{x_1} + \frac{d \ln \sigma_t}{dt} x_t$       ▷ Convert $\widehat{x_1}$ to vector field, Eq. (7)

6:     $g = (y - A\widehat{x_1})^\top (r_t^2 AA^\top + \sigma_y^2 I)^{-1} A \frac{\partial \widehat{x_1}}{\partial x_t}$       ▷ $\nabla_{x_t} \ln q^{app}(y|x_t)$

7:     $\widehat{v}_{\text{adapted}} = \widehat{v} + \sigma_t^2 \frac{d \ln(\alpha_t/\sigma_t)}{dt} \gamma_t g$       ▷ Adapt unconditional vector field $\widehat{v}$, Eq. (11)

8:     $x_{t+\Delta t} = \text{ODESolverStep}(x_t, \widehat{v}_{\text{adapted}})$       ▷ One step of ODE solver to update $x_t$

9: **end for**

10: **return** $x_1$       ▷ This is the solution of ODE integration.

---

**Algorithm 3** A training-free approach to solve inverse problems via flows with a pretrained vector field

---

**Require:** Pretrained vector field $\widehat{v}(x_t)$ converted to Gaussian probability path with $\alpha_t$ and $\sigma_t$, noisy measurement $y$,
measurement matrix $A$, initial time $t_0$, adaptive weights $\gamma_t$, and std $\sigma_y$

1: Initialize $x_{t_0} = \alpha_{t_0} y + \sigma_{t_0} \epsilon$, where $\epsilon \sim \mathcal{N}(0, I)$       ▷ Initialize $x_t$, Eq. (18)

2: $x_t = x_{t_0}$

3: **for** each time step $t$ of ODE integration **do**       ▷ Integrate ODE from $t = t_0$ to 1.

4:     $\widehat{v} = \widehat{v}(x_t)$       ▷ $x_t$ is value of $x_t$ at time $t$ during ODE integration

5:     $r_t^2 = \frac{\sigma_t^2}{\sigma_t^2 + \alpha_t^2}$       ▷ Value of $r_t^2$ from Eq. (16)

6:     $\widehat{x_1} = \left( \alpha_t \frac{d \ln(\alpha_t/\sigma_t)}{dt} \right)^{-1} \left( \widehat{v} - \frac{d \ln \sigma_t}{dt} x_t \right)$       ▷ Convert vector field to $\widehat{x_1}$, Eq. (7)

7:     $g = (y - A\widehat{x_1})^\top (r_t^2 AA^\top + \sigma_y^2 I)^{-1} A \frac{\partial \widehat{x_1}}{\partial x_t}$       ▷ $\nabla_{x_t} \ln q^{app}(y|x_t)$

8:     $\widehat{v}_{\text{adapted}} = \widehat{v} + \sigma_t^2 \frac{d \ln(\alpha_t/\sigma_t)}{dt} \gamma_t g$       ▷ Adapt unconditional vector field $\widehat{v}$, Eq. (11)

9:     $x_{t+\Delta t} = \text{ODESolverStep}(x_t, \widehat{v}_{\text{adapted}})$       ▷ One step of ODE solver to update $x_t$

10: **end for**

11: **return** $x_1$       ▷ This is the solution of ODE integration.

---

# B  Ablation Study

**Choice of initialization.**    We initialize the flow at time $t > 0$ as $\boldsymbol{x}_t = \alpha_t \boldsymbol{y} + \sigma_t \epsilon$ (y-init) where $\epsilon \sim \mathcal{N}(0, \boldsymbol{I})$. Another choice of initialization is to use $\boldsymbol{x}_t = \alpha_t \boldsymbol{A}^\dagger \boldsymbol{y} + \sigma_t \epsilon$. However, empirically we find that this initialization performs worse that y-init on cond-OT model with OT-ODE sampling. We summarize the results of our ablation study in Table 1. We find that on Gaussian deblurring, initialization with $\boldsymbol{A}^\dagger \boldsymbol{y}$ does worse than y-init, while the performance of both the initializations is comparable for super-resolution. In all our experiments, we use y-init, due to its better performance on Gaussian deblurring.

Table 1: Quantitative evaluation of choice of initialization for conditional OT flow model with OT-ODE sampling on AFHQ dataset. We find that y-init outperforms $\boldsymbol{A}^\dagger \boldsymbol{y}$ on Gaussian deblurring.

| Initialization | Start time | NFEs ↓ | Gaussian deblur, $\sigma_y = 0.05$ | | | | SR 4×, $\sigma_y = 0.05$ | | | |
|---|---|---|---|---|---|---|---|---|---|---|
| | | | FID ↓ | LPIPS ↓ | PSNR ↑ | SSIM ↑ | FID ↓ | LPIPS ↓ | PSNR ↑ | SSIM ↑ |
| y init | 0.2 | 100 | **7.57** | **0.268** | **30.28** | **0.626** | **6.03** | **0.219** | **31.12** | **0.739** |
| $\boldsymbol{A}^\dagger \boldsymbol{y}$ | 0.1 | 100 | 41.22 | 0.449 | 28.79 | 0.392 | 12.93 | 0.292 | 30.46 | 0.664 |
| $\boldsymbol{A}^\dagger \boldsymbol{y}$ | 0.2 | 100 | 56.42 | 0.554 | 28.11 | 0.249 | 6.09 | 0.219 | 31.12 | 0.739 |

**Ablation over $\gamma_t$ for VP-ODE sampling.**    We compare the performance of $\gamma_t = 1$ against $\gamma_t = \sqrt{\frac{\alpha_t}{\alpha_t^2 + \sigma_t^2}}$. We show results of VP-ODE sampling with VP-SDE model in Table 2 and Table 3. As seen our choice of $\gamma_t$ outperform $\gamma_t = 1$ across all the metrics on face-blurred ImageNet-128.

Table 2: Quantitative evaluation of value of $\gamma_t$ in VP-ODE sampling with VP-SDE model on face-blurred ImageNet-128 dataset.

| $\gamma_t$ | Start time | NFEs ↓ | SR 2×, $\sigma_y = 0.05$ | | | | Gaussian deblur, $\sigma_y = 0.05$ | | | |
|---|---|---|---|---|---|---|---|---|---|---|
| | | | FID ↓ | LPIPS ↓ | PSNR ↑ | SSIM ↑ | FID ↓ | LPIPS ↓ | PSNR ↑ | SSIM ↑ |
| 1 | 0.4 | 60 | 32.66 | 0.371 | 29.06 | 0.530 | 29.31 | 0.346 | 29.12 | 0.554 |
| $\sqrt{\frac{\alpha_t}{\alpha_t^2 + \sigma_t^2}}$ | 0.4 | 60 | **9.14** | **0.167** | **32.06** | **0.838** | **10.14** | **0.196** | **31.59** | **0.800** |

Table 3: Quantitative evaluation of value of $\gamma_t$ in VP-ODE sampling with VP-SDE model on face-blurred ImageNet-128 dataset.

| $\gamma_t$ | Start time | NFEs ↓ | Inpainting-*Center*, $\sigma_y = 0.05$ | | | | Denoising, $\sigma_y = 0.05$ | | | |
|---|---|---|---|---|---|---|---|---|---|---|
| | | | FID ↓ | LPIPS ↓ | PSNR ↑ | SSIM ↑ | FID ↓ | LPIPS ↓ | PSNR ↑ | SSIM ↑ |
| 1 | 0.3 | 70 | 53.03 | 0.285 | 31.55 | 0.737 | 28.37 | 0.238 | 31.63 | 0.786 |
| $\sqrt{\frac{\alpha_t}{\alpha_t^2 + \sigma_t^2}}$ | 0.3 | 70 | **8.47** | **0.129** | **34.43** | **0.876** | **5.83** | **0.087** | **35.85** | **0.938** |

**Variation of performance with NFEs.**    We analyze the variation in performance of OT-ODE, VP-ODE and ΠGDM for solving linear inverse problems as NFEs are varied. The results have been summarized in Figure 6. We observe that OT-ODE consistently outperforms VP-ODE and ΠGDM across all measurements in terms of FID and LPIPS metrics, even for NFEs as small as 20. We also note that the choice of starting time matters to achieve good performance with OT-ODE. For instance, starting at $t = 0.4$ outperforms $t = 0.2$ when NFEs are small, but eventually as NFEs is increased, $t = 0.2$ performs better. We also note that ΠGDM achieves higher values of PSNR and SSIM at smaller NFEs for super-resolution but has inferior FID and LPIPS compared to OT-ODE.

**Choice of starting time.**    We plot the variation in performance of OT-ODE and VP-ODE sampling with change in start times for conditional OT model and VP-SDE model on AFHQ dataset in Figure 7 and Figure 8, respectively.

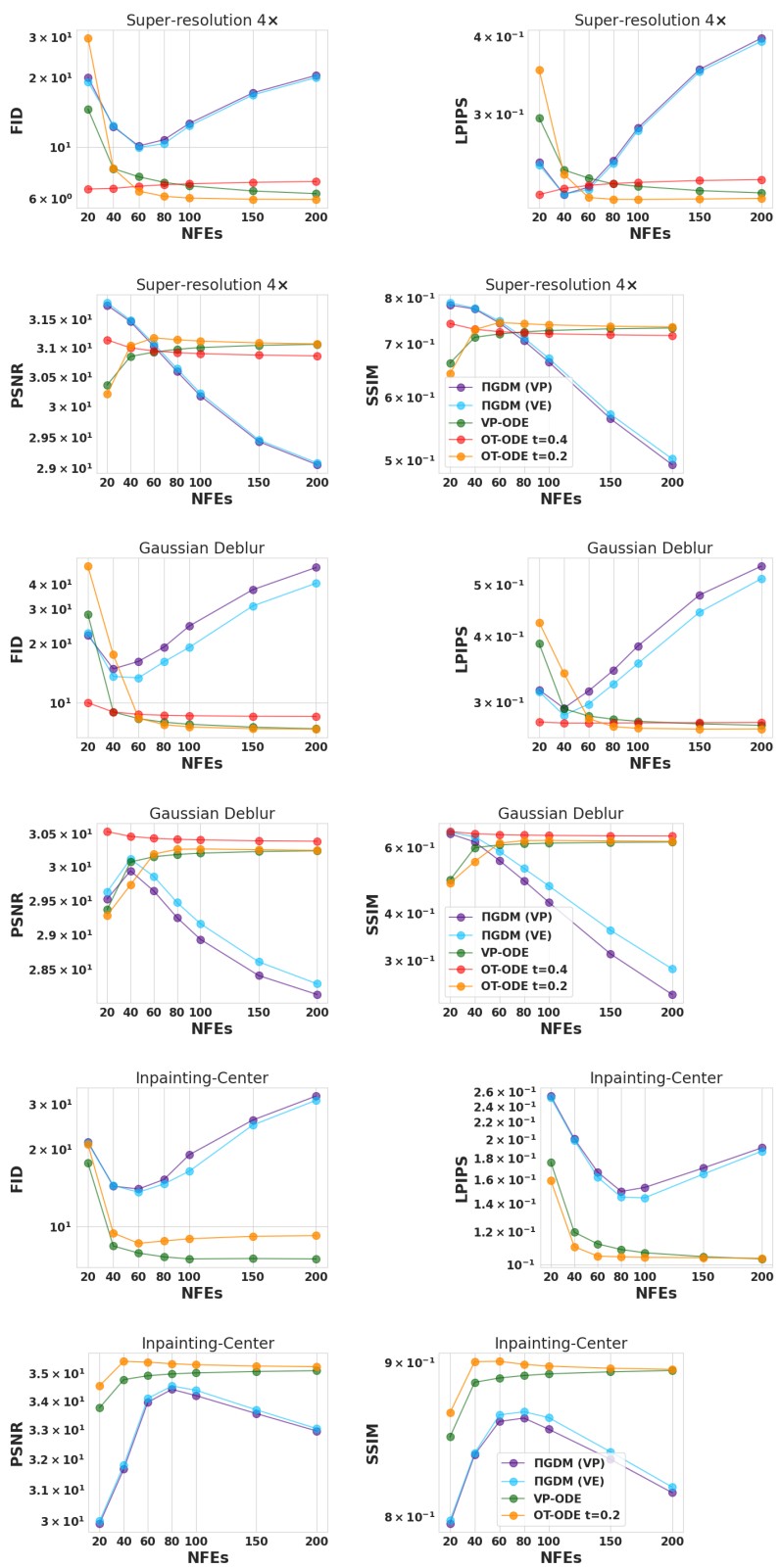

Figure 6: Performance of different procedures for solving linear inverse problems with variation in NFEs on AFHQ dataset. We use pretrained conditional OT model and set $\sigma_y = 0.05$. The legends VP and VE indicate the choice of $r_t^2$ used in ΠGDM (See Appendix E.1). Time $t = 0.2$ and $0.4$ indicates the starting time of sampling with OT-ODE.

We note that in general, OT-ODE sampling achieves optimal performance across all measurements and all metrics at $t = 0.2$ while VP-ODE sampling achieves optimal performance between start times of $t = 0.3$ and $0.4$. In this work, for all the experiments, we use $t = 0.2$ for OT-ODE sampling and $t = 0.4$ for VP-ODE sampling.

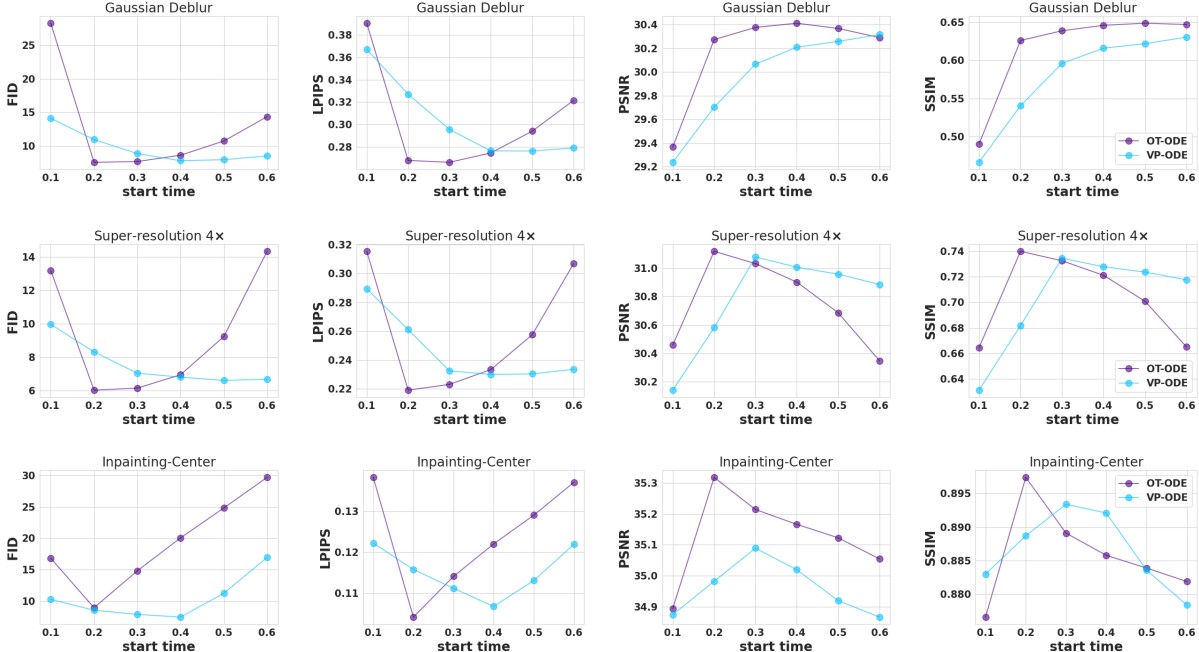

Figure 7: Performance of OT-ODE and VP-ODE in solving linear inverse problems with varying start times on AFHQ dataset. We use pretrained cond-OT model and set $\sigma_y = 0.05$.

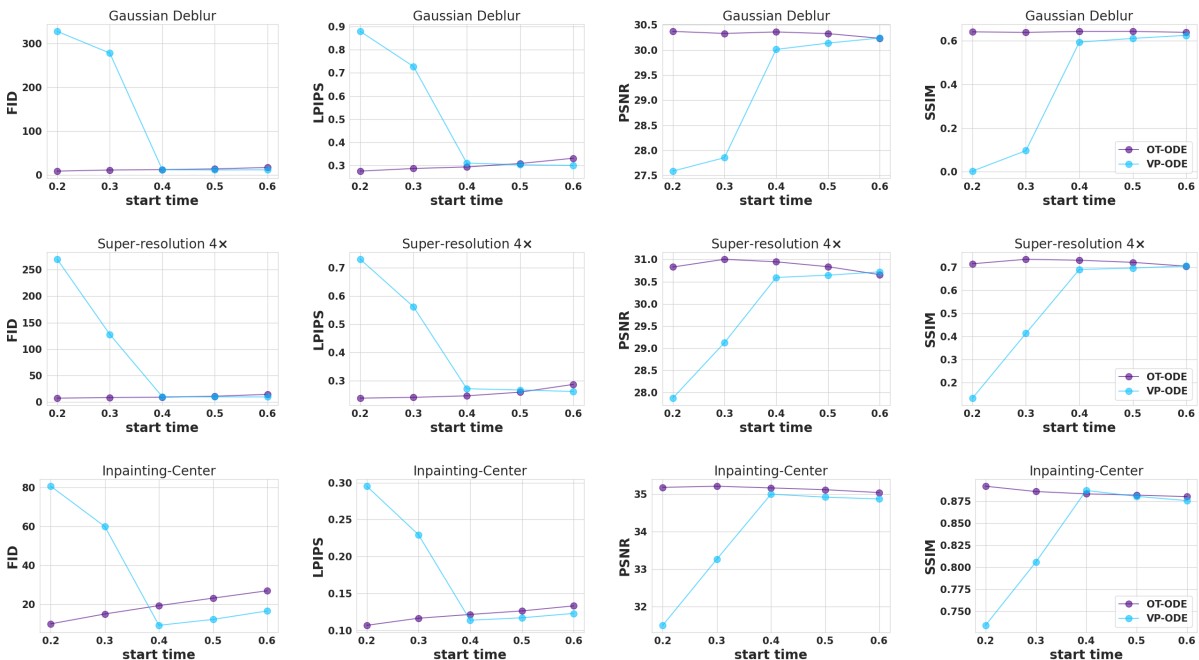

Figure 8: Performance of OT-ODE and VP-ODE in solving linear inverse problems with varying start times on AFHQ dataset. We use pretrained VP-SDE model and set $\sigma_y = 0.05$.

## C    Additional Empirical Results

The main text includes Figure 2 with $\sigma_y = 0.05$ produced with the denoiser from the continuous-time VP-SDE diffusion model showing plots of various metrics across all datasets and tasks. Here we provide the same for our pre-trained conditional OT flow matching model using Algorithm 3 in Figure 11, and noiseless figures for both models in Figure 9 and 10. To save compute, for the flow model we only include our ΠGDM baseline as RED-Diff required extensive hyperparameter tuning. The qualitative results using the flow model instead of diffusion model checkpoint are identical.

This section also includes tables containing the numerical values of metrics across all datasets and tasks. The tables are hierarchically organized by noise, dataset, and task in consistent ordering. Noisy results with $\sigma_y = 0.05$ are in Table 4 to 9 and noiseless results with $\sigma_y = 0$ are in Table 8 to 13.

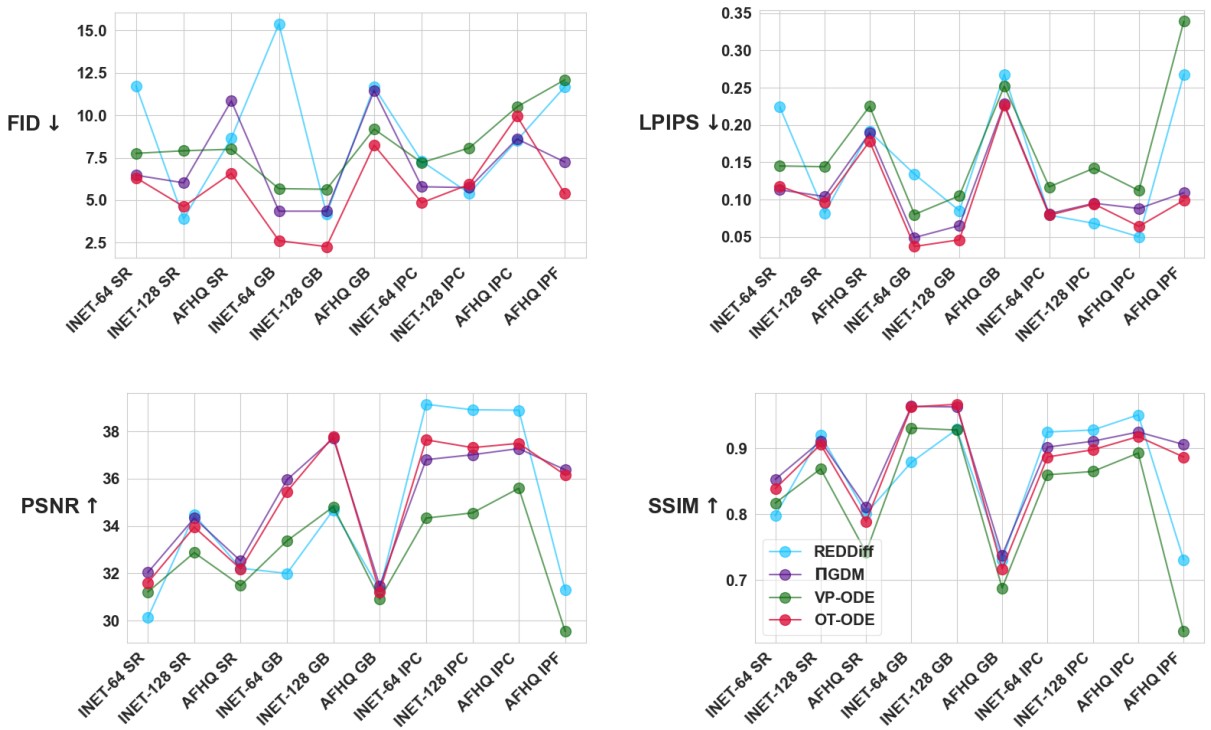

Figure 9: Quantitative evaluation of pretrained VP-SDE model for linear inverse problems on super-resolution (SR), gaussian deblurring (GB), image inpainting - centered mask (IPC) and inpainting - free-form (IPF) with $\sigma_y = 0$. We show results on face-blurred ImageNet-64 (INET-64), face-blurred ImageNet-128 (INET-128), and AFHQ-256 (AFHQ).

Table 4: Quantitative evaluation of linear inverse problems on face-blurred ImageNet-$64 \times 64$

| Model | Inference | NFEs ↓ | SR 2×, $\sigma_y = 0.05$ | | | | Gaussian deblur, $\sigma_y = 0.05$ | | | |
|---|---|---|---|---|---|---|---|---|---|---|
| | | | FID ↓ | LPIPS ↓ | PSNR ↑ | SSIM ↑ | FID ↓ | LPIPS ↓ | PSNR ↑ | SSIM ↑ |
| OT | OT-ODE | 80 | **6.07** | **0.157** | **30.88** | **0.799** | **6.83** | **0.185** | **30.51** | **0.773** |
| OT | VP-ODE | 80 | 7.82 | 0.163 | 30.75 | 0.792 | 8.72 | 0.190 | 30.40 | 0.765 |
| OT | ΠGDM | 100 | 6.52 | 0.168 | 30.54 | 0.753 | 55.19 | 0.374 | 28.74 | 0.516 |
| VP-SDE | OT-ODE | 80 | **5.57** | **0.155** | 30.88 | 0.799 | **6.33** | **0.181** | **30.52** | 0.773 |
| VP-SDE | VP-ODE | 80 | 7.40 | 0.160 | 30.75 | 0.792 | 8.16 | 0.187 | 30.42 | 0.766 |
| VP-SDE | ΠGDM | 100 | 6.84 | 0.174 | 30.48 | 0.743 | 54.77 | 0.376 | 28.74 | 0.511 |
| VP-SDE | RED-Diff | 1000 | 23.02 | 0.187 | **31.22** | **0.839** | 51.20 | 0.236 | 30.19 | **0.776** |

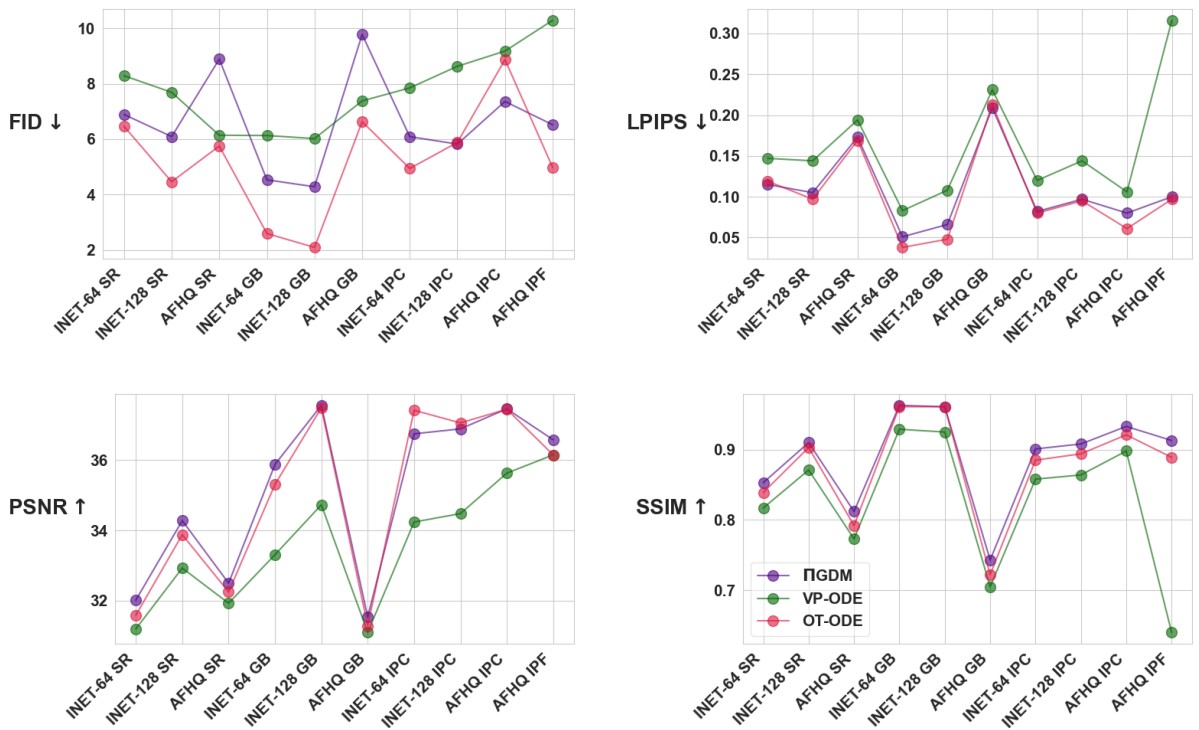

Figure 10: Quantitative evaluation of pretrained conditional OT model for linear inverse problems on super-resolution (SR), gaussian deblurring (GB), image inpainting - centered mask (IPC) and inpainting - freeform (IPF) with $\sigma_y = 0$. We show results on face-blurred ImageNet-64 (INET-64), face-blurred ImageNet-128 (INET-128), and AFHQ-256 (AFHQ).

Table 5: Quantitative evaluation of linear inverse problems on face-blurred ImageNet-$64 \times 64$

| Model | Inference | NFEs ↓ | Inpainting-*Center*, $\sigma_y = 0.05$ | | | | Denoising, $\sigma_y = 0.05$ | | | |
|---|---|---|---|---|---|---|---|---|---|---|
| | | | FID ↓ | LPIPS ↓ | SSIM ↑ | PSNR ↑ | FID ↓ | LPIPS ↓ | SSIM ↑ | PSNR ↑ |
| OT | OT-ODE | 80 | **5.45** | **0.101** | **34.21** | **0.870** | 2.91 | 0.044 | 35.96 | 0.968 |
| OT | VP-ODE | 80 | 5.70 | 0.105 | 33.87 | 0.865 | 3.54 | 0.049 | 35.37 | 0.960 |
| OT | ΠGDM | 100 | 9.25 | 0.111 | 34.13 | 0.863 | 16.59 | 0.102 | 34.60 | 0.906 |
| VP-SDE | OT-ODE | 80 | **5.03** | **0.098** | **34.25** | **0.872** | 2.76 | 0.042 | 36.02 | 0.969 |
| VP-SDE | VP-ODE | 80 | 5.26 | 0.103 | 33.93 | 0.866 | 3.29 | 0.048 | 35.45 | 0.961 |
| VP-SDE | ΠGDM | 100 | 9.75 | 0.113 | 34.03 | 0.860 | 17.19 | 0.107 | 34.25 | 0.901 |
| VP-SDE | RED-Diff | 1000 | 12.18 | 0.119 | 33.97 | 0.881 | 6.02 | 0.041 | 35.64 | 0.964 |

Table 6: Quantitative evaluation of linear inverse problems on face-blurred ImageNet-$128 \times 128$

| Model | Inference | NFEs ↓ | SR 2×, $\sigma_y = 0.05$ | | | | Gaussian deblur, $\sigma_y = 0.05$ | | | |
|---|---|---|---|---|---|---|---|---|---|---|
| | | | FID ↓ | LPIPS ↓ | PSNR ↑ | SSIM ↑ | FID ↓ | LPIPS ↓ | PSNR ↑ | SSIM ↑ |
| OT | OT-ODE | 70 | **3.22** | **0.141** | **32.35** | 0.820 | **4.84** | **0.175** | **31.94** | **0.821** |
| OT | VP-ODE | 70 | 7.52 | 0.162 | 32.24 | **0.847** | 8.49 | 0.191 | 31.76 | 0.809 |
| OT | ΠGDM | 100 | 4.38 | 0.148 | 32.07 | 0.831 | 30.30 | 0.328 | 29.96 | 0.606 |
| VP-SDE | OT-ODE | 70 | **3.21** | **0.139** | **32.40** | **0.855** | **4.49** | **0.173** | **32.02** | **0.824** |
| VP-SDE | VP-ODE | 70 | 9.14 | 0.166 | 32.06 | 0.838 | 9.35 | 0.193 | 31.66 | 0.804 |
| VP-SDE | ΠGDM | 100 | 7.55 | 0.183 | 31.61 | 0.785 | 55.61 | 0.463 | 28.57 | 0.414 |
| VP-SDE | RED-Diff | 1000 | 10.54 | 0.182 | 31.82 | 0.852 | 21.43 | 0.229 | 31.41 | 0.807 |

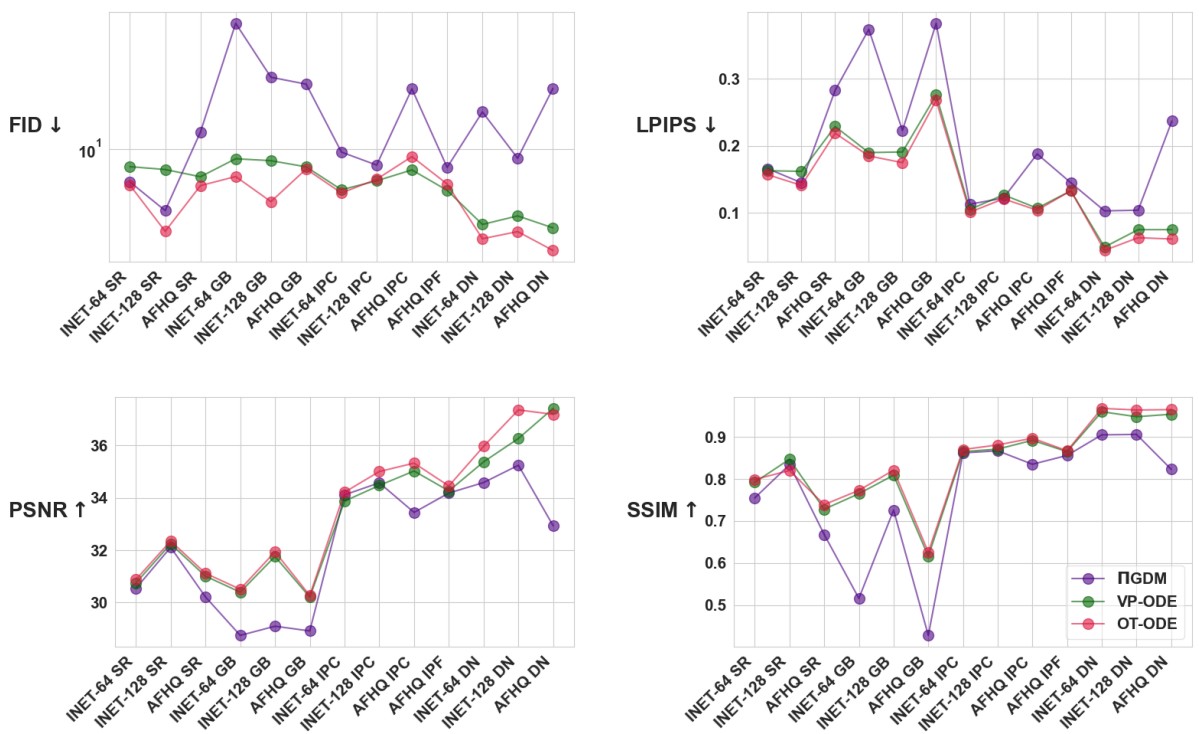

Figure 11: Quantitative evaluation of pretrained conditional OT model for linear inverse problems on super-resolution (SR), gaussian deblurring (GB), image inpainting - centered mask (IPC) and denoising (DN) with $\sigma_y = 0.05$. We show results on face-blurred ImageNet-64 (INET-64), face-blurred ImageNet-128 (INET-128), and AFHQ-256 (AFHQ).

Table 7: Quantitative evaluation of linear inverse problems on face-blurred ImageNet-128 $\times$ 128

| Model | Inference | NFEs ↓ | Inpainting-*Center*, $\sigma_y = 0.05$ | | | | Denoising, $\sigma_y = 0.05$ | | | |
|---|---|---|---|---|---|---|---|---|---|---|
| | | | FID ↓ | LPIPS ↓ | PSNR ↑ | SSIM ↑ | FID ↓ | LPIPS ↓ | PSNR ↑ | SSIM ↑ |
| OT | OT-ODE | 70 | 6.58 | **0.121** | **35.00** | **0.881** | 3.21 | **0.063** | **37.35** | **0.964** |
| OT | VP-ODE | 70 | **6.44** | 0.127 | 34.47 | 0.871 | 3.98 | 0.075 | 36.26 | 0.948 |
| OT | ΠGDM | 100 | 7.99 | 0.122 | 34.57 | 0.867 | 9.60 | 0.107 | 35.11 | 0.903 |
| VP-SDE | OT-ODE | 70 | **6.39** | **0.120** | **35.04** | **0.882** | 3.25 | **0.062** | **37.41** | **0.965** |
| VP-SDE | VP-ODE | 70 | 8.47 | 0.129 | 34.43 | 0.876 | 5.83 | 0.087 | 35.85 | 0.938 |
| VP-SDE | ΠGDM | 100 | 9.75 | 0.130 | 34.45 | 0.858 | 10.69 | 0.124 | 34.72 | 0.882 |
| VP-SDE | RED-Diff | 1000 | 14.63 | 0.171 | 32.42 | 0.820 | 9.19 | 0.105 | 33.52 | 0.895 |

Table 8: Quantitative evaluation of linear inverse problems on AFHQ-256 $\times$ 256

| Model | Inference | NFEs ↓ | SR 4×, $\sigma_y = 0.05$ | | | | Gaussian deblur, $\sigma_y = 0.05$ | | | |
|---|---|---|---|---|---|---|---|---|---|---|
| | | | FID ↓ | LPIPS ↓ | PSNR ↑ | SSIM ↑ | FID ↓ | LPIPS ↓ | PSNR ↑ | SSIM ↑ |
| OT | OT-ODE | 100 | **6.03** | **0.219** | **31.12** | **0.739** | 7.57 | **0.268** | **30.27** | **0.626** |
| OT | VP-ODE | 100 | 6.81 | 0.229 | 31.01 | 0.728 | 7.80 | 0.276 | 30.21 | 0.616 |
| OT | ΠGDM | 100 | 12.69 | 0.285 | 30.18 | 0.665 | 24.60 | 0.383 | 28.93 | 0.429 |
| VP-SDE | OT-ODE | 100 | **7.28** | **0.238** | 30.83 | 0.714 | **8.53** | **0.276** | **30.37** | 0.641 |
| VP-SDE | VP-ODE | 100 | 8.02 | 0.243 | **30.96** | **0.727** | 10.21 | 0.289 | 30.21 | 0.621 |
| VP-SDE | ΠGDM | 100 | 77.49 | 0.469 | 29.34 | 0.469 | 116.42 | 0.535 | 28.49 | 0.313 |
| VP-SDE | RED-Diff | 1000 | 20.84 | 0.331 | 29.97 | 0.675 | 15.81 | 0.341 | 30.15 | **0.645** |

Table 9: Quantitative evaluation of linear inverse problems on AFHQ-$256 \times 256$

| Model | Inference | NFEs ↓ | Inpainting-*Center*, $\sigma_y = 0.05$ | | | | Denoising, $\sigma_y = 0.05$ | | | |
|---|---|---|---|---|---|---|---|---|---|---|
| | | | FID ↓ | LPIPS ↓ | PSNR ↑ | SSIM ↑ | FID ↓ | LPIPS ↓ | PSNR ↑ | SSIM ↑ |
| OT | OT-ODE | 100 | 8.98 | **0.104** | **35.32** | **0.897** | **2.48** | **0.061** | 37.18 | **0.965** |
| OT | VP-ODE | 100 | **7.48** | 0.107 | 35.02 | 0.892 | 3.38 | 0.075 | **37.41** | 0.954 |
| OT | ΠGDM | 100 | 19.09 | 0.153 | 34.20 | 0.855 | 22.87 | 0.237 | 32.93 | 0.823 |
| VP-SDE | OT-ODE | 100 | 9.93 | **0.107** | **35.18** | **0.892** | 2.17 | **0.060** | **37.95** | **0.963** |
| VP-SDE | VP-ODE | 100 | **8.78** | **0.107** | 35.12 | 0.891 | 3.08 | 0.071 | 37.68 | 0.959 |
| VP-SDE | ΠGDM | 100 | 57.46 | 0.239 | 32.40 | 0.773 | 81.15 | 0.451 | 29.62 | 0.639 |
| VP-SDE | RED-Diff | 1000 | 11.02 | 0.124 | 34.97 | 0.893 | 4.93 | 0.112 | 34.18 | 0.899 |

Table 10: Quantitative evaluation of linear inverse problems on face-blurred ImageNet-$64 \times 64$

| Model | Inference | NFEs ↓ | SR 2×, $\sigma_y = 0$ | | | | Gaussian deblur, $\sigma_y = 0$ | | | |
|---|---|---|---|---|---|---|---|---|---|---|
| | | | FID ↓ | LPIPS ↓ | PSNR ↑ | SSIM ↑ | FID ↓ | LPIPS ↓ | PSNR ↑ | SSIM ↑ |
| OT | OT-ODE | 80 | **6.46** | 0.119 | 31.59 | 0.839 | **2.59** | **0.038** | 35.31 | 0.961 |
| OT | VP-ODE | 80 | 8.29 | 0.147 | 31.20 | 0.817 | 6.13 | 0.083 | 33.31 | 0.929 |
| OT | ΠGDM | 100 | 6.89 | **0.115** | **32.02** | **0.853** | 4.53 | 0.051 | **35.88** | **0.963** |
| VP-SDE | OT-ODE | 80 | **6.32** | 0.118 | 31.60 | 0.839 | **2.61** | **0.037** | 35.45 | 0.963 |
| VP-SDE | VP-ODE | 80 | 7.76 | 0.145 | 31.21 | 0.817 | 5.68 | 0.080 | 33.37 | 0.931 |
| VP-SDE | ΠGDM | 100 | 6.47 | **0.113** | **32.03** | **0.853** | 4.35 | 0.049 | **35.95** | **0.964** |
| VP-SDE | RED-Diff | 1000 | 11.74 | 0.224 | 30.12 | 0.798 | 15.39 | 0.134 | 31.99 | 0.879 |

Table 11: Quantitative evaluation of linear inverse problems on face-blurred ImageNet-$64 \times 64$

| Model | Inference | NFEs ↓ | Inpainting-*Center*, $\sigma_y = 0$ | | | |
|---|---|---|---|---|---|---|
| | | | FID ↓ | LPIPS ↓ | PSNR ↑ | SSIM ↑ |
| OT | OT-ODE | 80 | **4.94** | **0.080** | **37.42** | 0.885 |
| OT | VP-ODE | 80 | 7.85 | 0.120 | 34.24 | 0.858 |
| OT | ΠGDM | 100 | 6.09 | 0.082 | 36.75 | **0.901** |
| VP-SDE | OT-ODE | 80 | **4.85** | **0.079** | 37.64 | 0.887 |
| VP-SDE | VP-ODE | 80 | 7.21 | 0.117 | 34.33 | 0.860 |
| VP-SDE | ΠGDM | 100 | 5.79 | 0.081 | 36.81 | 0.902 |
| VP-SDE | RED-Diff | 1000 | 7.29 | **0.079** | **39.14** | **0.925** |

Table 12: Quantitative evaluation of linear inverse problems on face-blurred ImageNet-$128 \times 128$

| Model | Inference | NFEs ↓ | Inpainting-*Center*, $\sigma_y = 0$ | | | |
|---|---|---|---|---|---|---|
| | | | FID ↓ | LPIPS ↓ | PSNR ↑ | SSIM ↑ |
| OT | OT-ODE | 70 | 5.88 | **0.095** | **37.06** | 0.894 |
| OT | VP-ODE | 70 | 8.63 | 0.144 | 34.48 | 0.864 |
| OT | ΠGDM | 100 | **5.82** | 0.097 | 36.89 | **0.908** |
| VP-SDE | OT-ODE | 70 | 5.93 | 0.094 | 37.31 | 0.898 |
| VP-SDE | VP-ODE | 70 | 8.08 | 0.142 | 34.55 | 0.865 |
| VP-SDE | ΠGDM | 100 | 5.74 | 0.095 | 37.01 | 0.911 |
| VP-SDE | RED-Diff | 1000 | **5.40** | **0.068** | **38.91** | **0.928** |

Table 13: Quantitative evaluation of linear inverse problems on face-blurred ImageNet-$128 \times 128$

| Model | Inference | NFEs ↓ | SR 2×, $\sigma_y = 0$ | | | | Gaussian deblur, $\sigma_y = 0$ | | | |
|---|---|---|---|---|---|---|---|---|---|---|
| | | | FID ↓ | LPIPS ↓ | PSNR ↑ | SSIM ↑ | FID ↓ | LPIPS ↓ | PSNR ↑ | SSIM ↑ |
| OT | OT-ODE | 70 | **4.46** | **0.097** | 33.88 | 0.903 | **2.09** | **0.048** | 37.49 | **0.961** |
| OT | VP-ODE | 70 | 7.69 | 0.144 | 32.93 | 0.871 | 6.02 | 0.108 | 34.73 | 0.925 |
| OT | ΠGDM | 100 | 6.09 | 0.105 | **34.28** | **0.910** | 4.28 | 0.066 | **37.56** | **0.961** |
| VP-SDE | OT-ODE | 70 | 4.62 | 0.096 | 33.95 | 0.906 | **2.26** | **0.046** | **37.79** | **0.967** |
| VP-SDE | VP-ODE | 70 | 7.91 | 0.144 | 32.87 | 0.869 | 5.64 | 0.105 | 34.81 | 0.928 |
| VP-SDE | ΠGDM | 100 | 6.02 | 0.104 | 34.33 | 0.911 | 4.35 | 0.065 | 37.70 | 0.963 |
| VP-SDE | RED-Diff | 1000 | **3.90** | **0.082** | **34.47** | **0.92** | 4.19 | 0.085 | 34.68 | 0.929 |

Table 14: Quantitative evaluation of linear inverse problems on AFHQ-$256 \times 256$

| Model | Inference | NFEs ↓ | SR 4×, $\sigma_y = 0$ | | | | Gaussian deblur, $\sigma_y = 0$ | | | |
|---|---|---|---|---|---|---|---|---|---|---|
| | | | FID ↓ | LPIPS ↓ | PSNR ↑ | SSIM ↑ | FID ↓ | LPIPS ↓ | PSNR ↑ | SSIM ↑ |
| OT | OT-ODE | 100 | **5.75** | **0.169** | 32.25 | 0.792 | **6.63** | **0.213** | 31.29 | 0.722 |
| OT | VP-ODE | 100 | 6.14 | 0.194 | 31.93 | 0.773 | 7.38 | 0.231 | 31.10 | 0.705 |
| OT | ΠGDM | 100 | 8.89 | 0.173 | **32.57** | **0.812** | 9.78 | 0.209 | **31.54** | **0.743** |
| VP-SDE | OT-ODE | 100 | **6.58** | **0.178** | 32.18 | 0.789 | **8.24** | **0.226** | 31.21 | 0.717 |
| VP-SDE | VP-ODE | 100 | 8.00 | 0.225 | 31.48 | 0.742 | 9.19 | 0.252 | 30.91 | 0.688 |
| VP-SDE | ΠGDM | 100 | 10.85 | 0.189 | **32.52** | **0.811** | 11.46 | 0.228 | **31.47** | **0.738** |
| VP-SDE | RED-Diff | 1000 | 8.65 | 0.191 | 32.21 | 0.801 | 11.67 | 0.268 | 31.30 | 0.731 |

Table 15: Quantitative evaluation of linear inverse problems on AFHQ-$256 \times 256$

| Model | Inference | NFEs ↓ | Inpainting-*Center*, $\sigma_y = 0$ | | | | Inpainting-*Free-form*, $\sigma_y = 0$ | | | |
|---|---|---|---|---|---|---|---|---|---|---|
| | | | FID ↓ | LPIPS ↓ | PSNR ↑ | SSIM ↑ | FID ↓ | LPIPS ↓ | PSNR ↑ | SSIM ↑ |
| OT | OT-ODE | 100 | 8.87 | **0.061** | **37.45** | 0.921 | **4.98** | **0.097** | 36.15 | 0.889 |
| OT | VP-ODE | 100 | 9.18 | 0.106 | 35.63 | 0.898 | 6.92 | 0.135 | 34.72 | 0.869 |
| OT | ΠGDM | 100 | **7.36** | 0.080 | **37.45** | **0.933** | 6.52 | 0.100 | **36.58** | **0.913** |
| VP-SDE | OT-ODE | 100 | 9.95 | 0.064 | 37.49 | 0.918 | **5.39** | 0.099 | 36.15 | 0.887 |
| VP-SDE | VP-ODE | 100 | 10.50 | 0.112 | 35.59 | 0.893 | 7.36 | 0.139 | 34.65 | 0.865 |
| VP-SDE | ΠGDM | 100 | 8.61 | 0.088 | 37.27 | 0.925 | 7.25 | 0.109 | 36.37 | **0.906** |
| VP-SDE | RED-Diff | 1000 | **8.53** | **0.050** | **38.89** | **0.951** | 7.27 | **0.090** | **36.88** | 0.892 |

## C.1 Additional qualitative results

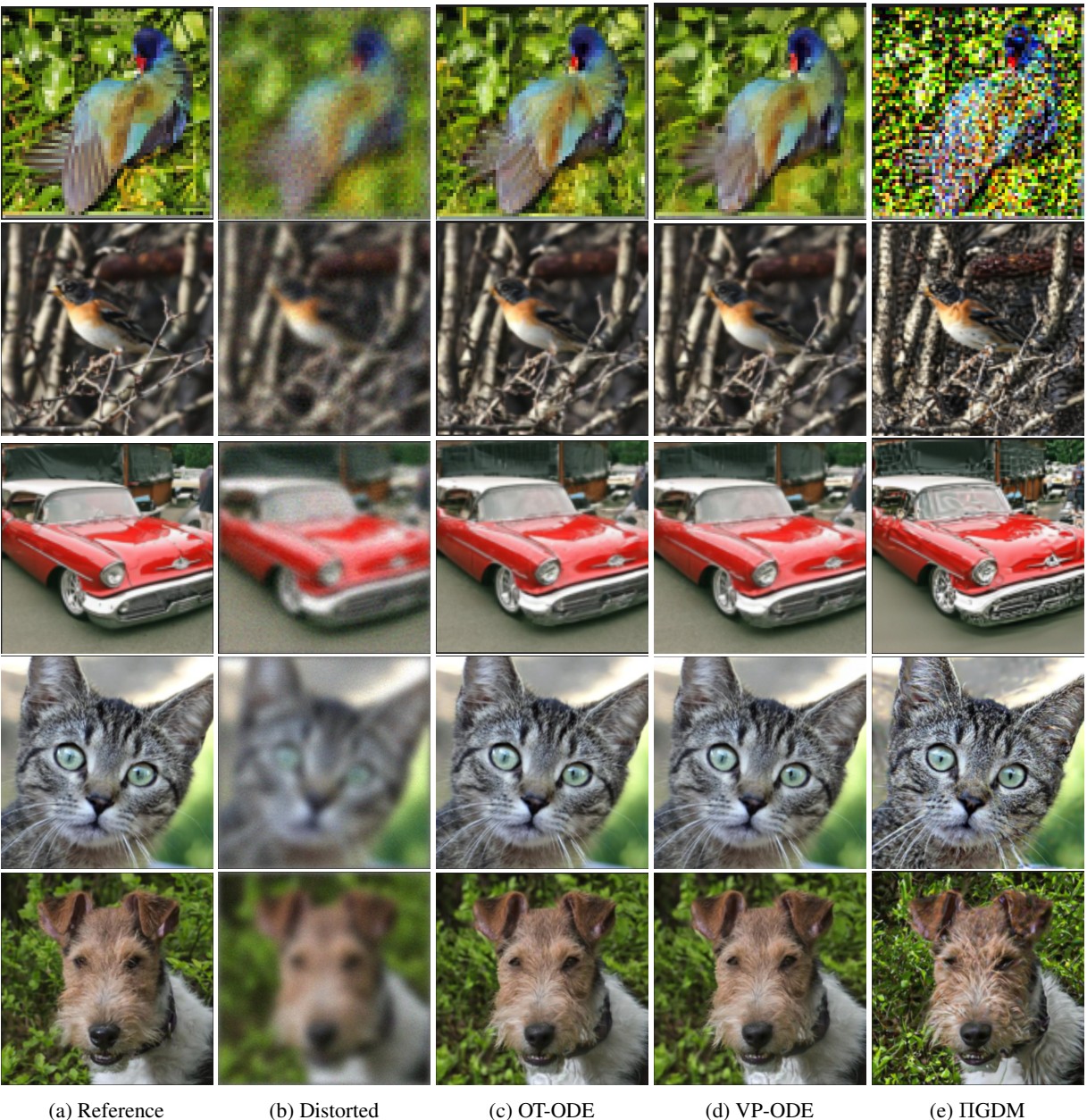

(a) Reference      (b) Distorted      (c) OT-ODE      (d) VP-ODE      (e) ΠGDM

Figure 12: Gaussian-deblur with conditional OT model and $\sigma_y = 0.05$ for (**first row**) face-blurred ImageNet-64, (**second and third row**) face-blurred ImageNet-128, and ( **fourth and fifth row**) AFHQ.

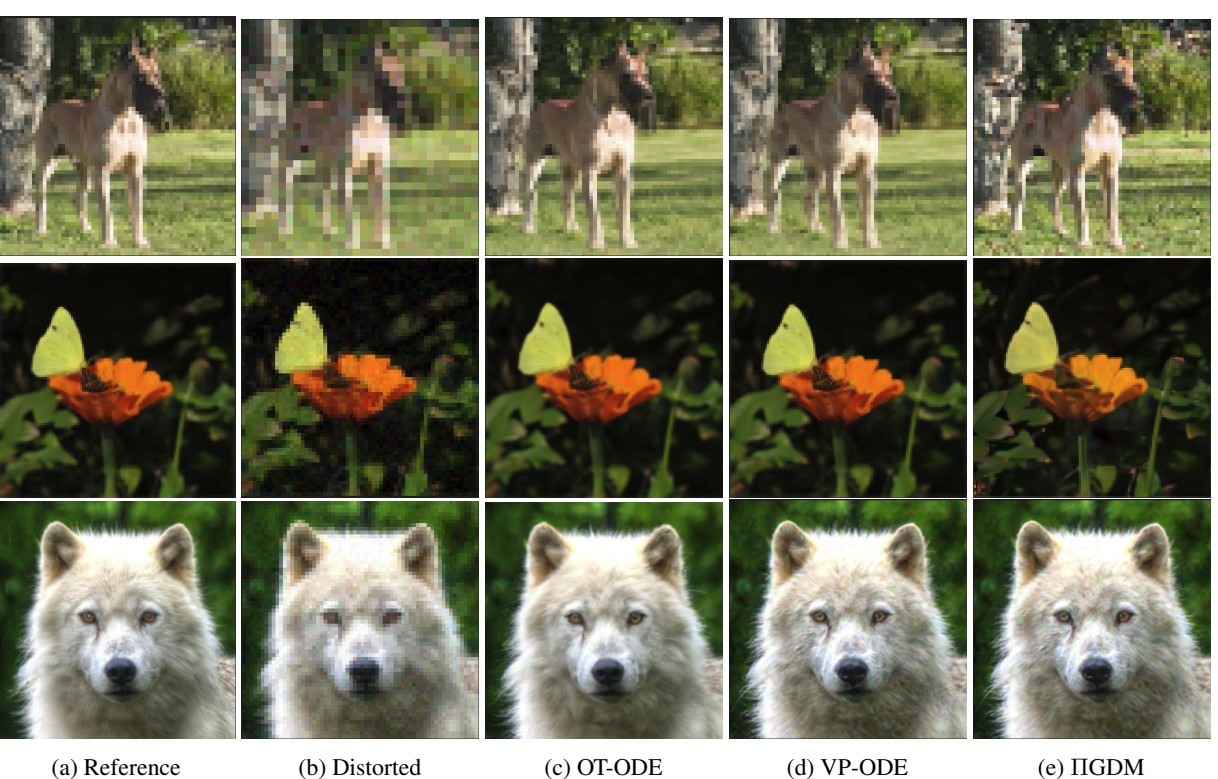

|   (a) Reference   |   (b) Distorted   |   (c) OT-ODE   |   (d) VP-ODE   |   (e) ΠGDM   |

Figure 13: Super-resolution with conditional OT model and $\sigma_y = 0.05$ for (**first row**) face-blurred ImageNet-64 2×, (**second row**) face-blurred ImageNet-128 2×, and (**third row**) AFHQ 4×.

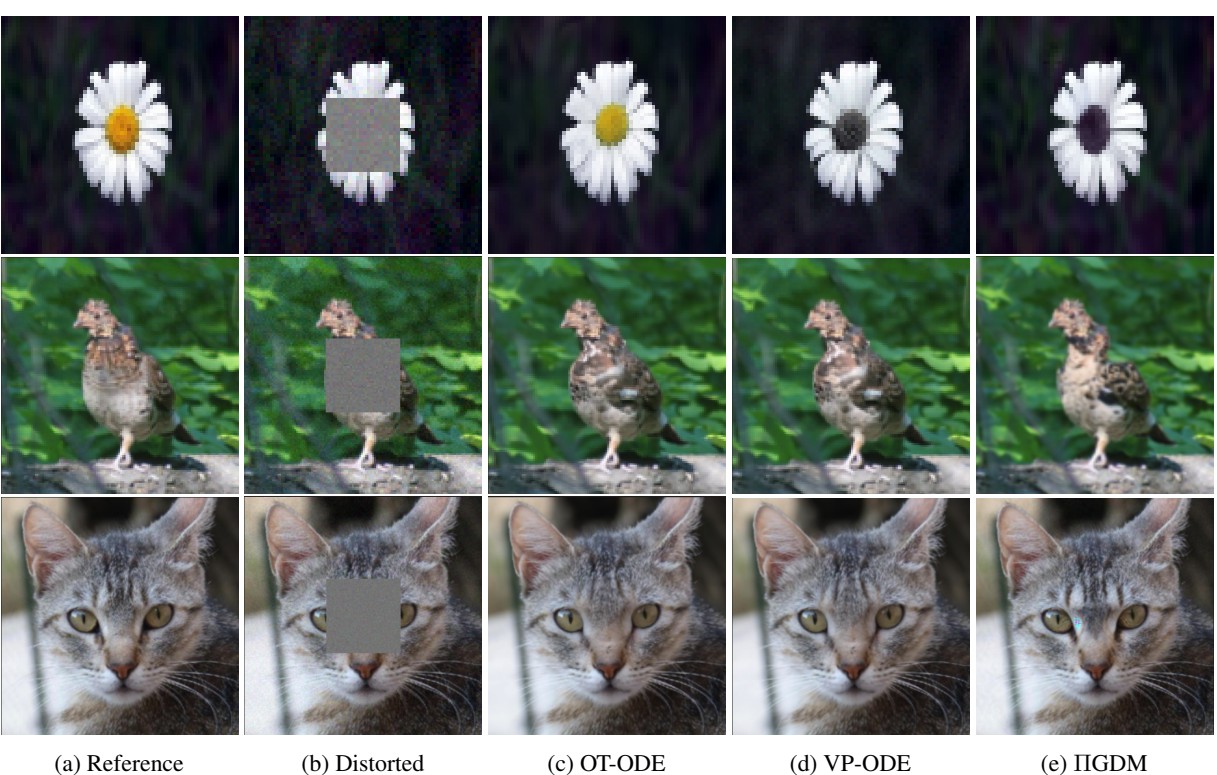

(a) Reference      (b) Distorted      (c) OT-ODE      (d) VP-ODE      (e) ΠGDM

Figure 14: Inpainting (Center mask) with conditional OT model and $\sigma_y = 0.05$ for (**first row**) face-blurred ImageNet-64, (**second row**) face-blurred ImageNet-128, and (**third row**) AFHQ.

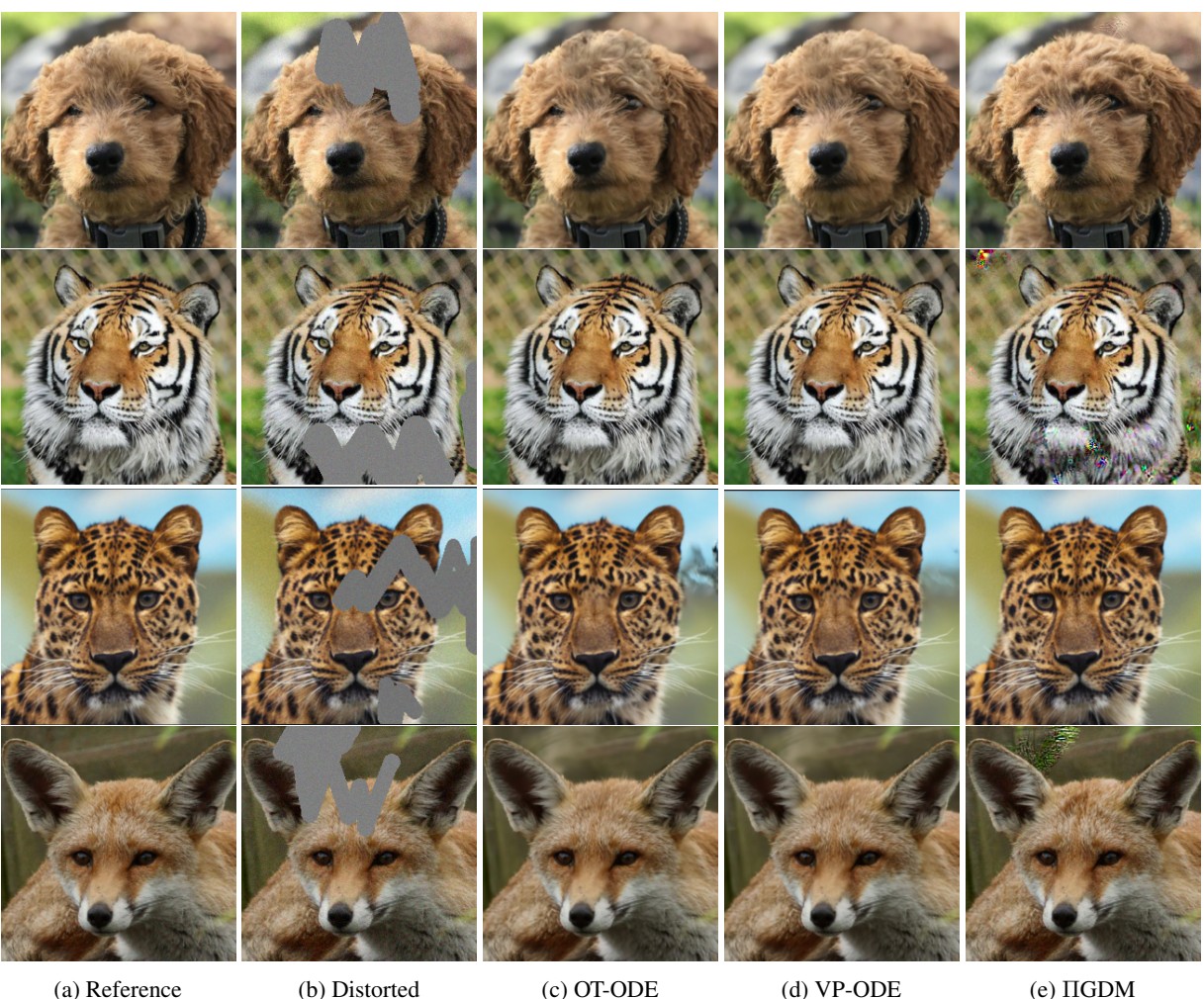

(a) Reference      (b) Distorted      (c) OT-ODE      (d) VP-ODE      (e) ΠGDM

Figure 15: Inpainting (Free-form mask) with conditional OT model and $\sigma_y = 0.05$ for AFHQ.

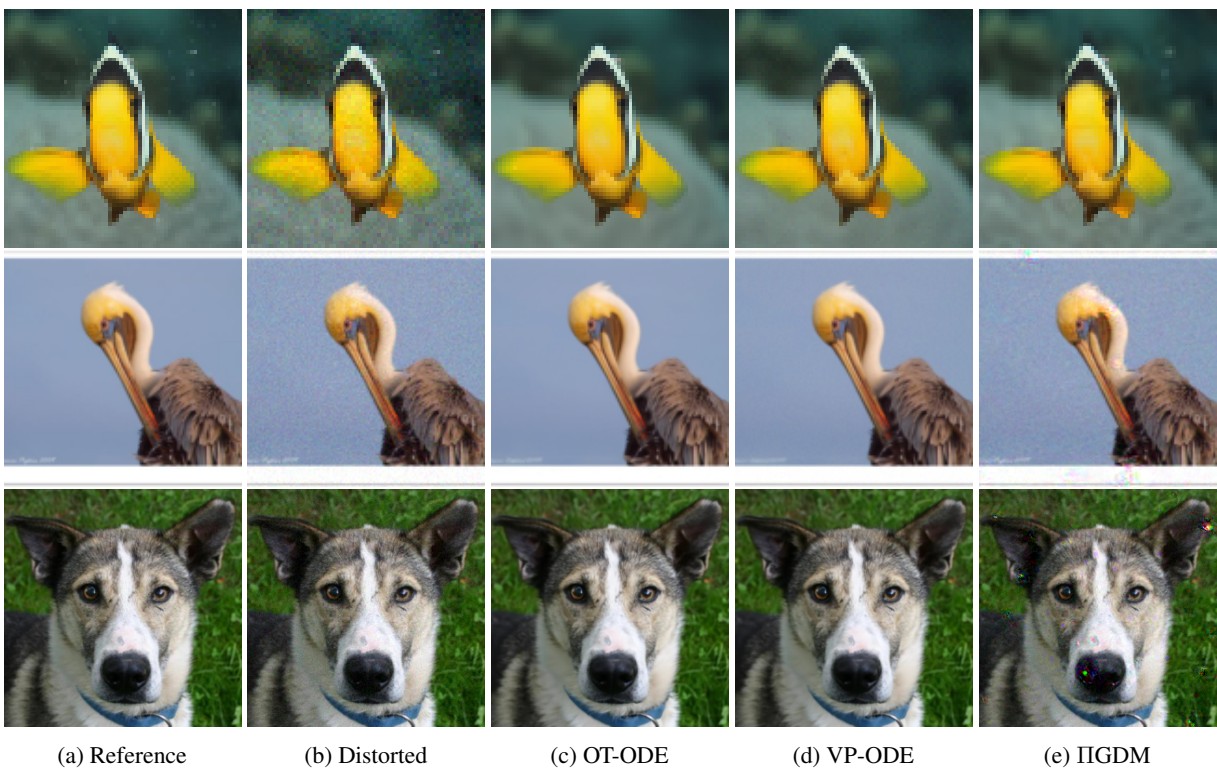

|     (a) Reference     |     (b) Distorted     |     (c) OT-ODE     |     (d) VP-ODE     |     (e) ΠGDM     |

Figure 16: Denoising with conditional OT model and $\sigma_y = 0.05$ for (**first row**) face-blurred ImageNet-64, (**second row**) face-blurred ImageNet-128, and (**third row**) AFHQ.

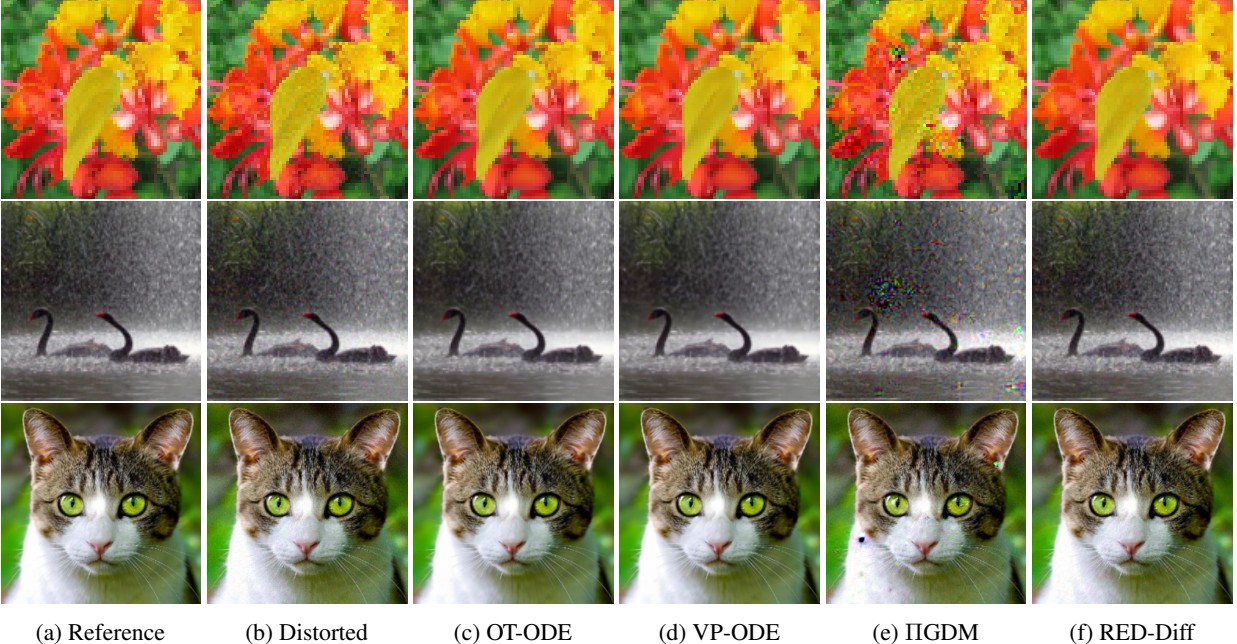

|  (a) Reference  |  (b) Distorted  |  (c) OT-ODE  |  (d) VP-ODE  |  (e) ΠGDM  |  (f) RED-Diff  |

Figure 17: Denoising with pretrained VP-SDE model and $\sigma_y = 0.05$ for (**first row**) face-blurred ImageNet-64, (**second row**) face-blurred ImageNet-128, and (**third row**) AFHQ.

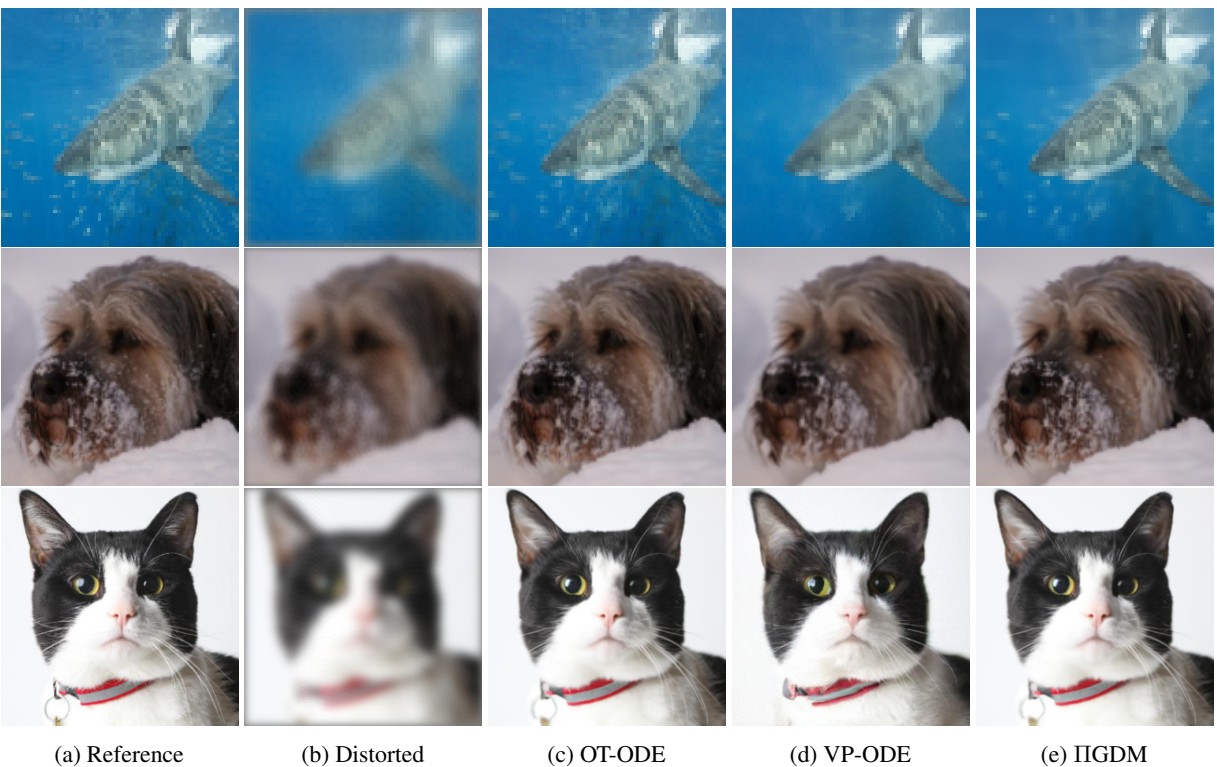

(a) Reference    (b) Distorted    (c) OT-ODE    (d) VP-ODE    (e) ΠGDM

Figure 18: Gaussian deblurring with conditional OT model and $\sigma_y = 0$ for (**first row**) face-blurred ImageNet-64, (**second row**) face-blurred ImageNet-128 and (**third row**) AFHQ.

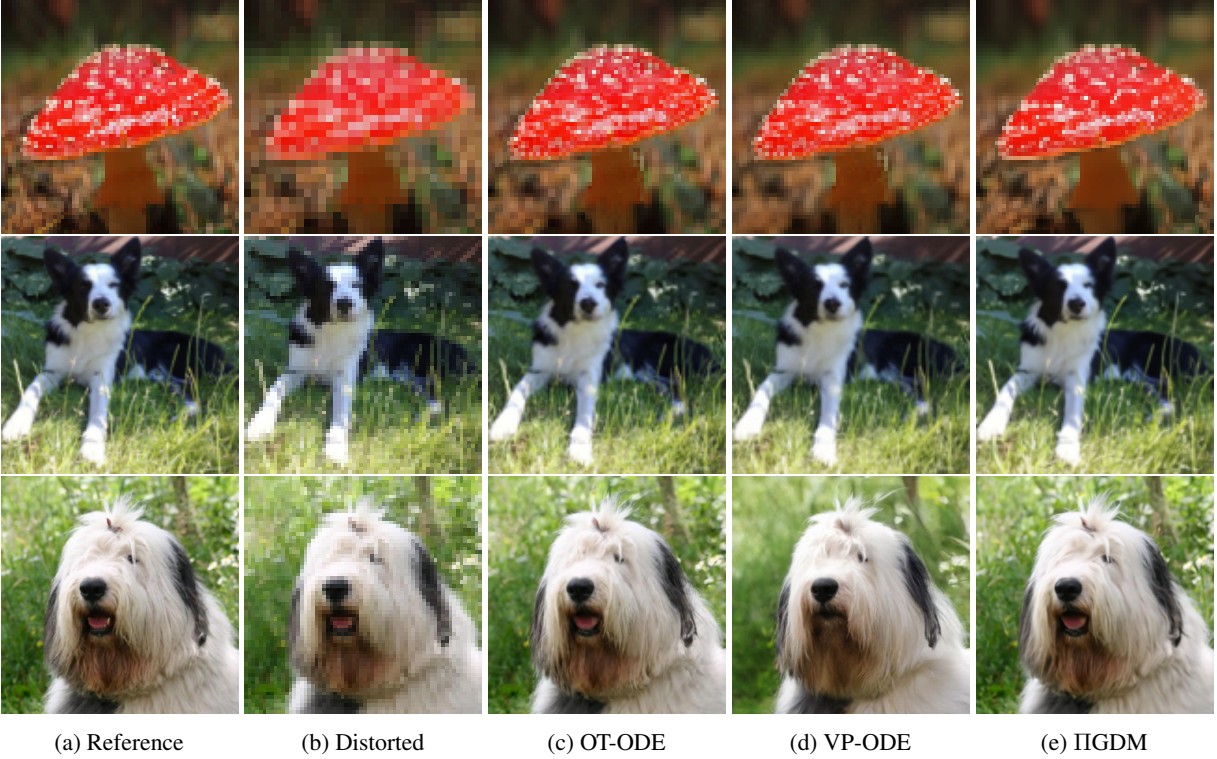

(a) Reference    (b) Distorted    (c) OT-ODE    (d) VP-ODE    (e) ΠGDM

Figure 19: Super-resolution with conditional OT model and $\sigma_y = 0$ for (**first row**) face-blurred ImageNet-64, (**second row**) face-blurred ImageNet-128 and (**third row**) AFHQ.

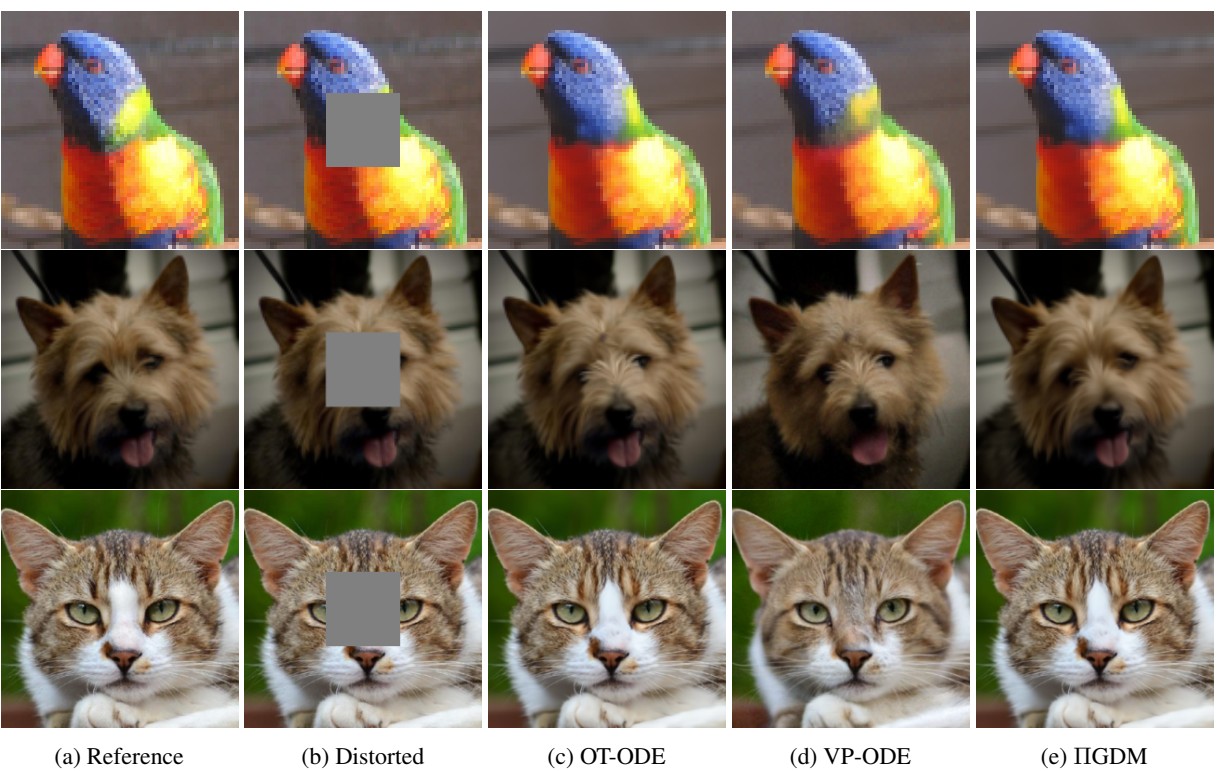

|(a) Reference|(b) Distorted|(c) OT-ODE|(d) VP-ODE|(e) ΠGDM|

Figure 20: Inpainting (centered mask) with conditional OT model and $\sigma_y = 0$ for (**first row**) face-blurred ImageNet-64, (**second row**) face-blurred ImageNet-128 and (**third row**) AFHQ.

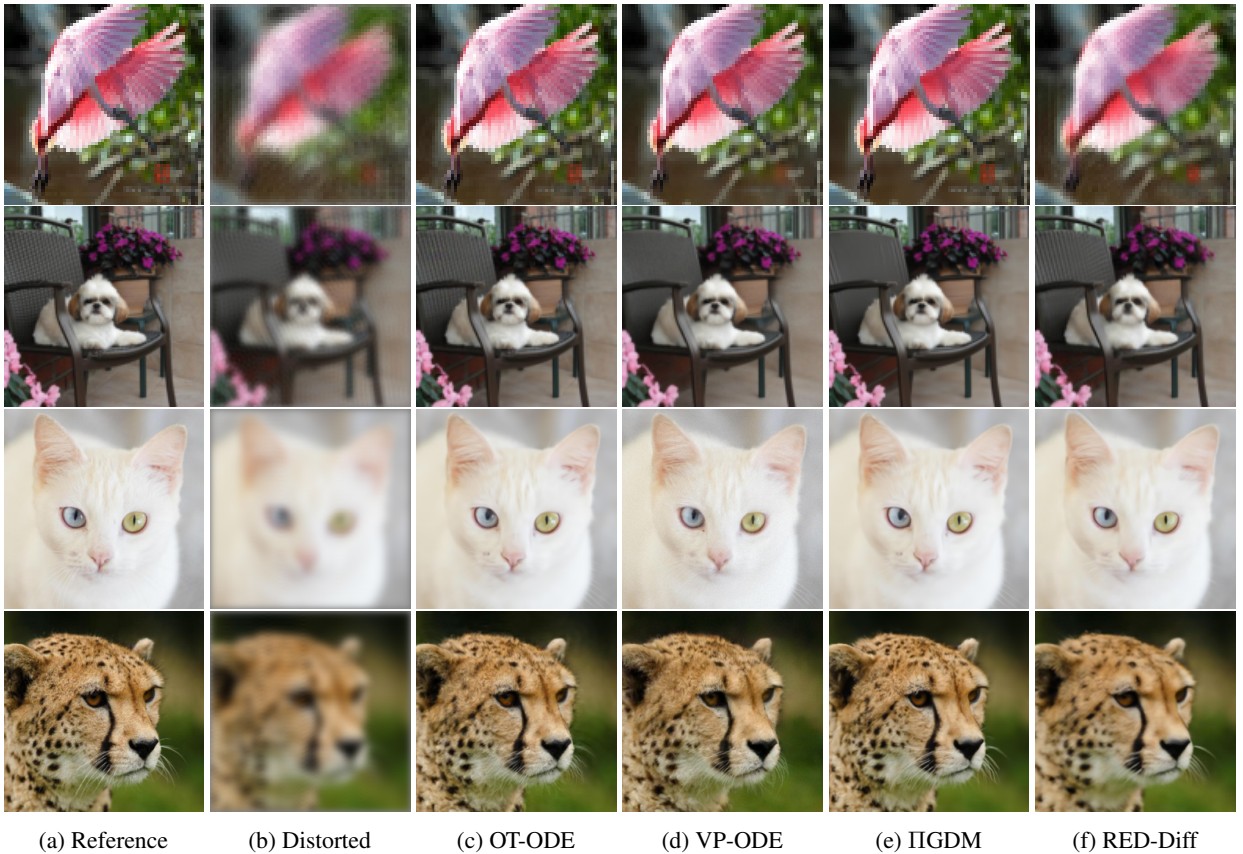

   (a) Reference      (b) Distorted      (c) OT-ODE      (d) VP-ODE      (e) ΠGDM      (f) RED-Diff

Figure 21: Gaussian deblurring with VP-SDE model and $\sigma_y = 0$ for (**first row**) face-blurred ImageNet-64, (**second row**) face-blurred ImageNet-128 and (**third and fourth row**) AFHQ.

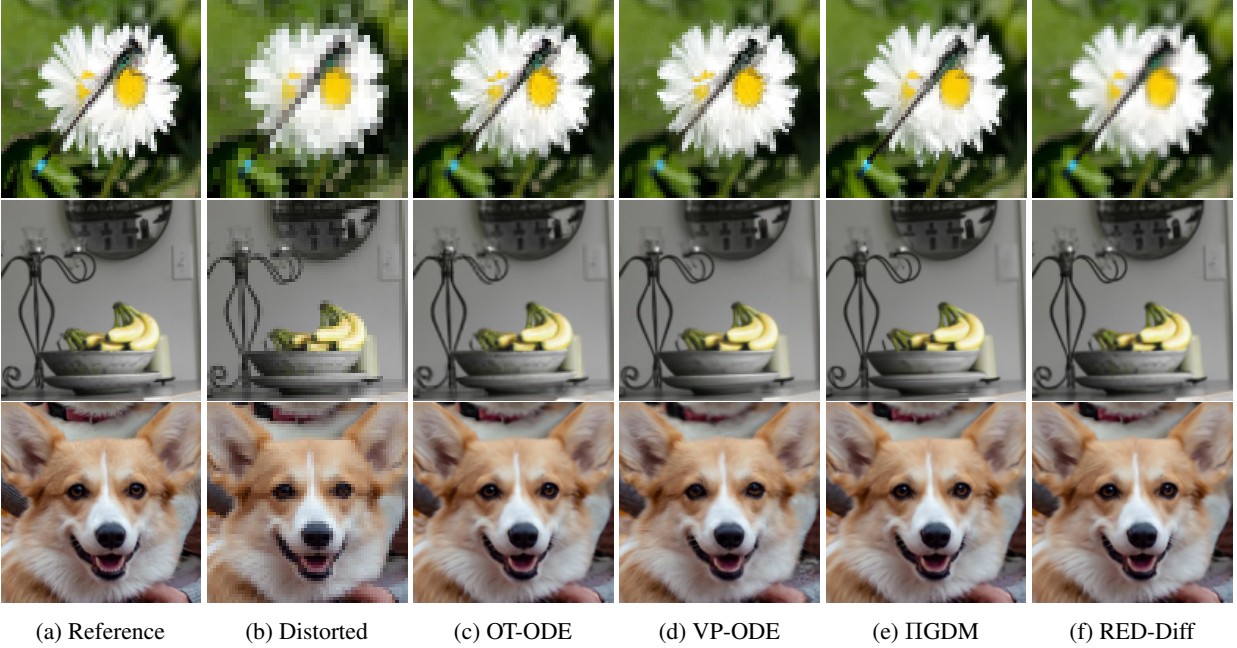

   (a) Reference      (b) Distorted      (c) OT-ODE      (d) VP-ODE      (e) ΠGDM      (f) RED-Diff

Figure 22: Super-resolution with VP-SDE model and $\sigma_y = 0$ for (**first row**) face-blurred ImageNet-64, (**second row**) face-blurred ImageNet-128 and (**third row**) AFHQ.

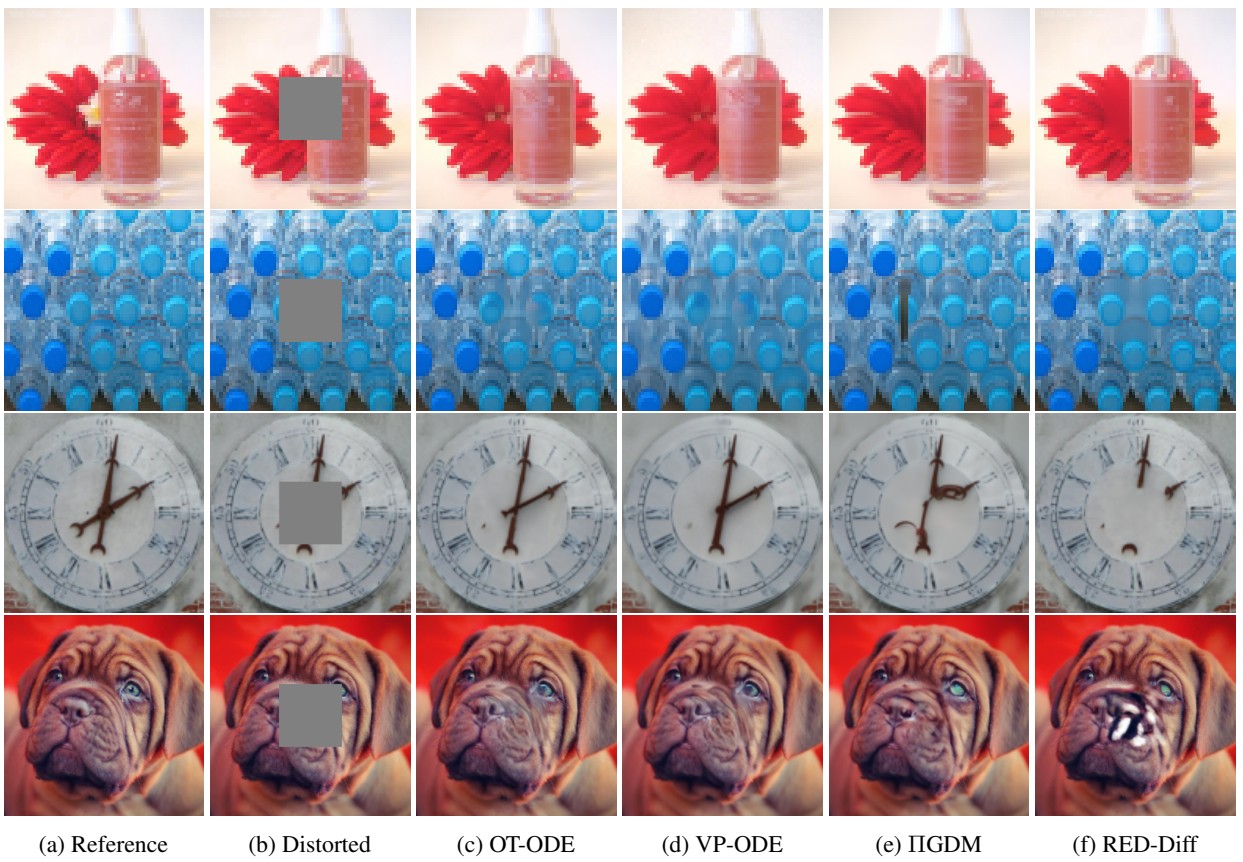

| (a) Reference | (b) Distorted | (c) OT-ODE | (d) VP-ODE | (e) ΠGDM | (f) RED-Diff |

Figure 23: Inpainting (centered mask) with VP-SDE model and $\sigma_y = 0$ for (**first and second row**) face-blurred ImageNet-64, (**third row**) face-blurred ImageNet-128 and (**fourth row**) AFHQ.

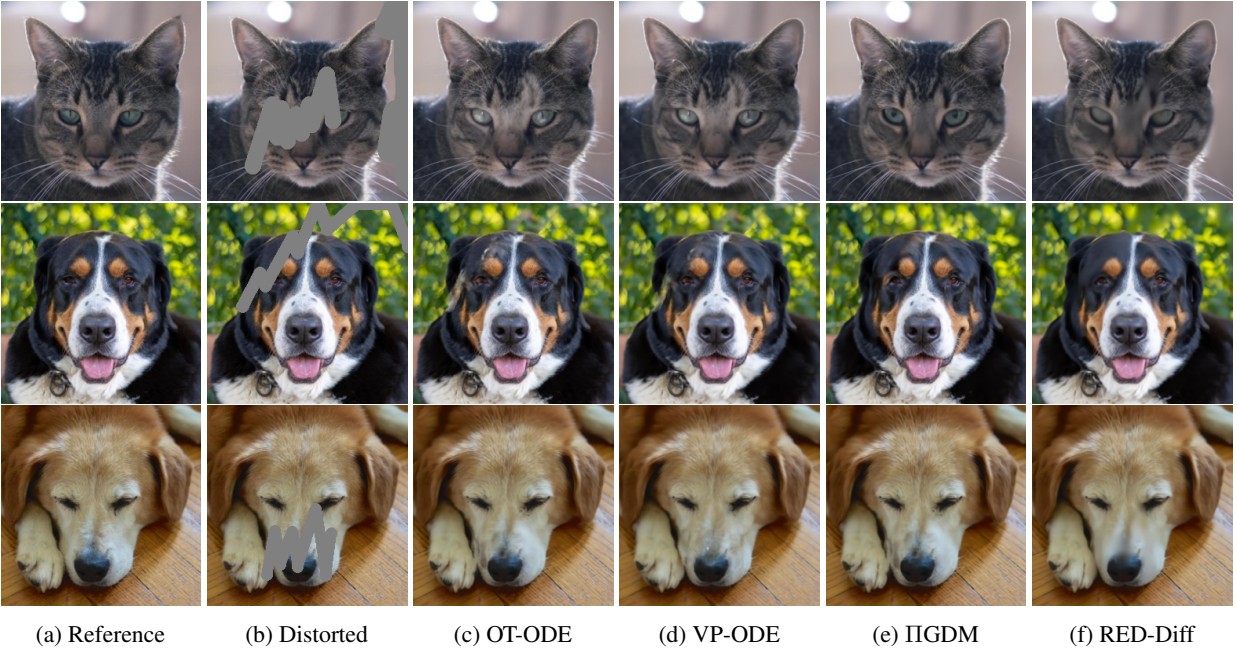

| (a) Reference | (b) Distorted | (c) OT-ODE | (d) VP-ODE | (e) ΠGDM | (f) RED-Diff |

Figure 24: Inpainting (freeform mask) with VP-SDE model and $\sigma_y = 0$ for AFHQ.

## C.2 Negative results from Inpainting

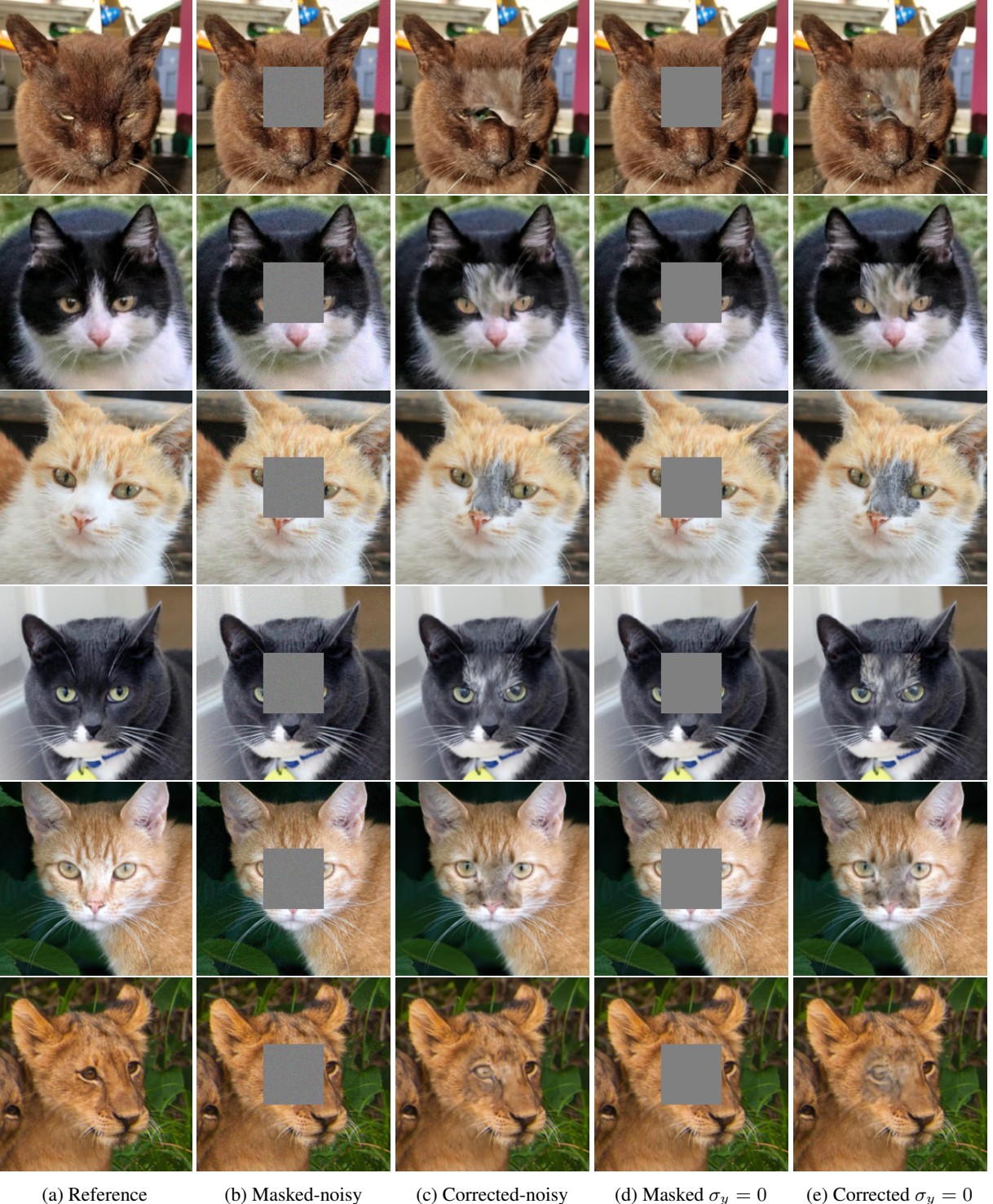

(a) Reference     (b) Masked-noisy     (c) Corrected-noisy     (d) Masked $\sigma_y = 0$     (e) Corrected $\sigma_y = 0$

Figure 25: Negative results for inpainting with OT-ODE on AFHQ. We can observe artifacts in high-resolution images where the masked region is not inpainted correctly and there are patches in the inpainted region that are semantically incorrect. The observed artifacts are present in both the noiseless (e) and noisy (c) columns.

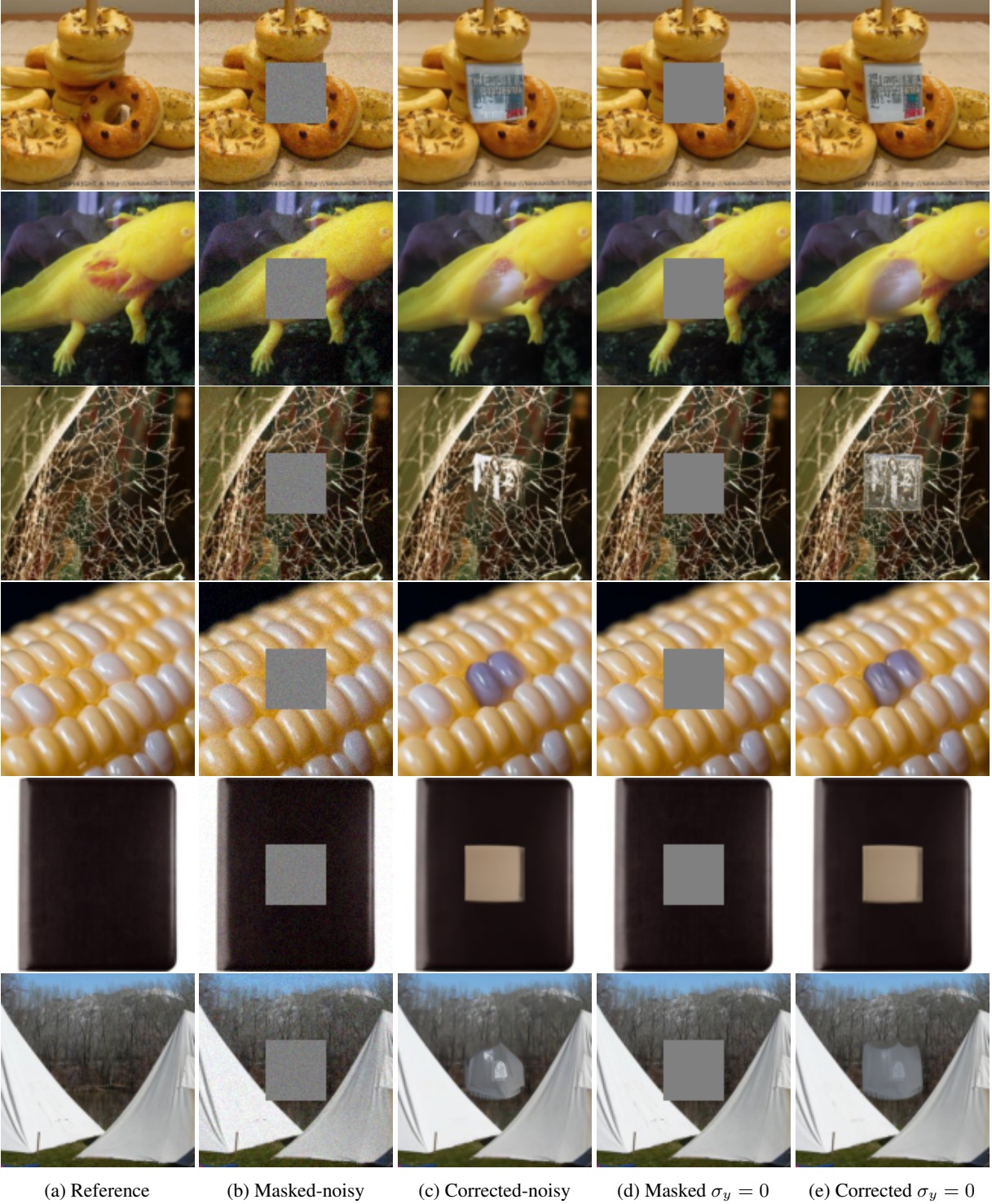

(a) Reference     (b) Masked-noisy     (c) Corrected-noisy     (d) Masked $\sigma_y = 0$     (e) Corrected $\sigma_y = 0$

Figure 26: Negative results for inpainting with OT-ODE on face-blurred ImageNet-128. We can observe artifacts in high-resolution images where the masked region is not inpainted correctly and there are patches in the inpainted region that are semantically incorrect. The observed artifacts are present in both the noiseless (e) and noisy (c) columns.

# D  Noiseless null and range space decomposition

When $\sigma_y^2 = 0$, we can produce a vector field approximation with even lower Conditional Flow Matching loss by applying a null-space and range-space decomposition motivated by DDNM (Wang et al., 2022). In particular, when $\boldsymbol{y} = \boldsymbol{A}\boldsymbol{x}_1$, we have that $\boldsymbol{A}^\dagger\boldsymbol{y} = \boldsymbol{A}^\dagger\boldsymbol{A}\boldsymbol{x}_1$ (where $\boldsymbol{A}^\dagger$ is the pseudo-inverse of $\boldsymbol{A}$) and so

$$\mathbb{E}_q[\boldsymbol{x}_1|\boldsymbol{x}_t, \boldsymbol{y}] = \mathbb{E}_q[\boldsymbol{A}^\dagger\boldsymbol{A}\boldsymbol{x}_1 + (\boldsymbol{I} - \boldsymbol{A}^\dagger\boldsymbol{A})\boldsymbol{x}_1|\boldsymbol{x}_t, \boldsymbol{y}] = \boldsymbol{A}^\dagger\boldsymbol{y} + (\boldsymbol{I} - \boldsymbol{A}^\dagger\boldsymbol{A})\mathbb{E}_q[\boldsymbol{x}_1|\boldsymbol{x}_t, \boldsymbol{y}]. \quad (19)$$

So when $\sigma_y^2 = 0$, it is only necessary to approximate the second term, as the first term is known through $\boldsymbol{y}$. The regression loss is minimized for the first term automatically and $\widehat{\boldsymbol{x}_1}(\boldsymbol{x}_t, \boldsymbol{y})$ is only responsible for predicting the second term.

In our experiments, we find that null space decomposition helps in inpainting but not other measurements. We summarize the results in Table 16 to 21 and show qualitative results for inpainting in Figure 27 to 29.

Table 16: Comparison of performance OT-ODE sampling and OT-ODE sampling with null and range space decomposition (NRSD) on face-blurred ImageNet-$64 \times 64$. For inpainting, OT-ODE sampling with null and range space decomposition outperforms simple OT-ODE sampling.

| Model | Inference | NFEs ↓ | Inpainting-*Center*, $\sigma_y = 0$ | | | |
|---|---|---|---|---|---|---|
| | | | FID ↓ | LPIPS ↓ | PSNR ↑ | SSIM ↑ |
| OT | OT-ODE | 80 | 4.94 | 0.080 | 37.42 | 0.885 |
| OT | OT-ODE-NRSD | 80 | **3.84** | **0.072** | **38.23** | **0.888** |
| OT | VP-ODE | 80 | 7.85 | 0.120 | 34.24 | 0.858 |
| VP-SDE | OT-ODE | 80 | 4.85 | 0.079 | 37.64 | 0.887 |
| VP-SDE | OT-ODE-NRSD | 80 | **3.77** | **0.072** | **38.24** | **0.888** |
| VP-SDE | VP-ODE | 80 | 7.21 | 0.117 | 34.33 | 0.860 |

Table 17: Comparison of performance OT-ODE sampling and OT-ODE sampling with null and range space decomposition (NRSD) on face-blurred ImageNet-$64 \times 64$. For tasks like super-resolution and Gaussian deblurring, OT-ODE sampling without null and range space decomposition outperforms other methods.

| Model | Inference | NFEs ↓ | SR 2×, $\sigma_y = 0$ | | | | Gaussian deblur, $\sigma_y = 0$ | | | |
|---|---|---|---|---|---|---|---|---|---|---|
| | | | FID ↓ | LPIPS ↓ | PSNR ↑ | SSIM ↑ | FID ↓ | LPIPS ↓ | PSNR ↑ | SSIM ↑ |
| OT | OT-ODE | 80 | **6.46** | **0.119** | **31.59** | **0.839** | **2.59** | **0.038** | **35.31** | **0.961** |
| OT | OT-ODE-NRSD | 80 | 7.37 | 0.134 | 31.05 | 0.799 | 3.05 | 0.044 | 35.19 | 0.956 |
| OT | VP-ODE | 80 | 8.29 | 0.147 | 31.20 | 0.817 | 6.13 | 0.083 | 33.31 | 0.929 |
| VP-SDE | OT-ODE | 80 | **6.32** | **0.118** | **31.60** | **0.839** | **2.61** | **0.037** | **35.45** | **0.963** |
| VP-SDE | OT-ODE-NRSD | 80 | 7.13 | 0.133 | 31.06 | 0.798 | 2.99 | 0.044 | 35.24 | 0.956 |
| VP-SDE | VP-ODE | 80 | 7.76 | 0.145 | 31.21 | 0.817 | 5.68 | 0.080 | 33.37 | 0.931 |

Table 18: Comparison of performance OT-ODE sampling and OT-ODE sampling with null and range space decomposition (NRSD) on face-blurred ImageNet-$128 \times 128$.

| Model | Inference | NFEs ↓ | SR 2×, $\sigma_y = 0$ | | | | Gaussian deblur, $\sigma_y = 0$ | | | |
|---|---|---|---|---|---|---|---|---|---|---|
| | | | FID ↓ | LPIPS ↓ | PSNR ↑ | SSIM ↑ | FID ↓ | LPIPS ↓ | PSNR ↑ | SSIM ↑ |
| OT | OT-ODE | 70 | 4.46 | **0.097** | **33.88** | **0.903** | 2.09 | 0.048 | 37.49 | 0.961 |
| OT | OT-ODE-NRSD | 70 | **3.62** | 0.099 | 33.24 | 0.876 | **1.42** | **0.036** | **38.35** | **0.969** |
| OT | VP-ODE | 70 | 7.69 | 0.144 | 32.93 | 0.871 | 6.02 | 0.108 | 34.73 | 0.925 |
| VP-SDE | OT-ODE | 70 | 4.62 | **0.096** | **33.95** | **0.906** | 2.26 | 0.046 | 37.79 | 0.967 |
| VP-SDE | OT-ODE-NRSD | 70 | **3.44** | 0.098 | 33.28 | 0.877 | **1.36** | **0.035** | **38.44** | **0.969** |
| VP-SDE | VP-ODE | 70 | 7.91 | 0.144 | 32.87 | 0.869 | 5.64 | 0.105 | 34.81 | 0.928 |

Table 19: Comparison of performance OT-ODE sampling and OT-ODE sampling with null and range space decomposition (NRSD) on face-blurred ImageNet-$128 \times 128$

| Model | Inference | NFEs ↓ | Inpainting-*Center*, $\sigma_y = 0$ | | | |
|---|---|---|---|---|---|---|
| | | | FID ↓ | LPIPS ↓ | PSNR ↑ | SSIM ↑ |
| OT | OT-ODE | 70 | 5.88 | 0.095 | 37.06 | 0.894 |
| OT | OT-ODE-NRSD | 70 | **3.95** | **0.074** | **38.27** | **0.906** |
| OT | VP-ODE | 70 | 8.63 | 0.144 | 34.48 | 0.864 |
| VP-SDE | OT-ODE | 70 | 5.93 | 0.094 | 37.31 | 0.898 |
| VP-SDE | OT-ODE-NRSD | 70 | **3.84** | **0.073** | **38.27** | **0.906** |
| VP-SDE | VP-ODE | 70 | 8.08 | 0.142 | 34.55 | 0.865 |

Table 20: Comparison of performance OT-ODE sampling and OT-ODE sampling with null and range space decomposition (NRSD) on AFHQ-$256 \times 256$

| Model | Inference | NFEs ↓ | SR 4×, $\sigma_y = 0$ | | | | Gaussian deblur, $\sigma_y = 0$ | | | |
|---|---|---|---|---|---|---|---|---|---|---|
| | | | FID ↓ | LPIPS ↓ | PSNR ↑ | SSIM ↑ | FID ↓ | LPIPS ↓ | PSNR ↑ | SSIM ↑ |
| OT | OT-ODE | 100 | 5.75 | **0.169** | **32.25** | **0.792** | **6.63** | **0.213** | **31.29** | **0.722** |
| OT | OT-ODE-NRSD | 100 | **5.73** | 0.179 | 31.69 | 0.753 | 7.32 | 0.237 | 30.72 | 0.665 |
| OT | VP-ODE | 100 | 6.14 | 0.194 | 31.93 | 0.773 | 7.38 | 0.231 | 31.10 | 0.705 |
| VP-SDE | OT-ODE | 100 | **6.58** | **0.178** | **32.18** | **0.789** | **8.24** | **0.226** | **31.21** | **0.717** |
| VP-SDE | OT-ODE-NRSD | 100 | 6.99 | 0.195 | 31.65 | 0.752 | 10.19 | 0.255 | 30.66 | 0.662 |
| VP-SDE | VP-ODE | 100 | 8.00 | 0.225 | 31.48 | 0.742 | 9.19 | 0.252 | 30.91 | 0.688 |

Table 21: Comparison of performance OT-ODE sampling and OT-ODE sampling with null and range space decomposition (NRSD) on AFHQ-$256 \times 256$

| Model | Inference | NFEs ↓ | Inpainting-*Center*, $\sigma_y = 0$ | | | | Inpainting-*Free-form*, $\sigma_y = 0$ | | | |
|---|---|---|---|---|---|---|---|---|---|---|
| | | | FID ↓ | LPIPS ↓ | PSNR ↑ | SSIM ↑ | FID ↓ | LPIPS ↓ | PSNR ↑ | SSIM ↑ |
| OT | OT-ODE | 100 | 8.87 | 0.061 | 37.45 | **0.921** | 4.98 | 0.097 | 36.15 | 0.889 |
| OT | OT-ODE-NRSD | 100 | **7.95** | **0.046** | **38.01** | **0.921** | **4.12** | **0.083** | **36.62** | **0.890** |
| OT | VP-ODE | 100 | 9.18 | 0.106 | 35.63 | 0.898 | 6.92 | 0.135 | 34.72 | 0.869 |
| VP-SDE | OT-ODE | 100 | **9.95** | 0.064 | 37.49 | **0.918** | 5.39 | 0.099 | 36.15 | **0.887** |
| VP-SDE | OT-ODE-NRSD | 100 | 10.96 | **0.052** | **37.95** | 0.916 | **4.87** | **0.089** | **36.52** | 0.884 |
| VP-SDE | VP-ODE | 100 | 10.50 | 0.112 | 35.59 | 0.893 | 7.36 | 0.139 | 34.65 | 0.865 |

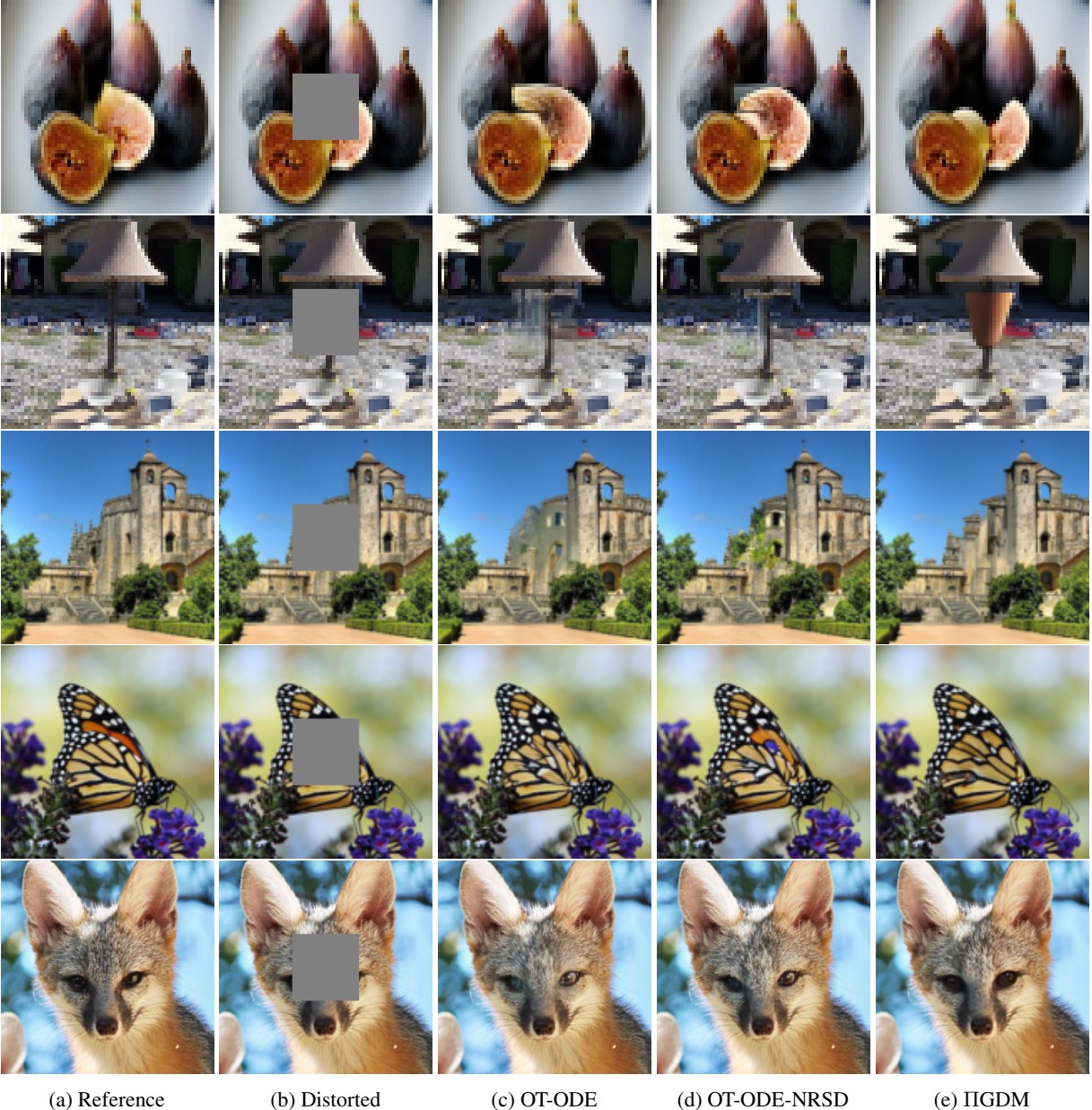

(a) Reference    (b) Distorted    (c) OT-ODE    (d) OT-ODE-NRSD    (e) ΠGDM

Figure 27: Comparison of inpainting (center mask) via OT-ODE sampling with and without null and range space decomposition (NRSD). We use conditional OT model and $\sigma_y = 0$ for (**first and second row**) face-blurred ImageNet-64, (**third row**) face-blurred ImageNet-128, and (**fourth row**) AFHQ.

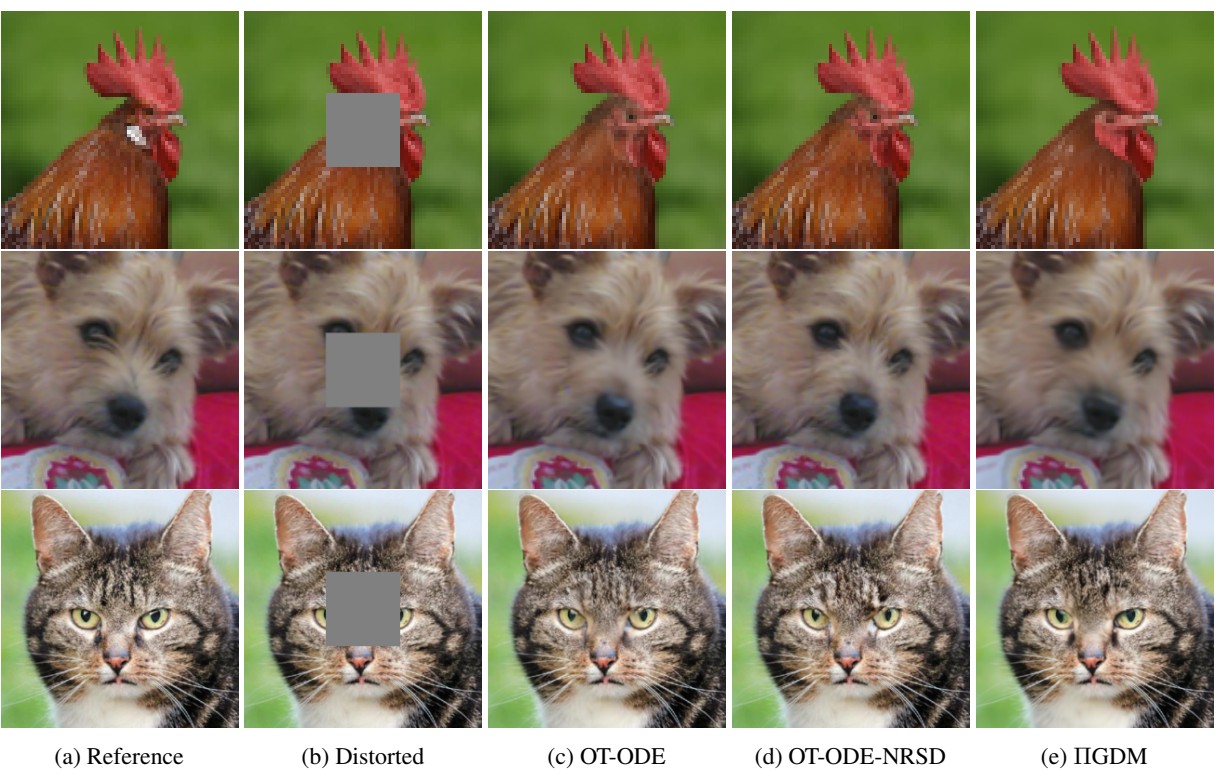

(a) Reference      (b) Distorted      (c) OT-ODE      (d) OT-ODE-NRSD      (e) ΠGDM

Figure 28: Comparison of inpainting (center mask) via OT-ODE sampling with and without null and range space decomposition (NRSD) for (**first row**) face-blurred ImageNet-64, (**second row**) face-blurred ImageNet-128, and (**third row**) AFHQ. We use VP-SDE model and $\sigma_y = 0$.

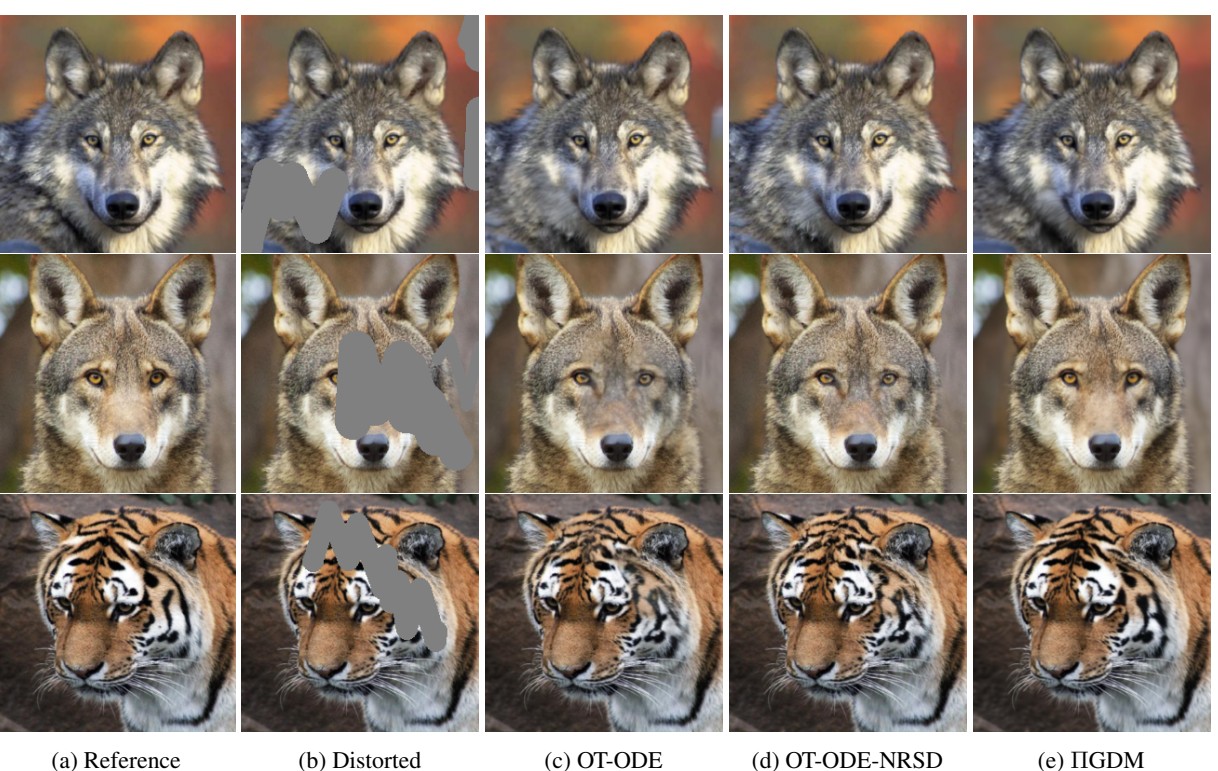

(a) Reference      (b) Distorted      (c) OT-ODE      (d) OT-ODE-NRSD      (e) ΠGDM

Figure 29: Comparison of inpainting (free-form mask) via OT-ODE sampling with and without null and range space decomposition (NRSD) for AFHQ. We use conditional OT model and $\sigma_y = 0$.

# E   Baselines

## E.1   ΠGDM

**Implementation details.**   We closely follow the official code available on github while implementing ΠGDM. For noisy case, we closely follow the Algorithm 1 in the appendix of Song et al. (2022). We use adaptive weighted guidance for both noiseless and noisy cases as in the original work. We always use uniform spacing while iterating the timestep over 100 steps. We use ascending time from 0 to 1. Note that the original paper uses descending time from $T$ to 0. According to the notational convention used in this paper, this is equivalent to ascending time from 0 to 1. For the choice of $r_t^2$, we consider the values derived from both variance exploding formulation and variance preserving formulation.

**Value of $r_t^2$.**   ΠGDM sets the value of $r_t^2 = \frac{\sigma_{1-t}^2}{1+\sigma_{1-t}^2}$ for VE-SDE, where $q(\boldsymbol{x}_t|\boldsymbol{x}_1) = \mathcal{N}(\boldsymbol{x}_1, \sigma_{1-t}^2\boldsymbol{I})$. We can follow the same procedure as outlined in Song et al. (2022), and solve for $r_t^2$ in closed form for VP-SDE. We know for that VP-SDE, $q(\boldsymbol{x}_t|\boldsymbol{x}_1) = \mathcal{N}(\alpha_{1-t}\boldsymbol{x}_1, (1-\alpha_{1-t}^2)\boldsymbol{I})$, where $\alpha_t = e^{-\frac{1}{2}T(t)}$, $T(t) = \int_0^t \beta(s)ds$, and $\beta(s)$ is the noise scale function. Using equation 16 for VP-SDE gives $r_t^2 = 1 - \alpha_{1-t}^2$. We can also obtain an alternate $r_t^2$ by plugging in value of $\sigma_t^2$ for VP-SDE into the expression of $r_t^2$ derived for VE-SDE, which evaluates to $r_t^2 = \frac{1-\alpha_{1-t}^2}{2-\alpha_{1-t}^2}$. Empirically, we find that $r_t^2$ for VE-SDE marginally outperforms VP-SDE. We report performance of ΠGDM with both choices of $r_t^2$ in Table 22 to 24.

Table 22: Relative performance of ΠGDM on face-blurred ImageNet-64 with VE and VP derived $r_t^2$ with $\sigma_y = 0.05$

| Measurement | Model | VP | | | | VE | | | |
|---|---|---|---|---|---|---|---|---|---|
| | | FID ↓ | LPIPS ↓ | PSNR ↑ | SSIM ↑ | FID ↓ | LPIPS ↓ | PSNR ↑ | SSIM ↑ |
| SR 2× | OT | 6.52 | 0.168 | 30.54 | 0.753 | 5.91 | 0.160 | 30.60 | 0.762 |
| Gaussian deblur | OT | 55.19 | 0.374 | 28.74 | 0.516 | 39.36 | 0.326 | 29.00 | 0.572 |
| Inpainting-*Center* | OT | 9.25 | 0.111 | 34.13 | 0.863 | 8.70 | 0.109 | 34.17 | 0.864 |
| Denoising | OT | 16.59 | 0.102 | 34.60 | 0.906 | 16.44 | 0.101 | 34.64 | 0.907 |
| SR 2× | VP-SDE | 6.84 | 0.174 | 30.48 | 0.743 | 6.11 | 0.166 | 30.54 | 0.753 |
| Gaussian deblur | VP-SDE | 54.77 | 0.376 | 28.74 | 0.511 | 39.14 | 0.329 | 28.99 | 0.567 |
| Inpainting-*Center* | VP-SDE | 9.75 | 0.113 | 34.03 | 0.860 | 9.36 | 0.112 | 34.06 | 0.862 |
| Denoising | VP-SDE | 17.19 | 0.107 | 34.25 | 0.901 | 15.54 | 0.102 | 34.41 | 0.906 |

Table 23: Relative performance of ΠGDM on face-blurred ImageNet-128 with VE and VP derived $r_t^2$ with $\sigma_y = 0.05$

| Measurement | Model | VP | | | | VE | | | |
|---|---|---|---|---|---|---|---|---|---|
| | | FID ↓ | LPIPS ↓ | PSNR ↑ | SSIM ↑ | FID ↓ | LPIPS ↓ | PSNR ↑ | SSIM ↑ |
| SR 2× | OT | 4.38 | 0.148 | 32.07 | 0.831 | 4.26 | 0.145 | 32.12 | 0.834 |
| Gaussian deblur | OT | 30.30 | 0.328 | 29.96 | 0.606 | 22.42 | 0.296 | 30.17 | 0.642 |
| Inpainting-*Center* | OT | 7.99 | 0.122 | 34.57 | 0.867 | 7.64 | 0.120 | 34.61 | 0.869 |
| Denoising | OT | 9.60 | 0.107 | 35.11 | 0.903 | 9.30 | 0.104 | 35.21 | 0.906 |
| SR 2× | VP-SDE | 7.55 | 0.183 | 31.61 | 0.785 | 6.14 | 0.168 | 31.79 | 0.803 |
| Gaussian deblur | VP-SDE | 55.61 | 0.463 | 28.57 | 0.414 | 41.69 | 0.404 | 28.98 | 0.493 |
| Inpainting-*Center* | VP-SDE | 9.75 | 0.130 | 34.45 | 0.858 | 9.46 | 0.129 | 34.49 | 0.859 |
| Denoising | VP-SDE | 10.69 | 0.124 | 34.72 | 0.882 | 10.11 | 0.119 | 34.92 | 0.886 |

**Choice of starting time.**   For OT-ODE sampling and VP-ODE sampling, we observe that starting at time $t > 0$ improves the performance. We therefore perform an ablation study on ΠGDM baseline, and vary the start time to verify whether starting at $t > 0$ helps to improve the performance. We plot the metrics for three different measurements in Figure 30. We observe that starting later at time $t > 0$ consistently leads to worse performance compared to starting at time $t = 0$. Therefore, for all our experiments with ΠGDM, we always start at time $t = 0$.

Table 24: Relative performance of $\Pi$GDM on AFHQ with VE and VP derived $r_t^2$ with $\sigma_y = 0.05$

| Measurement | Model | VP | | | | VE | | | |
|---|---|---|---|---|---|---|---|---|---|
| | | FID ↓ | LPIPS ↓ | PSNR ↑ | SSIM ↑ | FID ↓ | LPIPS ↓ | PSNR ↑ | SSIM ↑ |
| SR 4× | OT | 12.69 | 0.285 | 30.18 | 0.665 | 12.31 | 0.282 | 30.23 | 0.672 |
| Gaussian deblur | OT | 24.60 | 0.383 | 28.93 | 0.429 | 19.66 | 0.355 | 29.16 | 0.475 |
| Inpainting-*Center* | OT | 19.09 | 0.153 | 34.20 | 0.855 | 16.51 | 0.145 | 34.40 | 0.863 |
| Denoising | OT | 11.20 | 0.159 | 34.49 | 0.876 | 10.92 | 0.153 | 34.78 | 0.883 |
| SR 4× | VP-SDE | 77.49 | 0.469 | 29.34 | 0.469 | 54.12 | 0.413 | 29.73 | 0.549 |
| Gaussian deblur | VP-SDE | 116.42 | 0.535 | 28.49 | 0.313 | 95.09 | 0.493 | 28.74 | 0.368 |
| Inpainting-*Center* | VP-SDE | 57.46 | 0.239 | 32.40 | 0.773 | 56.86 | 0.238 | 32.42 | 0.775 |
| Denoising | VP-SDE | 81.15 | 0.451 | 29.62 | 0.639 | 35.33 | 0.278 | 31.72 | 0.776 |

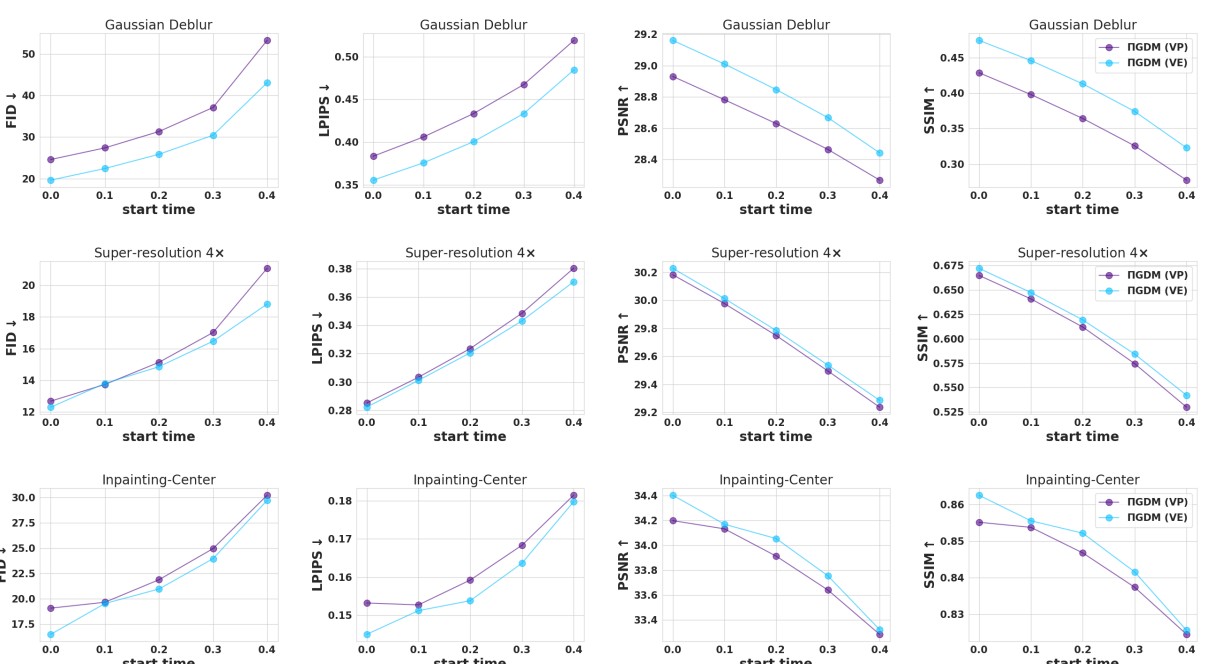

Figure 30: Variation in performance $\Pi$GDM sampling with variation in start times on AFHQ dataset. We use pretrained conditional OT model and set $\sigma_y = 0.05$. We observe similar trends with VP-SDE checkpoint. We plot metrics for both choices of $r_t^2$ that can be derived from variance preserving and variance exploding formulations.

## E.2 RED-Diff

**Implementation details.** We use VP-SDE model for all experiments with RED-Diff. We closely follow the official code available on github while implementing RED-Diff. Similar to (Mardani et al., 2023), we always use uniform spacing while iterating the timestep over 1000 steps. We use ascending time from 0 to 1. Note that the original paper uses descending time from $T$ to 0. According to the notational convention used in this paper, this is equivalent to ascending time from 0 to 1. We use Adam optimizer and use the momentum pair $(0.9, 0.99)$ similar to the original work. Further, we use initial learning rate of 0.1 for AFHQ and ImageNet-128, as used in the original work, and learning rate of 0.01 for ImageNet-64. We use batch size of 1 for all the experiments. Finally, we extensively tuned the regularization hyperparameter $\lambda$ to find the value that results in optimal performance across all metrics. We summarize the results of our experiments in Table 25 to 30. We note that more extensive tuning may be able to find better performing hyperparameters but this goes against the intent of a training-free algorithm.

Table 25: Hyperparameter search for RED-Diff on face-blurred ImageNet-$64 \times 64$ with $\sigma_y = 0.05$. We use learning rate of $0.01$.

| $\lambda$ | SR 2×, $\sigma_y = 0.05$ | | | | Gaussian deblur, $\sigma_y = 0.05$ | | | |
|---|---|---|---|---|---|---|---|---|
| | FID ↓ | LPIPS ↓ | PSNR ↑ | SSIM ↑ | FID ↓ | LPIPS ↓ | PSNR ↑ | SSIM ↑ |
| 0.1 | 34.09 | 0.224 | 30.12 | 0.798 | **46.76** | 0.254 | 29.29 | 0.715 |
| 0.25 | 28.45 | 0.206 | 30.40 | 0.814 | 51.20 | **0.236** | **30.19** | **0.776** |
| 0.75 | **23.02** | **0.187** | **31.22** | **0.839** | 73.76 | 0.287 | 30.47 | 0.750 |
| 1.5 | 32.35 | 0.243 | 30.80 | 0.792 | 82.26 | 0.335 | 30.29 | 0.705 |
| 2.0 | 40.33 | 0.284 | 30.41 | 0.750 | 86.48 | 0.358 | 30.17 | 0.683 |
| $\lambda$ | Inpainting-*Center*, $\sigma_y = 0.05$ | | | | Denoising, $\sigma_y = 0.05$ | | | |
| | FID ↓ | LPIPS ↓ | PSNR ↑ | SSIM ↑ | FID ↓ | LPIPS ↓ | PSNR ↑ | SSIM ↑ |
| 0.1 | 15.71 | 0.155 | 31.74 | 0.840 | 12.47 | 0.085 | 32.24 | 0.907 |
| 0.25 | 15.56 | 0.155 | 31.73 | 0.839 | 11.80 | 0.083 | 32.36 | 0.908 |
| 0.75 | 13.31 | 0.139 | 32.65 | 0.857 | 8.43 | 0.062 | 33.65 | 0.932 |
| 1.5 | **12.18** | **0.119** | 33.97 | 0.881 | 6.11 | 0.041 | 35.34 | 0.958 |
| 2.0 | 12.87 | **0.119** | **34.19** | **0.886** | **6.02** | **0.041** | **35.64** | **0.964** |

Table 26: Hyperparameter search for RED-Diff on face-blurred ImageNet-$64 \times 64$ with $\sigma_y = 0$. We use learning rate of $0.01$.

| $\lambda$ | SR 2×, $\sigma_y = 0$ | | | | Gaussian deblur, $\sigma_y = 0$ | | | |
|---|---|---|---|---|---|---|---|---|
| | FID ↓ | LPIPS ↓ | PSNR ↑ | SSIM ↑ | FID ↓ | LPIPS ↓ | PSNR ↑ | SSIM ↑ |
| 0.1 | **11.74** | 0.224 | 30.12 | 0.798 | **15.39** | **0.134** | **31.99** | **0.879** |
| 0.25 | 12.65 | **0.130** | **32.34** | **0.886** | 29.56 | 0.236 | 30.19 | 0.776 |
| 0.75 | 20.36 | 0.187 | 31.22 | 0.839 | 55.43 | 0.287 | 30.47 | 0.750 |
| 1.5 | 33.13 | 0.243 | 30.80 | 0.792 | 71.64 | 0.335 | 30.29 | 0.705 |
| 2.0 | 41.56 | 0.288 | 30.46 | 0.752 | 78.55 | 0.358 | 30.22 | 0.685 |

| $\lambda$ | Inpainting-*Center*, $\sigma_y = 0$ | | | |
|---|---|---|---|---|
| | FID ↓ | LPIPS ↓ | PSNR ↑ | SSIM ↑ |
| 0.1 | **7.29** | **0.079** | **39.14** | **0.925** |
| 0.25 | 7.40 | 0.155 | 31.73 | 0.839 |
| 0.75 | 8.47 | 0.083 | 38.59 | 0.922 |
| 1.5 | 10.75 | 0.095 | 37.42 | 0.916 |
| 2.0 | 12.54 | 0.119 | 34.19 | 0.886 |

Table 27: Hyperparameter search for RED-Diff on face-blurred ImageNet-$128 \times 128$ with $\sigma_y = 0.05$. We use learning rate of $0.1$.

| $\lambda$ | SR 2×, $\sigma_y = 0.05$ | | | | Gaussian deblur, $\sigma_y = 0.05$ | | | |
|---|---|---|---|---|---|---|---|---|
| | FID ↓ | LPIPS ↓ | PSNR ↑ | SSIM ↑ | FID ↓ | LPIPS ↓ | PSNR ↑ | SSIM ↑ |
| 0.1 | 23.25 | 0.272 | 30.12 | 0.731 | 37.83 | 0.42 | 28.54 | 0.473 |
| 0.75 | 14.56 | 0.224 | 30.71 | 0.782 | **21.43** | **0.229** | 31.41 | 0.807 |
| 1.5 | **10.54** | **0.182** | 31.82 | 0.852 | 22.85 | 0.247 | **31.65** | **0.809** |
| 2.0 | 11.65 | 0.187 | **31.93** | **0.859** | 24.71 | 0.259 | 31.61 | 0.802 |
| $\lambda$ | Inpainting-*Center*, $\sigma_y = 0.05$ | | | | Denoising, $\sigma_y = 0.05$ | | | |
| | FID ↓ | LPIPS ↓ | PSNR ↑ | SSIM ↑ | FID ↓ | LPIPS ↓ | PSNR ↑ | SSIM ↑ |
| 0.1 | 19.68 | 0.191 | 31.75 | 0.795 | 12.83 | 0.134 | 32.27 | 0.854 |
| 0.75 | 19.03 | 0.202 | 31.36 | 0.779 | 12.69 | 0.14 | 32.09 | 0.846 |
| 1.5 | 16.33 | 0.189 | 31.81 | 0.794 | 10.67 | 0.121 | 32.89 | 0.874 |
| 2.0 | **14.63** | **0.171** | **32.42** | **0.819** | **9.19** | **0.105** | **33.52** | **0.895** |

Table 28: Hyperparameter search for RED-Diff on face-blurred ImageNet-$128 \times 128$ with $\sigma_y = 0$. We use learning rate of $0.1$.

| $\lambda$ | SR 2×, $\sigma_y = 0$ | | | | Gaussian deblur, $\sigma_y = 0$ | | | |
|---|---|---|---|---|---|---|---|---|
| | FID ↓ | LPIPS ↓ | PSNR ↑ | SSIM ↑ | FID ↓ | LPIPS ↓ | PSNR ↑ | SSIM ↑ |
| 0.1 | **3.90** | **0.082** | **34.47** | **0.922** | **4.19** | **0.085** | **34.68** | **0.929** |
| 0.75 | 6.52 | 0.105 | 33.54 | 0.905 | 12.59 | 0.177 | 32.71 | 0.864 |
| 1.5 | 10.46 | 0.142 | 32.98 | 0.894 | 19.29 | 0.225 | 32.15 | 0.831 |
| 2.0 | 13.08 | 0.165 | 32.65 | 0.884 | 22.57 | 0.245 | 31.94 | 0.816 |
| $\lambda$ | Inpainting-*Center*, $\sigma_y = 0$ | | | | Inpainting-*Freeform*, $\sigma_y = 0$ | | | |
| | FID ↓ | LPIPS ↓ | PSNR ↑ | SSIM ↑ | FID ↓ | LPIPS ↓ | PSNR ↑ | SSIM ↑ |
| 0.1 | **5.39** | **0.068** | **38.91** | **0.928** | **8.94** | **0.162** | **35.54** | **0.830** |
| 0.75 | 5.52 | 0.073 | 38.11 | 0.924 | 9.26 | 0.166 | 35.05 | 0.826 |
| 1.5 | 6.09 | 0.079 | 37.32 | 0.920 | 10.13 | 0.172 | 34.58 | 0.821 |
| 2.0 | 6.68 | 0.083 | 36.87 | 0.917 | 10.87 | 0.176 | 34.30 | 0.818 |

Table 29: Hyperparameter search for RED-Diff on AFHQ with $\sigma_y = 0.5$. We use learning rate of 0.1.

| $\lambda$ | SR 4×, $\sigma_y = 0.05$ | | | | Gaussian deblur, $\sigma_y = 0.05$ | | | |
|---|---|---|---|---|---|---|---|---|
| | FID ↓ | LPIPS ↓ | PSNR ↑ | SSIM ↑ | FID ↓ | LPIPS ↓ | PSNR ↑ | SSIM ↑ |
| 0.1 | 21.59 | 0.385 | 29.51 | 0.607 | 17.36 | 0.379 | 29.95 | 0.639 |
| 0.25 | 22.47 | 0.374 | 29.66 | 0.635 | **15.81** | **0.341** | **30.15** | **0.645** |
| 0.75 | **20.84** | **0.331** | **29.97** | **0.675** | 25.41 | 0.366 | 29.76 | 0.588 |
| 1.5 | 22.46 | 0.355 | 29.68 | 0.642 | 38.66 | 0.409 | 29.34 | 0.525 |
| 2.0 | 25.02 | 0.376 | 29.49 | 0.618 | 45.01 | 0.427 | 29.18 | 0.500 |

| $\lambda$ | Inpainting-*Center*, $\sigma_y = 0.05$ | | | | Denoising, $\sigma_y = 0.05$ | | | |
|---|---|---|---|---|---|---|---|---|
| | FID ↓ | LPIPS ↓ | PSNR ↑ | SSIM ↑ | FID ↓ | LPIPS ↓ | PSNR ↑ | SSIM ↑ |
| 0.1 | **28.39** | 0.216 | 31.53 | 0.756 | 8.32 | 0.159 | 32.18 | 0.827 |
| 0.25 | 28.85 | 0.217 | 31.51 | 0.755 | 8.35 | 0.161 | 32.16 | 0.826 |
| 0.75 | 28.80 | 0.218 | 31.64 | 0.759 | 7.94 | 0.156 | 32.35 | 0.833 |
| 1.5 | 28.74 | 0.205 | 32.19 | 0.784 | 6.63 | 0.138 | 33.12 | 0.862 |
| 2.0 | 28.55 | 0.190 | 32.63 | 0.802 | 5.71 | 0.124 | 33.70 | 0.882 |
| 2.5 | 28.71 | **0.177** | **32.99** | **0.818** | **4.93** | **0.111** | **34.18** | **0.899** |

Table 30: Hyperparameter search for RED-Diff on AFHQ with $\sigma_y = 0$. We use learning rate (lr) of 0.1 unless mentioned otherwise.

| $\lambda$ | SR 4×, $\sigma_y = 0$ | | | | Gaussian deblur, $\sigma_y = 0$ | | | |
|---|---|---|---|---|---|---|---|---|
| | FID ↓ | LPIPS ↓ | PSNR ↑ | SSIM ↑ | FID ↓ | LPIPS ↓ | PSNR ↑ | SSIM ↑ |
| 0.005 | 11.67 | 0.197 | **32.93** | **0.837** | 14.69 | 0.278 | **31.73** | **0.760** |
| 0.05 | **8.65** | **0.191** | 32.21 | 0.801 | 11.67 | **0.268** | 31.30 | 0.731 |
| 0.1 | 9.65 | 0.204 | 31.84 | 0.781 | **11.53** | 0.273 | 31.05 | 0.711 |
| 0.25 | 11.65 | 0.222 | 31.53 | 0.768 | 13.22 | 0.293 | 30.63 | 0.675 |
| 0.75 | 14.98 | 0.274 | 30.72 | 0.726 | 23.34 | 0.351 | 29.91 | 0.598 |
| 1.5 | 19.40 | 0.332 | 29.95 | 0.665 | 36.96 | 0.402 | 29.39 | 0.529 |
| 2.0 | 22.72 | 0.361 | 29.65 | 0.632 | 43.64 | 0.422 | 29.22 | 0.504 |

| $\lambda$ | Inpainting-*Center*, $\sigma_y = 0$, lr=0.01 | | | | Inpainting-*Freeform*, $\sigma_y = 0$ | | | |
|---|---|---|---|---|---|---|---|---|
| | FID ↓ | LPIPS ↓ | PSNR ↑ | SSIM ↑ | FID ↓ | LPIPS ↓ | PSNR ↑ | SSIM ↑ |
| 0.005 | **8.53** | **0.050** | **38.89** | **0.951** | 7.22 | 0.091 | **36.89** | **0.892** |
| 0.05 | 8.53 | 0.050 | 38.89 | 0.951 | 7.27 | **0.090** | 36.88 | **0.892** |
| 0.1 | 8.53 | 0.050 | 38.88 | 0.951 | **7.23** | 0.091 | 36.82 | 0.891 |
| 0.25 | 8.53 | 0.050 | 38.83 | 0.950 | 7.32 | 0.094 | 36.69 | 0.889 |
| 0.75 | 8.88 | 0.056 | 38.60 | 0.948 | 7.74 | 0.102 | 36.26 | 0.884 |
| 1.5 | 10.32 | 0.071 | 38.04 | 0.942 | 8.41 | 0.112 | 35.69 | 0.877 |
| 2.0 | 11.62 | 0.084 | 37.54 | 0.937 | 8.76 | 0.119 | 35.37 | 0.872 |

# F  Additional Background

In this section, we follow the notation used in the prior work by Lipman et al. (2022).

**Continuous Normalizing Flows (CNFs).**  A Continuous Normalizing Flow (Chen et al., 2018a) is a time-dependent diffeomorphic map $\phi_t : [0, 1] \times \mathbb{R}^d \rightarrow \mathbb{R}^d$ that is defined by the ODE:

$$\frac{d}{dt}\phi_t(\boldsymbol{x}) = \boldsymbol{v}_t(\phi_t(\boldsymbol{x})); \quad \phi_0(\boldsymbol{x}) = \boldsymbol{x} \tag{20}$$

where $\boldsymbol{x} \in \mathbb{R}^d$ and $\boldsymbol{v}_t : [0, 1] \times \mathbb{R}^d \rightarrow \mathbb{R}^d$ is a time-dependent vector field that is usually parametrized with a neural network. The generative process of a CNF involves sampling from a simple prior distribution $\boldsymbol{x}_0 \sim p_0(\boldsymbol{x}_0)$ (e.g. standard Gaussian distribution) and then solving the initial value problem defined by the ODE in Eq. (20) to obtain a sample from the target distribution $\boldsymbol{x}_1 \sim p_1(\boldsymbol{x}_1)$. Thus, a CNF reshapes a simple prior distribution $p_0$ to a more complex distribution $p_t$, via a push-forward equation based on the instantaneous change of variables formula.

$$p_t = [\phi]_* p_0 \tag{21}$$

$$[\phi]_* p_0(\boldsymbol{x}) = p_0(\phi_t^{-1}(\boldsymbol{x}))\det\left[\frac{\partial \phi_t^{-1}}{\partial \boldsymbol{x}}(\boldsymbol{x})\right] \tag{22}$$

CNFs are usually trained by optimizing the maximum likelihood objective. As shown in Chen et al. (2018a), the exact likelihood computation can be done via relatively cheap operations despite the Jacobian term. However, this requires restricting the architecture of the neural network to constrain the Jacobian term. FFJORD (Grathwohl et al., 2018) improves upon this by proposing a method that uses Hutchinson's trace estimator to compute log density, and allows CNFs with free-form Jacobians, thereby removing any restrictions on the architecture. This approach has difficulties for high-dimensional images where the trace estimator is noisy. Flow Matching provides an alternative, scalable approach to training CNFs with arbitrary architectures.

**Flow Matching.**  Suppose we have samples from an unknown data distribution $\boldsymbol{x}_1 \sim q(\boldsymbol{x}_1)$. Let $p_t$ denote a probability path from the prior distribution $p_0$ to the data distribution $p_1$ that is approximately equal to $q$. Flow Matching loss is defined as

$$\mathcal{L}_{FM} = \mathbb{E}_{t,p_t(\boldsymbol{x})}\|\boldsymbol{v}_t(\boldsymbol{x};\theta) - \boldsymbol{u}_t(\boldsymbol{x})\|^2 \tag{23}$$

where $\boldsymbol{u}_t(\boldsymbol{x})$ is a vector field that generates the probability path $p_t(\boldsymbol{x})$, and $\theta$ denotes trainable parameters of the CNF. In practice, we usually do not have any prior knowledge on $p_t$ and $u_t$, and thus this objective is intractable. Inspired by diffusion models, Lipman et al. (2022) propose Conditional Flow Matching, where both the probability paths and the vector fields are conditioned on the sample $\boldsymbol{x}_1 \sim q(\boldsymbol{x}_1)$. The exact objective for Conditional Flow matching is given by

$$\mathcal{L}_{CFM} = \mathbb{E}_{t,q(\boldsymbol{x}_1),p_t(\boldsymbol{x}|\boldsymbol{x}_1)}\|\boldsymbol{v}_t(\boldsymbol{x};\theta) - \boldsymbol{u}_t(\boldsymbol{x}|\boldsymbol{x}_1)\|^2 \tag{24}$$

where, $p_t(\boldsymbol{x}|\boldsymbol{x}_1)$ denotes a conditional probability path, and $\boldsymbol{u}_t(\boldsymbol{x}|\boldsymbol{x}_1)$ denotes the corresponding conditional vector field that generates the conditional probability path. Interestingly, both the loss objectives in Eq. (24) and Eq. (23) have identical gradients w.r.t. $\theta$. More importantly, past research has proven that $\boldsymbol{u}_t(\boldsymbol{x}) = \mathbb{E}[\boldsymbol{u}_t(\boldsymbol{x}|\boldsymbol{x}_1)|\boldsymbol{x}_t = \boldsymbol{x}]$. The optimal solution to the conditional Flow Matching recovers $\boldsymbol{u}_t(\boldsymbol{x})$ and therefore $\boldsymbol{v}_t(\boldsymbol{x};\theta)$ generates the desired probability path $p_t(\boldsymbol{x})$.Thus, we can train a CNF without access to the marginal vector field $\boldsymbol{u}_t(\boldsymbol{x})$ or probability path $p_t(\boldsymbol{x})$. Compared to the prior approaches to train flow models, Flow Matching allows simulation-free training with unbiased gradients, and scales easily to high dimensions.

# G   Additional Related Work

Inverse problems are ubiquitous in various domains like image processing (Krishnan & Fergus, 2009; Rick Chang et al., 2017; Gilton et al., 2019; Bertalmio et al., 2000; Yang et al., 2010), medical imaging (Ribes & Schmitt, 2008; Jin et al., 2017; Liang et al., 2020; Song et al., 2021c), and remote sensing (Krasnopolsky, 2009; Krasnopolsky & Schiller, 2003; Dong et al., 2018). Many approaches have been developed over years to solve inverse problems. Variational methods (Agrawal et al., 2022) formulate the inverse problem as an optimization task with a regularization term (Benning & Burger, 2018) for certain desirable properties in the solution. Some of the well-known frameworks in this category include plug-and-play prior ($P^3$) (Venkatakrishnan et al., 2013; Chan et al., 2016; Kamilov et al., 2017; Meinhardt et al., 2017; Zhang et al., 2017; Vidal et al., 2020), deep image prior (DIP) (Ulyanov et al., 2018; Van Veen et al., 2018), and regularization by denoising (RED) (Romano et al., 2017; Cohen et al., 2021). Subsequent works, inspired by these prior works, have extended these frameworks to include flow models (Whang et al., 2021a;b), optimal transport (Vidal et al., 2020), and more recently, diffusion models (Mardani et al., 2023; Graikos et al., 2022; Liu et al., 2023b) as prior. Optimization-based inversion methods were also extended to include GANs (Bora et al., 2017; Shah & Hegde, 2018; Raj et al., 2019; Daras et al., 2021; Pan et al., 2021). Optimization-based approaches for solving inverse problems, despite their widespread popularity, have certain drawbacks. These methods are often computationally expensive as they involve optimizing an objective, which might require many steps to converge to a solution. Further, designing the optimization objective itself can be challenging. In addition, these methods are sensitive to the choice of hyperparameters like regularization parameter, as noted in our experiments with RED-Diff (Mardani et al., 2023).

With emergence of diffusion models, another family of gradient-based approaches for inverse problems have emerged. These approaches do not explicitly optimize an objective, *i.e.,* they are training-free, but they use gradients to guide the sampling process with diffusion model as prior. These approaches usually involve iterative denoising through a SDE and gradient-based correction that is applied at each step of the process. Some of the approaches in this category include Diffusion Posterior Sampling (DPS) (Chung et al., 2022a), Manifold Constraint Gradient (MCG) (Chung et al., 2022b), ΠGDM (Song et al., 2022), and shortcut sampling for diffusion (SSD) (Liu et al., 2023a). Our proposed approach for solving linear inverse problems with flow models also falls under this category. Computing gradients at each step of denoising can be expensive. There are many gradient-free iterative methods for inversion that utilize diffusion models as generative prior. Some prominent approaches in this category are denoising diffusion restoration models (DDRM) (Kawar et al., 2022) and denoising diffusion null-space model (DDNM) (Wang et al., 2022). We have covered important distinctions between these approaches briefly in Sec. 5 of the main paper.

The aforementioned approaches for solving inverse problems with diffusion models use pre-trained diffusion models and are not specific to a particular measurement operator. There are works such as (Saharia et al., 2021; 2022a) that train a conditional diffusion model to solve a specific inverse problem. This approach for solving inverse problems is more computationally expensive as it involves training a model from scratch. Further, the resulting model is specific to the measurement operator used in the training data and cannot be reused to solve inverse problems with a different measurement operator.In addition to the above, there is also a line of research that considers the more general setting of blind inverse problem where the method to solve an inverse problem is agnostic to the measurement operator. Some works that have advanced this line of research are Chung et al. (2023b); Gan et al. (2024); Laroche et al. (2024). Finally, we note that there are previously proposed methods such as Whang et al. (2021a;b); Hong et al. (2023) which solve inverse problems using CNFs. As noted in these prior works, using CNFs for solving inverse problems presents computational challenges as well as challenges due to restricted architecture. In this work, we consider CNFs that are trained with flow matching (or similarly converted diffusion models) which are more computationally more efficient and do not suffer from drawbacks observed due to restricted architectures.

