# OpenReview forum: "Training-free linear image inverses via flows"
_TMLR — Accepted by TMLR_

### Review · Reviewer_GG8v · 2024-03-13

**Summary Of Contributions:**

The paper proposes to couple a fixed linear guidance to pre-trained diffusion/flow models for the purpose of image cleaning. I think the main contributions are turning unguided models into guided ones, and some ODE correction. I found the contributions to be vague, so I'm not totally sure.

**Audience:**

Yes

**Broader Impact Concerns:**

No issues.

**Claims And Evidence:**

Yes

**Requested Changes:**

Post-response update: The authors have addressed all of my comments, and made extensive improvements to the paper. I have no further comments, and would be happy to see this paper get accepted.

-------

~~To be accepted, the paper needs to clarify its contributions or claims, and pair them with empirical evidence, and clarify why the stuff that is happening is novel, significant or interesting for the community. Currently (i) I’m a bit confused of what are the contributions; (ii) I believe the correction and unguided-to-guided stuff is not backed by empirical evidence; and (iii) I’m not yet convinced that the paper’s material is interesting enough for the community, given that there is so much stuff that seems just common knowledge. Finally, lots of presentation clarifications are needed. I’m looking forward to author clarifications.~~

**Below comments are outdated after revisions**

- Before eq 1 it was stated that p_theta is noise-to-data, while in eq 1 q is noise-to-data. This is confusing.
- In eq1 we integrate over q. What does this mean? What is q? What do we integrate over? What variables are free and what are fixed? The denoiser is deterministic (I assume, not sure, is it?) so the expectation can only be over x_t. Ok, but this then can mean two different things. Either we keep x_t fixed and integrate over x_1; or we keep x_1 fixed and integrate over different x_t’s. Or maybe we integrate over both x_t and x_1 (which means in reality integrating over eps and x_1). Also the role of q is confusing, since it can be either noise-to-data or data-to-noise. Can you explain what is q, and how should we interpret it, and what are its inputs?
- Can you also explain what is p_theta, what are its inputs, and why its not present in almost any of the equations? Do we assume some kind of one-way diffusion models similar to GAN? Or is the denoiser hat{x} representing the p_theta?
- Can you also comment on the nature of xhat: is it deterministic or stochastic?
- The eq 1 continues by saying that we sample x0 (noise) from the data-to-noise process p. Not sure what this means. So do we start from a clean image, and turn it into pure noise? But why do we need to do this, why not just do pure noise right away? I think the two endpoint densities (pure noise, pure data) are special cases that need clearer treatment in the notation.
- Or maybe eq 1 is trying to imply that there is an SDE hidden inside xhat, and maybe q refers to that?
- In eq 2 I’m again confused what is q, or what variables it covers or touches. Does it give both xt and x1, or only xt?
- In eq 2 the LHS has only v and q, but RHS has x_1, y, x_t. This then implies that the variables (x_1, y, x_t) are all coming from somewhere, and the only thing that can give them is the q. But I’m not sure if this is true. I suspect that the notation is incorrect, and we should add some conditioning to the LHS by saying L_cfm(v, q, x1, y) or something like that. Can you explain what is the nature of (x_1, x_t, y) and where do they come from, and which of them are fixed and which are variable?
- What are the “marginal” distributions marginalised over? I assume we marginalise x1 and epsilon, is this true?
- What’s the difference between p(x|y) and q(x|y)? Somehow the presentation is difficult to follow since the notation is flipped wrt usual style, and lots of stuff is undefined. I would guess that q(x|y) represents the noising distribution marginalised over epsilon and x1, and p(x|y) represents denoising distribution marginalised over x0? Can you clarify?
- Before eq 4 we finally see that q is the Gaussian noising process, but earlier it was stated that p_theta is the noising process. It is also implied that q is mainly about x_t, which means that the E_q’s in previous equations are not integrating over the data distribution. Are these single-datapoint losses? What is the “q” in L_cfm(v,q)?
- In eq 4 the ds/dt(x-mu/s) looks like a function evaluation, but it also might be just a product. Can you clarify?
- I don’t understand what E[ x1 | xt ] means before eq 6. What is the q and expectation over? I think this is an average denoising given different xt’s, but that would then assume that we have some x1 reference that produces the xt’s. This is missing here. Or maybe this refers to an expectation over x1, which would be all possible denoising from a single fixed xt. But x1 usually means data, so maybe its not that. x1 can mean either denoising, or data. Or maybe the expectation is over the dataset, such that we look at all images, and see how likely each image is to produce a single fixed xt. But this also makes no sense: it does not matter how we arrived at xt. If we believe the eq 5, then this opens up as $E_q[x_1 | x_t] = \int q(x_t | x_1, y) p(x_1 | x_t) dx_t$, but this is circular since it has x1 twice. Can you help me understand?
- It is stated that “training-free approaches” aim to approximate E[x1|xt,y] from E[x1|xt]. Err… what? So training-free refers to adding the class/guidance conditionals. Is this really what you mean?
- I don’t understand what probability paths mean. Is it an infinite sequence ( q_t(x_t|..))_t=0..1 or something like that? What is the path over? Is it a path of images x_t or densities q_t?
- In eq 6 it seems that we obtain guided denoising by taking an unguided denoising, and adding a classifier gradient. Not sure if this makes sense. So we add the classifier shift only once? Isn’t this strange? Don’t we want to add the shift along the ODE or SDE?
- I’m also confused what E_q[ x_1 | x_t] means. I don’t know what is q over, or what are we integrating over. I would guess that x_t is fixed as the conditioning variable, but then it means that we are integrating over x_1’s? Not sure what this means either. I’m also confused if x_1 means observed datapoints, or perhaps denoisings. I think the only sensible interpretation is that this is the expected denoising given a single noisy image x_t. But the denoiser is usually deterministic, so I guess this becomes Dirac. That doesn’t feel sensible. Another option is that we take an expectation over x_t’s, but which ones? Are these all x_t’s from one source image x_1, or maybe all x_t’s that could ever emerge from any image? I can’t find any interpretation that makes sense.
- In eq 7 I don’t understand what measurements means. Measurement of what? Do we measure something from a sample x ~ q(x_1)? Or is this some general property vector we wish to fulfil? Do we actually go and measure something in the real world? I think the y was supposed to be a noisy image (eg. blur), so I wonder why is this called measurement. Can you help me understand the setting?
- The density q(x1|y) implies that a joint distribution q(x1,y) exists. Can you elaborate on what would this represent? Can we define this joint? It would help a lot to give comprehensive model definitions.
- I don’t understand the density q(x1|y). It looks like it represent the probability of all images assuming they are class y. But earlier we had a definition that q(x|..) is a simple Gaussian noising process. How does this then imply a density over clean images? Are the q in eq 5 and q in eq 7 different q’s? In sec 3.1. we are introducing even more q’s: Now we have q’ and q (not sure what these are). Also, is x1 an observed image from the dataset? Or is it a denoised image? Or just any image? Do we observe (x1,y) pair?
- Overall it would be very useful to write the joint densities and their conditional out in full, and describe what they mean, and also remark how many different densities there are.
- Lemma 1 is missing context. “optimal solution is known”, but wrt what? Of what? Are we talking about cfm loss eq 2? What does it mean that solution is “known”? Does it have a closed form solution, or does the solution just generally exist? This is too vague, and the assumptions and model setting has not been described precisely enough.
- What is \alpha_t and \alpha_t’ in eq 9? So we have two alphas? Err… what? I’m totally lost. Are we changing the schedule for some reason so that we have one diffusion model with two schedules (whatever that means..)? Do we train two diffusion models with two different schedules? Why would we do that? Or are these the alpha value at two different timepoints as \alpha_t and \alpha_{t’}? I’m struggling to decrypt the paper.
- What is X’_t’(t)? The notation is getting quite complicated. I think this is just showing that we can “stretch” xt from one schedule to equivalent xt on a slower/faster schedule. Is this true? But why do we care about this stuff in the first place? The section 3.1. seems to end without explaining why we did this stuff. I think the motivation could be that we want to use a “slow” schedule diffusion model, and sample it using a “fast” schedule, but I can’t see any usecases for this.
- In eq 6 we move from xt to x1, and add the guidance correction term once. This implies that the time is somehow discretised into windows, and each has its own single correction. For example, if we go from x0 to xt, and then from xt to x1, we apply guidance twice. The eq 11 is supposed to be a version of eq 6, but in eq 11 we have the correction as part of the ODE time derivative, which means that we apply the correction infinitely many times. Do you see a discrepancy in this, and can you explain it?
- Is theorem 1 novel?
- What is q^app? Where does this come from? Why did we ditch the q(y|xt)? I can’t follow this in eq 12. I also can’t follow why we add the gamma. So we added the q^app and then add more stuff to fix the errors that q^app brings. Err… why not just use the original q(y|xt)?
- So it seems that in eq 12 we use a linear-Gaussian regression from x1 to y. Why is this a good idea? I assume x1 refers to images and its pixels, and I guess y does the same (maybe?).   Surely a linear function is extremely bad at approximating any useful “measurements” of the image? Can you comment on this?
- Why do we choose the N( xhat, r) approximation for q(x1|xt)? This again looks bad: we make a clean prediction and add gaussian noise around it. Surely this results in either unnecessarily noisy denoisings, or it’s not able to represent any useful uncertainty in the denoising. I’m not really following why we are doing any of this stuff.
- The text says that q(x1) can be N(0,i). Err… wasn’t x1 supposed to be clean data? This is super confusing. Why would we ever assume data is pure noise?
- In eq 14 the y now needs to be same size as x. I think this is confusing. The paper started by defining y as noisy image, then in the middle y was ambiguous, and now y is again noisy image. It would be cleaner to keep y same through the paper.
- Can you describe the novelties of this work? Is this work just adding adaptive weighting to PiGDM?
- In alg 1 I don’t think the rows 6 and 7 have been explained by the paper. Alg 1 seems to use z while rest of the paper uses x.
- In experiments it seems that you use two conditional generative models. But wasn’t the whole point of the paper that you take unconditional models, and turn them to conditional ones by your method? What are you trying to demonstrate in the experiments?
- How do you get the A matrix? What does the A matrix represent or mean? Is the A-matrix dependent on the corruption type or just some global matrix? Do you train these somehow?
- How did you choose the adaptive weights? Based on what?
- In the end it seems that you method is taking a diffusion model, and following denoiser ODE while being also pushed by a linear regression towards y. Is this accurate? How come this hasn’t been done already, or has it? Isn’t this just standard classifier-guidance?
- I also frankly did not even understand what the “correction” is in this paper. Is it the q^app? Or gamma? Or maybe just eq 11? What do we “correct”? If we don’t correct, is there some error somewhere?

**Strengths And Weaknesses:**

Post-response update: After the discussion period and author revisions the paper is currently excellently written, provides a good contribution in linking flow matching and diffusion models, and provides a strong contribution in solving inverse problems using existing networks without further pre-training. This is a good paper that I'd be happy to see accepted.

------

S/W: The results show that the method works, and gives good performance. However, there are only 2 competing methods. This is not sufficient: there are tons of guided diffusion methods, and even a simple paper needs to have a wider comparison.

~~W: I found the paper to be poorly written with frustratingly ambiguous math. I think I understood the main ideas, but I’m not totally sure. The contributions and novelties are a bit unclear, and the paper claims contributions that to me seem trivial. For example, the second contribution bullet claims that paper offers a way to convert between diffusion and flow models (by just following the ODE part). This does not seem novel.~~

~~W: I think a main contribution is the PiGDM correction, but there is no ablation that shows if it does anything. This needs to be added: it’s one of the main claims, and currently it lacks evidence. Similarly the unguided-to-guided conversion seems to be a main contribution, but I can’t see any experiments about it. (Maybe these are in the paper, I’m not sure)~~

---

> ### Author Response · Authors · 2024-04-15
>
> We would like to thank the reviewer for carefully reading the paper and for their many detailed questions. We have tried our best to answer your key concerns and questions below. We have also incorporated some of your suggestions and have updated the text (highlighted in green) to improve clarity.
>
> **Key contributions**
> We identify that there is a gap in the literature for solving inverse problems via flow models. Flow models trained via flow matching naturally consider a broader range of probability paths than diffusion models, and therefore, leveraging them opens the possibility to use optimal transport paths for solving inverse problems.
> - Our primary methodological contribution is Theorem 1, which provides the mathematical expression to adapt the vector field $\hat{v}(x)$ of unconditional flow models with affine Gaussian probability paths. This allows us to retrieve the conditional vector field $\hat{v}(x,y)$ that can sample from the correct conditional posterior distribution $q(x_t|y)$.
> - We utilize Theorem 1 to propose an algorithm (See Algorithm 3) for solving linear inverse problems via flows with arbitrary affine Gaussian probability paths. We utilize $\Pi$GDM adaptation to estimate the intractable term $\nabla_{x_t} \log q(y | x_t)$ that arises in our expression to adapt the unconditional vector field.
> - We empirically demonstrate that conditional OT paths outperform diffusion probability paths in solving noisy linear inverse problems on a broad range of datasets (ImageNet 64/128 and AFHQ 256) and tasks (Gaussian deblurring, super-resolution, denoising and inpainting). For the noiseless setting, we find that our method performs comparably to diffusion probability paths on tasks like Gaussian deblurring and super-resolution, and slightly underperforms for inpainting.
> - As pretrained flow models are not widely available, we offer a way to convert between flow models and diffusion models, including crucially between arbitrary Gaussian probability paths (Lemma 2).  This allows applying our OT-ODE method to diffusion models trained under any affine Gaussian probability path, as demonstrated by our results using our VP-SDE model checkpoint.  Algorithm 1 in the main paper outlines our method for solving linear inverse problems via conditional OT probability paths with a pretrained diffusion model. Additionally, Algorithm 2, found in the supplementary, enables use of pretrained diffusion models for solving linear inverse problems via flows with any affine Gaussian probability paths.
>
> This work opens up avenues for using flow matching models for solving inverse problems which hasn’t been explored before.
>
> __What is the “correction” in our proposed method?__:  Given pretrained flow models or diffusion models, our method adapts the vector field of unconditional flow model such that the resulting “corrected” vector field can be used for conditional sampling, specifically for solving linear inverse problems and obtaining the clean image $x_1$. In the latest version of our draft, we have updated the term “correction” to “adaptation”.
>
> __The correction and unguided-to-guided stuff is not backed by empirical evidence__:
> Our experimental set up for solving linear inverse problems is similar to that used in the prior works such as DDRM [1],  $\Pi$GDM [2], Diffusion Posterior Sampling [3], RED-Diff [4]  etc. We present empirical results on three datasets with varying resolutions – (ImageNet 64/128 and AFHQ 256)-- and four different tasks: Gaussian deblurring, super-resolution, denoising and inpainting.  We have reported results using four different metrics – FID, LPIPS, SSIM and PSNR. Further, we provide results for both noisy and noiseless inverse problems, and have included additional ablation studies in the appendix. We also extensively tuned the hyperparameters for baselines  and have included these results in Appendix F. Given our problem setup, we do believe that we have provided sufficient empirical evidence to demonstrate the advantages as well as disadvantages of our proposed method.
>
> **Justifying the choice of baselines**:
> We have compared against $\Pi$GDM [1] and RED-Diff [3] which are two state-of-the-art methods for solving inverse problems with diffusion models. Diffusion Posterior Sampling (DPS) [2] is also closely related to $\Pi$GDM, however, from the previous results reported in $\Pi$GDM (See Table 9,10 and 11 in [1] ), we know that $\Pi$GDM outperforms DPS in the regime of few NFEs. Therefore, we prefer $\Pi$GDM over DPS.
>
> [1] Song, Jiaming, et al. "Pseudoinverse-guided diffusion models for inverse problems." International Conference on Learning Representations. 2022.
>
> [2] Chung, Hyungjin, et al. "Diffusion posterior sampling for general noisy inverse problems." arXiv preprint arXiv:2209.14687 (2022).
>
> [3] Mardani, Morteza, et al. "A variational perspective on solving inverse problems with diffusion models." arXiv preprint arXiv:2305.04391 (2023).

---

> ### Author Response · Authors · 2024-04-15
>
> **Distinction from classifier guidance**:
>
> While both solving inverse problems and classifier guidance involve introducing an additional gradient term $\nabla_{x_t} \log q(y | x_t)$ to guide the sampling process either towards a clean image, or towards a specific class, there are multiple differences between two approaches:
> - **Different goals and conditioning**: Classifier guidance (Dhariwal & Nichol 2021) involves guiding the samples towards a specific class to improve the sample quality. The conditioning $y$ is class labels. On the other hand, our approach conditions on measurements $y$ (such as blurry image, low resolution image etc.) and our goal is to retrieve a clean image $x_1$ given its noisy measurement.
> - **Different methodology**: Classifier guidance as proposed by Dhariwal & Nichol 2021  involves training an additional classifier. Further, this classifier is trained on noisy data, specific to one classification problem, and it is not possible to plug in pretrained classifiers. On the other hand, our method uses Theorem 1 to correct the vector field of unconditional flow models (conditioning is on noisy measurements $y$). We do not train or fine-tune any problem-specific model and hence our method is completely ‘training-free’.
>
> Following similar lines of reasoning, our method is also different from classifier-free guidance (Ho & Salimans 2022), as these methods have different goals and the methodology of guidance is different as our method does not involve any training. Classifier-free guidance involves jointly training a conditional and an unconditional diffusion model, and combining the resulting conditional and unconditional score estimates.
>
> **Problem Setup and Terminology**:
> We follow the standard setup used in prior works (DDRM [1],  $\Pi$GDM [2], Diffusion Posterior Sampling [3], RED-Diff [4]) for solving linear inverse problems. “Measurement” is a standard terminology used in prior work and denotes the distorted observations. As stated in equation 7, these measurements are generated as $y = Ax_1 + \epsilon$, where $A$ is known measurement matrix, $x_1$ is a clean image drawn from an unknown data distribution, and $\epsilon \sim N(0, \sigma^2_yI)$ is an unknown i.i.d. Gaussian noise. An example of $A$ would be a binary mask for inpainting.
>
> [1] Kawar, Bahjat, et al. "Denoising diffusion restoration models." Advances in Neural Information Processing Systems 35 (2022): 23593-23606.
>
> [2] Song, Jiaming, et al. "Pseudoinverse-guided diffusion models for inverse problems." International Conference on Learning Representations. 2022.
>
> [3] Chung, Hyungjin, et al. "Diffusion posterior sampling for general noisy inverse problems." arXiv preprint arXiv:2209.14687 (2022).
>
> [4] Mardani, Morteza, et al. "A variational perspective on solving inverse problems with diffusion models." arXiv preprint arXiv:2305.04391 (2023).
>
> **Implementation details and use of class-conditional models**:
>
> We would like to point out that our method conditions on the measurements $y = Ax_1 + \epsilon$ and our pretrained models are **not trained** on these measurements. Our experiments measure the ability of our method that uses pretrained models (that are not conditioned on these measurements such as blurry image, low resolution image, etc.) to correctly solve linear inverse problems and retrieve the clean image $x_1$. We also reiterate that we are not doing classifier guidance as described above.
>
> We believe that the standard of evaluating unconditional models does not correspond well to practice. Image generative models are often conditioned on text as well as other inputs. Our empirical results, by using class-conditional models, are therefore closer to practice which we believe increases their relevance. We did explore unconditional models at lower resolutions, and found similar trends in the results. Finally, all the methods in our paper used exactly the same model checkpoint, and therefore, there was no unfairness.
>
> **Novelty of Theorem 1**: Prior works have explored use of diffusion models for solving inverse problems. Theorem 1 specifically provides the closed-form relationship between the conditional vector field and the unconditional vector field along with a vector field adaptation term, and we leverage this equation to adapt the vector field of unconditional flow models for solving inverse problems in practice. To the best of our knowledge, we are the first to propose a method to solve linear inverse problems with unconditional flow models.

---

> ### Author Response · Authors · 2024-04-15
>
> **Clarification about Notation and Definitions**:
>
> __Forward and backward process in conditional diffusion models__: We use $q$ to denote the forward process i.e. data-to-noise process and $p_\theta$ to denote noise-to-data process. We assume that data is drawn from an unknown distribution $q(x_1, y)$. Given some conditioning $y$, conditional diffusion models have latent variables $x_{0: 1} =  \\{ x_t| t \in [0, 1)\\}$ that model the joint distribution $p_\theta(x_{0:1}, x_1 | y)$. In discrete time, we can write the joint distribution as $p_\theta(x_0, …, x_1 |y) = p_\theta(x_0 |y) \prod_{t} p_\theta(x_{t + \Delta} | x_{t}, y)$ where $\Delta$ denotes an appropriate step size of time discretization in $[0, 1]$.
>
> The forward process or diffusion process approximates the posterior $q(x_{0:1} | x_1, y)$ and is a Markov chain that adds Gaussian noise to data and in continuous time satisfies a stochastic differential equation (SDE). The parameters of $p_\theta$ are learned via minimizing the regression loss as specified in Eq (1) which is derived from the variational bound on negative log likelihood. Once trained, for a given condition $y$, samples can be generated by first sampling from $x_0 \sim p_{\theta}(x_0 | y)$ and integrating the SDE to $t=1$.
>
> Note that Eq (1) uses the forward process i.e. data-to-noise process, and we reiterate that $q$ in Eq (1) indeed refers to data-to-noise process. We have updated text (highlighted in green) so that these definitions are more clear.
>
> __Clarification about Eq 1__ :  Equation 1 is the standard loss used to train denoising diffusion models derived from the variational lower bound on negative loss likelihood. In particular, the $\widehat{x_1}$ in Eq 1 is a denoiser with parameters $\theta$ that predicts the clean image $x_1$ from noisy input image $x_t$. Previous works [1] have shown the formulation of loss in Eq (1) to be equivalent to the formulation where the denoiser instead predicts noise in input $x_t$.
> In equation 1, we are integrating over time $t$ from [0, 1] as we want to compute expectation with respect to $t$. Note that $t$ is uniformly drawn from [0, 1], and thus the probability density function $p(t) = 1$.
>
> [1] Salimans, Tim, and Jonathan Ho. "Progressive distillation for fast sampling of diffusion models." arXiv preprint arXiv:2202.00512 (2022).
>
> __Nature of denoiser $\widehat{x}$__: The denoiser $\widehat{x}$ is usually parametrized with a neural network and is trained via Equation 1. Once trained, this denoiser predicts clean image $x_1$ from noisy input $x_t$. Thus, for a given input $x_t$ and time $t$, the denoiser will indeed deterministically predict the clean image for a fixed neural network architecture. Finally, from statistical perspective, the optimal estimator  $\widehat{x}$ that minimizes the squared $L_2$ error in Eq 1 is given by $\widehat{x_1} = \mathbb{E}_{x_t \sim q(x_1|x_t, y)}[ x_1 | x_t, y]$.
>
> **Why start sampling from $p_\theta(x_0 | y)$**?: For a given conditioning y, this is indeed the right prior to start sampling. For instance, the reverse process of  discrete time conditional diffusion models is given by $p_\theta(x_0, x_\Delta, …, x_{1 - \Delta}, x_1| y)  = p_\theta(x_0 | y) \prod_t p_\theta(x_{t + \Delta} | x_t, y)$ where $\Delta$ is an appropriate step size of time discretization in $[0, 1]$. Note that $p_\theta(x_0 | y)$  is usually assumed to be pure Gaussian $\mathcal{N}(0, I)$.  Depending on the probability path, this can deviate from the optimal choice of matching $q(x_0 | y)$.  For our method’s choice of conditional OT paths, this initialization does match.
>
> __Clarification about Eq 2__: We have updated Eq. 2 to improve clarity. In conditional flow matching loss (Eq 2), $q$ refers to data-to-noise process, specifically, it refers to the conditional probability path $q(x_t | x_1, y)$ and density $q(x_1, y)$. The conditional flow matching loss is given by
>
> $L_{\text{cfm}}(\widehat{v}, q) = \int_{0}^{1} \mathbb{E}_{x_1, y, q(x_t | x_1, y)} \left[ \| \widehat{v}(x_t, y) -  v(x_t, y, x_1) \|_2^2 \right] dt$
>
> where $x_1, y \sim q(x_1, y)$ and  $y$ is given conditioning.
>
> Finally, conditional probability paths $q(x_t | x_1, y)$ are indeed defined on a per-data sample basis, in such a way that for some $x_1, y \sim q(x_1, y)$,  $q(x_t | x_1, y) = q(x_0| y) = \mathcal{N}(0, I)$ at $t=0$ and $q(x_t | x_1, y) = \mathcal{N}(x_1,\sigma_{\text{min}}^2I)$ where the mean is given by $x_1$ and $\sigma_{\text{min}}$ is sufficiently small standard deviation. In the latter case for $t=1$, it is a distribution centered around $x_1$.
>
> __Clarification about Eq 5 (Eq 4 in old version)__: The term $\dfrac{d\sigma_t}{dt}$ is separate and $\dfrac{x_t - \alpha_t x_1}{\sigma_t}$ is a product and not input to $\sigma_t$.

---

> ### Author Response · Authors · 2024-04-15
>
> **Difference between $p_\theta$ and $q$ in context of conditional flow matching**: $p_\theta$ refers to noise to data process that is used to generate samples by integrating the flow ODE defined in Eq. (3). Specifically, $p_\theta(x_t | y)$ refers to the probability path generated by the parameterized vector field $\widehat{v}(x_t, y)$.  On the other hand, as defined above, $q$ refers to data-to-noise process, specifically, it uses the  conditional probability path $q(x_t | x_1, y)$ which itself utilizes the known conditional vector field v(x_t, y, x_1) in Equation 5.
>
> **What are marginal distributions marginalized over?** : We follow the terminology and definition used in prior work by Lipman et al. 2022. Here, $p_\theta(x_t | y) = \int_{x_1} p_\theta(x_t | x_1, y) q(x_1 | y)$. Thus these distributions are indeed marginalized over $x_1$.
>
> **Questions about expectation $E_q(x_1 | x_t)$ in equation 6**: Following the notation used in previous parts, $q$ in this equation refers to data to noise process. In context of unconditional diffusion models, this expectation evaluates to $E_{x_1 \sim q(x_1 | x_t)}[x_1 | x_t] = \int_{x_1} x_1 \dfrac{ q(x_t | x_1) q(x_1)}{q(x_t)} dx_1$ and there is no circular definition.  The conditioning x_t is fixed.
>
> **”Training-free approaches”**: By training-free, we mean that our method does not require any re-training / fine-tuning of the pretrained flow model or diffusion model on specific inverse problems. Our method indeed uses a pretrained unconditional flow model / diffusion model which is trained to approximate $\mathbb{E}_q[x_1 | x_t]$ and adapts it to approximate the conditional expectation $\mathbb{E}_q[x_1 | x_t, y]$ without any fine-tuning of the pretrained unconditional model.
>
> **What is a probability path?**:  In this work, following the terminology used in Lipman, Yaron, et al 2022, we refer to $q(x_t | x_1)$ as probability paths and  $q(x_t | x_1, y)$ as conditional probability paths. These are indeed probability densities defined over all $x_t, t \in [0, 1]$.
>
> Lipman, Yaron, et al. "Flow matching for generative modeling." arXiv preprint arXiv:2210.02747 (2022).
>
> **Guided denoising in Eq 6**: Equation 6 states Tweedie’s identity for a single $x_t$ at time $t$. Indeed, we apply this equation at every step of ODE integration as shown in Algorithm 1, 2 and 3.
>
> **What is the joint distribution $q(x_1, y)$**: In context of this work, this joint distribution is defined over both clean images and distorted images (e.g. blurry images, masked images, downscaled images etc.). In our problem setup, we can evaluate this as $q(x_1, y) = q(x_1) q(y | x_1)$ where $q(y | x_1)$ is assumed known.
>
> **What is $q$ and $q’$ in Section 3.1**: $q$ and $q’$ refer to Gaussian probability paths of two different models (flow or diffusion). Specifically, following the definition stated in the previous parts of this response, both $q$ and $q’$ denote data to noise process $q(x_t | x_1, y)$ or $q’(x_t | x_1, y)$ but for two different models (and hence we use $q$ and $q’$ to make this distinction). This section proposes a way to convert between probability paths of diffusion models and flow models. This enables us to use a pretrained diffusion model for conditional OT sampling and vice versa.
>
> **Clarification for Lemma 1**:  Lemma 1 states that “For Gaussian probability path $q$ given by Eq. 5, the optimal solution for $\widehat{v}(x_t, y)$ is known given $\mathbb{E}_q[x_1 | x_t, y]$, and vice versa.”. The only assumption is that we have a Gaussian probability path $q$ for a flow model. As stated in the lemma, under this assumption, we know the optimal vector field $\widehat{v}(x_t, y)$ that minimizes conditional flow matching loss given $\mathbb{E}_q[x_1 | x_t, y]$ and vice-versa. The vector field indeed has a closed form solution. We refer the reviewer to the proof of Lemma 1 in the Appendix A for further details.

---

> ### Author Response · Authors · 2024-04-15
>
> **Clarification for Lemma 2 and why two alphas**:  Lemma 2 provides a way to translate between arbitrary affine Gaussian probability paths $q$ and $q’$ As stated in the lemma, these paths are parametrized as $q(x_t|y, x_1) = \mathcal{N}(\alpha_t x_1, \sigma_t^2 I)$ and  $q'(x_t|y, x_1) = \mathcal{N}(\alpha'_t x_1, \sigma_t^{'2} I)$. These two $\alpha_t$ and $\alpha_t’$ can indeed correspond to two different timesteps but for different probability paths $q$ and $q’$.
>
> The method for translating between affine Gaussian probability paths involves two key steps:
> 1) Evaluating the time where the Signal-to-Noise Ratio (SNR) of the two probability paths $q$ and $q’$ match.
> 2) Using the pretrained flow model/diffusion model to predict $\hat{x}_1$, where the model uses the converted time from step (1) and the noisy image is also rescaled (See Eq. 9 for the scaling factor.).
> This conversion enables us to use a pretrained diffusion model for conditional OT sampling and vice versa. Similarly, we can also use this conversion to do VP-ODE sampling with a pretrained conditional OT model. Overall, this conversion allows us to use the same model checkpoint for different linear inversion methods and ensures a fair comparison of all the methods.
>
> We would like to clarify that we are not using a “fast” schedule to sample a “slow” diffusion model. We also do not have a single diffusion model with two schedules.
>
> **Distinction between Eq 6 and Eq 11**: Note that we use the term “adaptation” instead of “correction”. There is no discrepancy between Eq 6 and Eq 11. Eq 6 uses the Tweedie’s identity to provide an adaptation term at a _single_ timestep so that we can use an unconditional diffusion model for conditional sampling. This adaptation is applied at all timesteps $t$. On the other hand, Eq 11 provides the adaptation term to adapt an unconditional vector field to a conditional vector field while solving the flow ODE. As shown in Algorithm 1, we indeed apply this adaptation at each step of ODE integration. Note that during sampling, both diffusion models and flow models estimate $E_q[x_1 | x_t]$ at each step, and we add the adaptation term at each step.
>
> **Clarification about measurements in Equation 12**: Equation 12 is our approximation for solving linear inverse problems for flow models with Gaussian probability paths. $q(y | x_1) = \mathcal{N}(A x_1, \sigma_y^2 I)$ is our problem set up for linear inverse problems. This is known because we know the measurement matrix $A$ as well as $\sigma_y$ as stated in the beginning of Section 3. There can indeed be more complex ways to generate non linear measurements $y$ from $x_1$. We leave these more complex settings as future work and in this work we only focus on linear inverse problems. Despite its simplicity, the problem setup of linear inverse problems captures interesting types of distortions such as deblurring, inpainting, super-resolution, denoising etc as shown in our experiments.
>
> **What is $q^{app}$ and what is its relation to q?** : $q^{app}(y|x_t)$ refers to the  _approximation_ of $q(y|x_t)$ (which is unknown and intractable) that is used for solving inverse problems by using the update in Theorem 1.  The graphical model with $y$, $x_1$ and $x_t$ is of form $y \leftarrow x_1 \rightarrow x_t$. Thus computing $q(y | x_t)$ involves marginalization over $x_1$ which is intractable. It also involves computing $q(x_1 | x_t)$ which is computationally expensive (check our next answer to know the details of how we approximate it).
>
> Note that we approximate $q^{app}(y|x_t)$ by following $\Pi$GDM and setting it as $q^{app}(y|x_t) = \mathcal{N}(A\widehat{x_1}(x_t), \sigma_t^2I + r_t^2 AA^\top)$ where $r_t$ is some time dependent standard deviation of $q(x_1 | x_t)$. We have updated the text (highlighted in green) in Section 3.2 to improve clarity and we encourage the reviewer to read the updated text. We reiterate that cannot use the original $q(y | x_t)$ because it is unknown and intractable.
>
> **Why approximate $q(x_1 | x_t)$**?:  We note that evaluating $q(x_1|x_t)$ is computationally expensive and even intractable. To draw samples from it, we need to do posterior inference by sampling from the entire diffusion sampling chain which is expensive, especially as $t \rightarrow 0$, making this evaluation very slow.  Therefore, many existing works approximate $q(x_1 | x_t)$ as well as approximate $q(y  | x_t)$  with heuristics. For instance, Diffusion Posterior Sampling (DPS), uses $q(y | x_t) = q(y | \hat{x}_0 = \hat{x}_t)$ (See Equation 11 in ). Reconstruction guidance [2] makes isotropic Gaussian assumption to estimate $q(y | x_t)$ which is worse than the assumption made here.
>
> [1] Chung, Hyungjin, et al. "Diffusion posterior sampling for general noisy inverse problems." arXiv preprint arXiv:2209.14687 (2022).
>
> [2] Ho, Jonathan, et al. "Video diffusion models." Advances in Neural Information Processing Systems 35 (2022): 8633-8646.

---

> ### Author Response · Authors · 2024-04-15
>
> **What is $\gamma_t$?**: $\gamma_t$ is a time-dependent scaling factor that helps in better approximation of  $q^{app}(y|x_t)$. We empirically find that  $\gamma_t=1$ works well for conditional OT flow models. We empirically find that $\gamma_t = \sqrt{\dfrac{\alpha_t}{\alpha_t^2 + \sigma_t^2 }}$ works well for VP-ODE paths. Please see Appendix C for more details.
>
> **How is $r_t^2$ derived?** :  $\Pi$GDM (See Section A.3 in their paper) derives $r_t^2$ by assuming that $q(x_1) = \mathcal{N}(0, I)$. By Bayes’ rule, we know that $q(x_1 | x_t) \propto q(x_1) q(x_t | x_1)$. In this work, $q(x_t | x_1) = \mathcal{N}(\alpha_t x_1, \sigma_t^2 I)$. Thus, we can write  $q(x_1 | x_t) \propto q(x_1) q(x_t | x_1) = \mathcal{N}\left( \dfrac{\alpha_t x_t}{\sigma_t^2 + \alpha_t^2} , \dfrac{\sigma_t^2}{\sigma_t^2 + \alpha_t^2}\right) I$.
>
> **Validity of $\Pi$GDM approximation for $r_t^2$** : As rightly pointed out by the reviewer, the distribution of data in real world is complex and assuming $q(x_1) = \mathcal{N}(0, I)$ is not realistic. We adapted this approximation for $r_t^2$ as $\Pi$GDM reported good results with this assumption.  $\Pi$GDM also further motivated this approximation by pointing out that the resulting $q(x_1 | x_t)$ has reasonable behavior near $t=1$. We agree with the reviewer though that  there are avenues to come up with better approximations of $r_t^2$ (and correspondingly improved approximations for $q^{app}(y | x_t)$). We leave this exploration as future work and note that our method will still apply with alternative $q^{app}(y | x_t)$.
>
> **$y$ in Eq 17 in the latest draft (Eq 14 in the older version)**: We first note that in our entire paper, $y$ always refers to measurements (noisy or noiseless), except in the background section where we include more general definitions of conditional diffusion models and conditional flow models, and $y$ could be any conditioning including measurements of an image (noiseless or noisy). Indeed, $y$ is of the same shape as $x_1$ in Equation 17. In tasks like super-resolution where $y$ is smaller than $x_1$, we rescale $y$ to ensure that it is of the same shape as $x_1$.
>
> **Rows 6 and 7 in Algorithm 1**:  Row 6 corresponds to the value of the gradient $\nabla_{x_t}q(y|x_t)$ after plugging in $\Pi$GDM adaptation. Row 7 corresponds to Eq 12 after plugging in $\alpha_t = t$ and $\sigma_t = 1-t$ for the conditional OT probability path.
>
> **What is the measurement matrix A and how is it computed?**: Our problem setup of solving linear inverse problems assumes that we know the measurement matrix A. This is a standard assumption made in this problem setup. Matrix A applies some transformation to the clean image $x_1$. For instance, in case of Gaussian (de)blurring, it applies some blur kernel to the input $x_1$ so that we get a blurry measurement of the input. Measurement matrix A is specific to a given inverse problem. For instance, for inpainting, A is a binary mask. We use open source implementation available in https://github.com/NVlabs/RED-diff/ for computing measurements for different linear inverse tasks. We do not train the matrix A.

---

> ### Author Response · Authors · 2024-04-15
>
> **Distinction between our method and $\Pi$GDM**
> We first emphasize that our method for solving linear inverse problems, as presented in Algorithm 1, is _not_ $\Pi$GDM that uses adaptive weighting.  We expand upon and enumerate the differences below. Instead, Algorithm 1 outlines our method for solving inverse problems via conditional Optimal Transport (OT) paths with a pretrained diffusion model. As demonstrated by our experiments, images restored via the conditional OT probability path outperform both VP-ODE paths and $\Pi$GDM in noisy linear inverse settings.
>
> **Key differences between our method and $\Pi$GDM**:
> - **Different sampling paths for solving inverse problems**:   Our algorithm employs conditional OT probability paths as opposed to VE-SDE/VP-SDE probability paths employed by $\Pi$GDM for solving linear inverse problems. VE-SDE and VP-SDE models are characterized by specific schedules for $\alpha_t$ and $\sigma_t$, and these parameters are coupled through the use of a $\beta_t$ schedule. In contrast, the OT schedule uses functions for $\alpha_t$ and $ \sigma_t$ that cannot be modeled by either VE-SDE or VP-SDE. This key distinction in our approach of using conditional OT probability path enables our method to outperform diffusion paths across a wide range of noisy linear inverse problems and datasets, as demonstrated in our paper.
> - **Early starting**: Our algorithm starts sampling from time $t > 0$ while $\Pi$GDM starts at $t=0$ (Gaussian noise).  As can be seen in the supplementary material in Figure 7 and Figure 8, early starting is beneficial to our method, but not beneficial to $\Pi$GDM as can be seen in Figure 28.
> - **Reduced hyperparameter tuning**: The use of conditional OT path significantly reduces the amount of manual hyperparameter tuning. The valid  hyperparameters in our case are the choice of start time and number of function evaluations (NFEs), that we keep constant for all tasks and datasets on OT-ODE.
>     -Our method exhibits more stable performance across a wide range of NFEs compared to both VP-ODE and  $\Pi$GDM. Further, our method outperforms both VP-ODE and $\Pi$GDM at smaller NFEs as shown in Figure 6.
>     -In contrast to $\Pi$GDM, our method does not incorporate any adaptive weights for solving linear inverse problems via conditional OT paths. We find that the adaptive weights used in $\Pi$GDM hurt performance on conditional OT paths.
>
> - Experiments with our VP-ODE method, an alternative to our OT-ODE also introduced in  our paper, could be considered similar to that of the $\Pi$GDM algorithm with $\eta=0.0$.  We point out though that $\Pi$GDM explored different values of $\eta$ in their Appendix (see their Table 6 and 7) and found that $\eta=1.0$ was superior at 100 NFEs, which is the number of NFEs used by $\Pi$GDM in our paper.  This is completely different from our results where our VP-ODE method performed better than $\Pi$GDM with the recommended $\eta=1.0$.  So while there are similarities, our VP-ODE method and $\Pi$GDM at $\eta=0.0$ are different. Further, we emphasize that our recommended OT-ODE method is even more conceptually different and performs even better than our VP-ODE method.
>
> Overall, our algorithm for solving linear inverse problems works for flow models with arbitrary affine Gaussian probability paths (Algorithm 1,2,and 3) while $\Pi$GDM is specifically designed for VP-SDE/VE-SDE paths.
>
> We are happy to answer any other questions that you might have.

---

> ### Comment · Reviewer_GG8v · 2024-04-22
> **response**
>
> Thanks for the clarifications and revisions. These address all of my concerns, except for a final one: The second bullet claims that you contribute a method to transform between diffusion and flow models. Can you argue why is this a novel contribution, or in in which precise way is this novel? Surely the literature has derived these connections already.

---

> > ### Author Response · Authors · 2024-04-23
> >
> > We pointed out in the existing Section 3.1 that the conversion between flows and diffusion, and across Gaussian probability paths was known. However, it was not emphasized or specifically noted in past literature that path conversion may require an adjusted initialization procedure.  To emphasize this, in the latest version, we moved conversion between flows and diffusion to preliminaries, changed the second contribution to emphasize the adjusted initialization procedure, and included more detail in the revised Section 3.2 on converting between probability paths.

---

> > > ### Comment · Reviewer_GG8v · 2024-04-23
> > >
> > > Thanks for the revisions. I'm happy with the paper.

---

### Review · Reviewer_XMQw · 2024-03-25

**Summary Of Contributions:**

This paper presents a method for solving inverse problems using flow models. It presents theory and algorithms for adapting a pretrained flow-based generative model to approximately sample from the posterior in an inverse problem given measurements and forward operator, using techniques from diffusion-based inverse solvers (specifically, PIGDM). The method is problem-agnostic (able to use the same model to solve different inverse tasks without retraining) and requires less hyperparameter tuning than diffusion-based approaches.

**Audience:**

Yes

**Claims And Evidence:**

No

**Requested Changes:**

The reviewer would suggest the authors make the following changes to make their claims well-supported.

Make their claims more accurate (see weakness 1).

Improve writing on methodology (section 3.2).

Make sure the performance of compared methods are sufficiently finetuned and optimized. Make sure the results are representative of state-of-the-art and the comparison is fair.

**Strengths And Weaknesses:**

**Strengths**

This is an interesting paper that adapts techniques in diffusion-based posterior sampling to flow models. The proposed method can reduce the need for hyperparameter tuning, which is beneficial in practice. The conversion formula between diffusion models and flow models looks novel and interesting.


**Weaknesses**

1. In the reviewer's humble opinion, some claims made in the paper are too absolute or not accurate.

- "without training"/"training-free"

The authors stated multiple times that their method solves inverse problems "without (any) training" or is "training-free", for example,
"The challenge of solving noisy linear inverse problems without any training has been tackled in many ways..."

The reviewer does not completely agree with this wording. The method is still data-driven and requires training a generative model, so it's not "training-free". It might be better to use phrases like "without retraining", "problem agnostic", "model agnostic", or "not trained on specific inverse problems", following conventions in inverse problem literature (e.g., see the PIGDM paper.)

- "no hyperparameter tuning", for example,

"The algorithm is simple, stable, and requires no hyperparameter tuning when used with conditional OT probability paths."

The reviewer is not sure if this is accurate. Based on "Methods and baselines" and appendix, the updating rule of gamma_t, the initial time step and NFEs are all tunable.


2. The performance of compared methods seems not as good as they should be. For example,

- Figure 3: row 1, columns (e) and (f); row 2, column (e). These results either exhibit strong (rippling) artifacts or are oversmoothed. The authors might want to double check the hyperparameters for these methods (e.g., step sizes) to make sure the results are representative of these methods' true performance. In the reviewer's opinion, these results do not exhibit the representative performance of the state-of-the-art for these tasks.

- Similar comments for Figure 4: Row 1, column (e) (strong local artifacts); Figure 5, column (f) (over-smoothed). These are not representative of a state-of-the-art method.


3. Most of Section 3.2 is hard to follow for the reviewer. Some statements are vague, inexact, or not self-contained. Some notations seem to be borrowed from another paper (PIGDM) but are not defined clearly in this paper. For example,

- Eq 12.: q^{app} seems undefined. What is the relationship between q^app and q?

- Following the above, "In general, we view adaptive weights γt ̸= 1 as an adjustment for error in qapp(y|xt)." What error is there in q^app?

- "For qapp(y|xt), we generalize ΠGDM to any Gaussian probability path described by Eq. 5 via updating r_t^2." Again, r_t^2 seems undefined.

- "We choose r2t by following ΠGDM’s derivation, noting that ..." How is r2t chosen exactly? What is the PIGDM derivation exactly?

- "if q(x1) is N(0, I) and q(xt|x1) is N(αtx1, σ2(\sigma_t I), then rt2 = sgimat2 / (sigmat2 + alphat2)" How is this (eq 13) derived? Why is it reasonable to assume the data distribution q(x1) is a Gaussian? (The distribution of most real-world data is more complext than a Gaussian.)


In short, it is clear at a high level that the authors are trying to adapt PIGDM to flow model. However, the writting is a little bit informal and inconcrete, making the exact methodology unclear. Perhaps a reader very familiar with the notations in PIGDM paper can understand this fairly easily, but for a general reader, this paragraph is a little bit hard-to-follow.



4. "Vector field correction"/"correcting the vector field of flow models"

The reviewer feels that the term "correction" is a little bit misleading. "Correction" implies the original vector field has some "error" and the proposed method tries to fix the error. However, based on the reviewer's understanding, the original vector field is correct, but intended for unconditional sampling. The method aims to adapt/modify it to a new vector field enabling sampling conditioned on the measurements y.


Other comments:

- "For Gaussian deblurring, we apply Gaussian blur kernel of size 61 × 61 with intensity value 1 for ImageNet-64 and ImageNet-128, and 61 × 61 with intensity value 3 for AFHQ." The 61x61 kernel size is too large for blurring. Is it a typo? What does intensity value mean here?

- In Algorithm 1, what's the final output of the algorithm? After obtaining the "corrected" vector field, how is xt updated? Also, it might be better to denote the initial time point as t0, rather than t.

- Figure 2: The reviewer is not sure if connecting the dots is appropriate, since in this figure there is no ordinal relation in x.

- "Under Gaussian probability paths, the two terms are related as Tweedie’s identity (Robbins, 1992) expresses..." Sentence reads wrong.

- "In contrast, the inpainted regions generated by RED-Diff tend be blurry and less semantically meaningful." Should be "tend to"

---

> ### Author Response · Authors · 2024-04-15
>
> We thank the reviewer for their valuable comments and suggestions. We are happy to learn that the reviewer finds our method for solving inverse problems and our approach for the conversion between flow models and diffusion models interesting. We have tried our best to address the questions and concerns below. We have also incorporated some of your suggestions and have updated the text (highlighted in green) to improve clarity in the latest draft.
>
> **Use of the term “Training-free”**:
> By training-free, we mean that our method does not require any re-training / fine-tuning of the pretrained flow model or diffusion model on specific inverse problems. Our method indeed uses a pretrained flow model / diffusion model and corrects the vector field appropriately to solve inverse problems. We apologize if our use of the term “training-free” resulted in any confusion. We have updated the text to clarify this.  We have used the term “without fine-tuning” to emphasize this point. The suggested alternatives like “problem-agnostic” and “model-agnostic” do not emphasize the point that the method does not require any further training/fine-tuning. We therefore retain the term “training-free” for now.
>
> **Use of the term “no hyperparameter tuning” in conclusion**
> It is indeed true that $\gamma_t$, the initial time step and NFEs are all tunable, however, we keep the values of these hyperparameters fixed for all our experiments on all tasks and datasets with OT-ODE, and therefore we claim that this method does not require problem-specific hyperparameter tuning. We however agree with the reviewer, and have updated this line in conclusion to reflect that our method needs minimal hyperparameter tuning instead of no hyperparameter tuning. We note that previous state-of-the-art methods like Diffusion Posterior Sampling and RED-Diff need problem-specific and dataset-specific hyperparameter tuning, and the need for minimal hyperparameter tuning is one notable advantage of our method over previous methods.
>
> **Performance of baselines**
>
> We have tried to summarize our efforts to ensure faithful replication of these prior works.
>
> - __Code__
> To replicate the results of $\Pi$GDM and RED-Diff, we use the open source code available from the authors on github (https://github.com/NVlabs/RED-diff/tree/master/algos). We made minimal changes to this code to use continuous-time diffusion models and flow models.
>
> - __Hyperparameter tuning__:
>     - For RED-Diff, we tried our best to find optimal hyperparameters per task and per dataset for both noiseless and noisy settings. We have included these results in Appendix F. We use Adam optimizer and use the momentum pair (0.9, 0.99) similar to the original work. Further, we use an initial learning rate of 0.1 for AFHQ and ImageNet-128, as used in the original work, and learning rate of 0.01 for ImageNet-64, which we found to work better in practice.
>     - For the noiseless case of $\Pi$GDM, we use the optimal hyperparameters reported in the paper and for the noisy setting, we tuned both the scaling factor $r_t^2$ and starting time. We found an alternate value of scaling factor $r_t^2$ that performs better in practice. Please refer to Appendix F for further details.
>
> - __Experimental settings__:
> We note that $\Pi$GDM includes results on uniform kernel deblurring for the noiseless case (Table 5, Table 12 in $\Pi$GDM paper (Open-review version https://openreview.net/pdf?id=9_gsMA8MRKQ)). On the other hand, we report results for noisy Gaussian deblurring in Figures 3.
>
>     - In our experiments, we find that $\Pi$GDM performs very well in the noiseless setting. Please refer to Tables 10 -15 in the appendix for the performance of $\Pi$GDM in the noiseless setting, and Figures 18-24 in the appendix for qualitative results. However, in the setting for solving noisy inverse problems, the performance deteriorates (Fig. 3-5). The results reported by Song et al. 2022 for $\Pi$GDM on noisy inpainting indicate this as well (See the results in Table 3 of $\Pi$GDM paper for noisy inpainting). It is unclear if Song et al. 2022 use noisy settings for any other tasks as it is not indicated explicitly in the paper.
>     - Qualitative results in RED-Diff also indicate that this method suffers from some oversmoothing. Please see the first row, second column of Figure 2 in Mardani et al, 2023 (https://arxiv.org/pdf/2305.04391.pdf). As indicated in Figure 14 in this paper, it does seem that the choice of optimization algorithm matters in RED-Diff, and can affect the observed smoothing. In our efforts to replicate this work, we used Adam optimizer, which is the standard setting considered in the original work. Note that the artifacts in $\Pi$GDM are also observed by Mardani et al, 2023. See first row, third column of Figure 2 in their paper.
>
> Therefore, we believe that our reported observations in Figures 3, 4 and 5 are accurate to the best of our ability for the experimental setting we consider in our work.

---

> ### Author Response · Authors · 2024-04-15
>
> **Use of “correction-term”**: The reviewer has a correct understanding that the original vector field is correct, but intended for unconditional sampling. The method aims to adapt/modify it to a new vector field enabling sampling conditioned on the measurements y.  We adjusted the title of Section 3.2 for clarification. We have also updated the term “correction” to “adaptation” everywhere.
>
> **Further clarification of Section 3.2**
> Thank you for pointing out the missing details. We have updated Section 3.2 to improve clarity of notation, sentences and mathematical expressions used in this section. The corrected text is  highlighted in green. We also answer the questions below:
>
> - __What is $q^{app}$ and what is its relation to q?__: $q^{app}(y|x_t)$ refers to the  _approximation_ of the ground truth (but unknown and intractable) $q(y|x_t)$ that is used for solving inverse problems.  Note that we approximate $q^{app}(y|x_t)$ by following $\Pi$GDM and setting it as $q^{app}(y|x_t) = \mathcal{N}(A\widehat{x_1}(x_t), \sigma_t^2I + r_t^2 AA^\top)$ where $r_t$ is some time dependent standard deviation of $q(x_1 | x_t)$. Please also check our answer to the next question for further clarification.
>
> - __What is error the in $q^{app}(y|x_t)$?__ :  We first note that  $q(y|x_t)$ is unknown when we have a pretrained unconditional flow model. Further, $q(y|x_t)$ is also intractable to compute as $q(y|x_t) = \int_{x_1} q(y | x_1) q(x_1 | x_t) dx_1$ which involves marginalization over $x_1$ and the second term $q(x_1 | x_t)$ is computationally infeasible. We therefore need to approximate the second term and we follow $\Pi$GDM approximation and set it to $q(x_1 | x_t) \approx \mathcal{N}(\widehat{x_1}(x_t), r_t^2I)$ where $r_t$ is some time dependent standard deviation. We then approximate $q^{app}(y|x_t)$ based on this as described in the previous answer. Therefore, there is an error in  $q^{app}(y|x_t)$.
>
>
> - __What is $r_t^2$?__: We approximate $q(x_1 | x_t) \approx \mathcal{N}(\widehat{x_1}(x_t), r_t^2I)$ where $r_t$ is some unknown time dependent standard deviation.
>
>
> - __How is $r_t^2$ chosen and how is it derived?__ :  $\Pi$GDM (See Section A.3 in their paper) derives $r_t^2$ by assuming that $q(x_1) = \mathcal{N}(0, I)$. By Bayes’ rule, we know that $q(x_1 | x_t) \propto q(x_1) q(x_t | x_1)$. In this work, $q(x_t | x_1) = \mathcal{N}(\alpha_t x_1, \sigma_t^2 I)$. Thus, we can write  $q(x_1 | x_t) \propto q(x_1) q(x_t | x_1) = \mathcal{N}\left( \dfrac{\alpha_t x_t}{\sigma_t^2 + \alpha_t^2} , \dfrac{\sigma_t^2}{\sigma_t^2 + \alpha_t^2} I \right)$.
>
>
> - __Validity of $\Pi$GDM approximation for $r_t^2$__ : As rightly pointed out by the reviewer, the distribution of data in real world is complex and assuming $q(x_1) = \mathcal{N}(0, I)$ is not valid. We adapted this unrealistic approximation as $\Pi$GDM reported good results with this assumption. We agree with the reviewer and acknowledge that there is a gap here to come up with better approximations of $r_t^2$. We leave this exploration as future work.
>
> **Size of blur kernel**:  We use the experimental setting used in Diffusion Posterior Sampling for Gaussian deblurring where they used a blur kernel of size $61\times61$ and standard deviation $3$ on FFHQ $256\times256$ images. On images with smaller resolution, we reduce the value of standard deviation to 1. Finally, “intensity value” should be standard deviation instead. Thank you for pointing this out.
>
> **Further details of Algorithm 1**:
> - _Output of Algorithm 1_ : The output is the solution of ODE after integration which is also the corrected image. We will explicitly indicate this. Further, we have updated Algorithm 1 to use $t_0$ instead of $t$.
>
>
> - _How is $x_t$ updated with the corrected vector field?_ :  We can use any standard numerical method for ODE integration to solve the ODE with the corrected vector field. For example, if we use Euler method for integration, the update will be of the form $x_{t + \Delta t} = x_t + \Delta t \widehat{v}_{\text{adapted}}$ where $\Delta t$ is step size of time schedule discretization.
>
> **Plots of Figure 2**:  We agree that the styling choice for plotting figures might seem counterintuitive, and bar plots might be a more reasonable choice. We decided to plot it in this way because we felt the results are easier to interpret. The corresponding bar plot for Figure 2 would have $13 \times 4=52$ vertical bars in each plot which can be tricky to interpret. On the other hand, in the current plot, it is relatively easy to interpret that the red line, that corresponds to OT-ODE, outperforms other methods by always being above (or below) the competing methods.
>
> **Grammatical errors**: Thanks for pointing this out! We have updated the text with corrected sentences.
>
> We are happy to answer any other questions that you might have.

---

> > ### Comment · Reviewer_XMQw · 2024-04-28
> > **Reviewer's Comment**
> >
> > Thank the authors for answering my questions and concerns and further clarifying their contributions. The paper, especially the methodology section, has become substantially more clear and self-contained.
> >
> > I have one more comment. There is a line of works on using flow models for inverse problems. See, e.g.,
> > https://proceedings.mlr.press/v139/whang21b/whang21b.pdf
> > https://openaccess.thecvf.com/content/ICCV2023/papers/Hong_On_the_Robustness_of_Normalizing_Flows_for_Inverse_Problems_in_ICCV_2023_paper.pdf
> > https://arxiv.org/pdf/2003.08089
> >
> > Those works are not mentioned in the current paper.
> > I'm wondering if the authors could comment on the connection between their work and those previous works, to further clarify their contributions and enhance the literature review part of this paper.

---

> ### Author Response · Authors · 2024-04-29
>
> __Difference of the proposed method from prior works on solving inverse problems with flows__:
>
> We have updated the paper with more extensive related work in Appendix G. Below we highlight the differences between our work and these prior works.
>
> - _On the robustness of normalizing flows for inverse problems in imaging by Hong et al. 2023_: Continuous normalizing flows (CNFs) used in this work are trained by optimizing the maximum likelihood objective. To optimize this objective in a tractable manner, these CNFs have restricted neural network architectures. See Appendix F for more details. The work by Hong et al. 2023 tries to understand and mitigate the cause behind the artifacts observed in the outputs of CNFs when they are used for solving inverse problems. They attribute these artifacts to the “exploding inverse” caused by affine coupling layer that is commonly used in architectures of CNFs. We note that our proposed method does not use any restricted architecture as we use pretrained flow models that are trained with the flow matching objective. Therefore, the problem statement considered by Hong et al. 2023 is very different from our problem setup, and the limitations observed in conventional CNFs due to restricted architectures are not applicable to us.
> - _Composing Normalizing Flows for Inverse Problems by Whang et al. 2021_:  This work considers the problem of conditional sampling with an unconditional flow model which is similar to our problem setup. However, this work is also based on CNFs which are trained on the maximum likelihood objective and have restricted architecture. Conditional sampling with these CNFs is hard and this work proposes an alternate framework for approximate inference that makes this inference tractable. Specifically, this work composes the unconditional flow model with another flow model, and derives a variational objective on this composed model. The composed model is optimized on this loss. In contrast, our approach leverages unconditional flow models trained with flow matching  (or converted diffusion models), and is completely training free. We draw inspiration from diffusion models ($\Pi$GDM) to approximate the conditional posterior.
> - _Solving Inverse Problems with a Flow-based Noise Model by Whang et al. 2021_:  This work considers a more general problem setup of solving noisy inverse problems $y = f(x) + \epsilon$ where the forward operator $f(x)$ can be non-linear and $\epsilon$ is an additive noise. The proposed solution to solve this more general inverse problem is based on MAP estimation with flow prior. This problem setting is very different from our work, where we consider the problem of solving linear inverse problems via posterior sampling. Further, we do not do any MAP based optimization but use gradient-based adaptation during the sampling process of flow models to approximate posterior sampling.

---

### Review · Reviewer_RAno · 2024-04-17

**Summary Of Contributions:**

This work adapts the $\Pi$GDM framework for training-free inverse problem conditional sampling of diffusion models for solving linear inverse problems to a restricted class of flow matching models, namely those with Gaussian probability paths, that intersect exactly with those achievable by a similarly restricted class of diffusion noising processes.

There are some experiments performed to show this method performs reasonably on tasks compared to a diffusion counterpart.

**Audience:**

Yes

**Claims And Evidence:**

Yes

**Requested Changes:**

- I believe a refining of 3.1/3.2 to be slightly less flow/diffusion specific would be beneficial, and then to show how the derived results apply to both cases would be a good refinement.

- I think making 3.2 more standalone with less reliance on referring to $\Pi$GDM would benefit readability, and including a short section on the similarities afterwards.

- The paper presents itself as focusing on OT flows specifically. The results derived apply to all diffusions and flows with Gaussian probability paths. I would recommend that the authors separate out the theoretical claims, to show this method has wider applicability than to OT flows, and the specific findings in the experiments section that OT flows perform best. I think this will expand your readership scope and appeal a great deal. I would also highlight that your results can be applied to diffusion models as well, under the Gaussian probability path assumption. For example, the first contribution

    "We present a training-free approach for solving linear inverse problems with pretrained conditional OT flow models that incorporates the ΠGDM gradient adaptation, proposed for diffusion models, to flow sampling."

(incidentally, this sentence looks malformed and needs fixing), could be stated much more generally. For example, one could say

    "We derive a training-free approach to solving linear inverse problems that can be applied to any diffusion or flow-based model, under the assumption that they use Gaussian probability paths, that extends the work of $\Pi$GDM to this more generic setting".

The empirical claim

    "We demonstrate that images restored via our algorithm using conditional OT probability paths ..."

could be restated more widely as

    "We empirically test a range of different flow and diffusion models with different Gaussian probability paths. Empirically we find that OT probability paths in flow matching models match or outperform the diffusion setting of $\Pi$GDM and RED-Diff, demonstrating the usefulness of this more framework for general Gaussian probability paths."

These are meant as examples, not hard suggestions.

**Strengths And Weaknesses:**

Strengths:
- The paper is reasonably well written and clean.
- The main experimental section is reasonably convincing, and ablation studies quite comprehensive.

Weaknesses:
- The theoretical contributions appear to reasonably minor. The main analysis flow pretty directly from $\Pi$GDM with some small variation in the terms to encompass Gaussian probability paths.
- While I appreciate the presentation, I do wonder if there is a better way of presenting the main derivation. It relies on reference to $\Pi$GDM at many points, rather than standing alone. A better format may be to refer to $\Pi$GDM less for critical steps and write a self containing presentation. To be more concrete, removing the constant references to $\Pi$GDM between the start of 3.2 and equation 16, making this section self consistent, and just after this writing a subsection on "Relationship to $\Pi$GDM" to make clear the links to this work.

- It also seems possible to improve the presentation in sections 3.1 and 3.2 to make the derivations less dependent on the flow matching point of view, as the derived method could be applied easily to diffusion models with Gaussian probability paths also.
- The contribution "We offer a way to convert between flow models and diffusion models" is well known and not novel to this work, nor is it to be pedantic true - there are many flow models that do not have a diffusion counterpart. Showing how the conditional sampling framework derived applies to both (under the gaussian probability path assumption) would be novel.

---

> ### Author Response · Authors · 2024-04-23
>
> We thank the reviewer for their kind and constructive feedback. We have updated our draft as per the suggestions and have uploaded a new version of the paper on OpenReview. The updated text in the latest version is highlighted in blue and the changes in the previous revision are highlighted in green.
>
> 1. __Refinement of Section 3.1 and 3.2__: We have updated Section 3.1 and 3.2 as per reviewer’s suggestions to make these sections less reliant on $\Pi$GDM. We have also updated the text to highlight that this method is applicable to both diffusion models and flow models with affine Gaussian probability paths. We have also inlined all the lemmas, theorems and proofs in Section 3 to avoid an overemphasis on theoretical contribution.
>
> 2. __Updated Contributions__: We have updated our contributions which we have stated below for your reference.
> - We present a training-free approach to solve linear inverse problems that can be applied to any continuous-time diffusion or flow model under affine Gaussian probability paths that extends $\Pi$GDM gradient adaptation to this more generic setting.
> - We explain subtleties in converting models between different affine Gaussian probability paths.  Specifically, we enable the use of pre-trained continuous-time diffusion models with conditional OT probability paths by an adjusted initialization procedure.
> - We demonstrate that images restored via our ODE algorithm using conditional OT probability paths have perceptual quality that is largely on par with, or better than that achieved by diffusion probability paths, and other recent methods like $\Pi$GDM and RED-Diff, without the need for problem-specific hyperparameter tuning.
>
> 3. __Miscellaneous grammatical errors__: Thank you for pointing this out! We have updated the text accordingly.

---

### Author Response · Authors · 2024-04-23

We thank all the reviewers for their valuable feedback, which has helped us improve our draft.

We have made extensive updates to our draft, especially Preliminaries and Section 3, based on the reviewers' suggestions. We have also inlined all the lemmas and the theorem along with their proofs in the main paper.  We have uploaded two revisions: changes in the first revision are highlighted in green, and those in the second revision are highlighted in blue. We believe these changes have clarified and expanded upon our relationship to past research, our contributions, and methodological development.

Below, we have provided detailed responses to all the individual questions raised.

--- Update ---

We have uploaded another revision of our paper that includes more extensive related work in Appendix G as per suggestions of Reviewer XMQw.

---

### Decision · Action_Editor_CY1d · 2024-05-23

**Recommendation:** Accept with minor revision

**Comment:**

The text is almost ready and the revision is merely mentioned to ensure that the suggestions of the reviewers are included in the final version.

Concretely, I encourage the authors to follow **all** the recommendations of the reviewers and appropriately adjust their contributions. For instance, as the reviewers point out, the link between diffusion models and flow models have emerged before. Therefore, it would be recommended to make the claims match the findings in the literature. More importantly though, the text should be finalized by including all the comments and suggestions of reviewers in the regular text.

**Audience:**

Inverse problems have been traditionally a very important task, both in academia and industry, so it is likely that this paper will be relevant to the TMLR audience.

**Claims And Evidence:**

This work focuses on solving inverse problems using pretrained diffusion models. The reviewers agree that the method is reasonable, even if the link of flow with diffusion models have emerged before in the literature. The results are interesting, and using a pretrained model for solving inverse tasks has the additional benefit of reducing the demand for hyper-parameter tuning. Both the claims and the empirical evidence provided in the paper are supported with appropriate visualizations and experiments.